# Constructing ordered and tunable extrinsic porosity in covalent organic frameworks via water-mediated soft-template strategy

Ningning He [1], Yingdi Zou[1], Cheng Chen[1], Minghao Tan[1], Yingdan Zhang[1], Xiaofeng Li [2], Zhimin Jia[1], Jie Zhang[1], Honghan Long[1], Haiyue Peng[3], Kaifu Yu[1], Bo Jiang[1], Ziqian Han[1], Ning Liu[3], Yang Li [1] ✉ & Lijian Ma [1] ✉

As one of the most attractive methods for the synthesis of ordered hierarchically porous crystalline materials, the soft-template method has not appeared in covalent organic frameworks (COFs) due to the incompatibility of surfactant self-assembly and guided crystallization process of COF precursors in the organic phase. Herein, we connect the soft templates to the COF backbone through ionic bonds, avoiding their crystallization incompatibilities, thus introducing an additional ordered arrangement of soft templates into the anionic microporous COFs. The ion exchange method is used to remove the templates while maintaining the high crystallinity of COFs, resulting in the construction of COFs with ordered hierarchically micropores/mesopores, herein named OHMMCOFs (OHMMCOF-1 and OHMMCOF-2). OHMMCOFs exhibit significantly enhanced functional group accessibility and faster mass transfer rate. The extrinsic porosity can be adjusted by changing the template length, concentration, and ratio. Cationic guanidine-based COFs (OHMMCOF-3) are also constructed using the same method, which verifies the scalability of the soft-template strategy. This work provides a path for constructing ordered and tunable extrinsic porosity in COFs with greatly improved mass transfer efficiency and functional group accessibility.

Covalent organic frameworks (COFs) are a class of crystalline, porous materials that possess permanent porosity, which arises from the linkage of organic building blocks through reversible covalent bonds[1–3]. Owing to their well-defined and adjustable pore sizes and ordered distributions, COFs have garnered significant interest in diverse applications such as gas storage, catalysis, adsorption separation, and energy storage[4–8]. Nonetheless, the limited accessibility of functional groups in single micropore (<2 nm) COFs has hindered their widespread utilization. To overcome this limitation, the development of hierarchical porous COFs with additional mesopores or macropores

has gained significant attention, as it can enhance both the mass transfer rate and the accessibility of functional groups.

In recent years, researchers have employed various methods to develop hierarchically porous COFs that exhibit simultaneous microporosity (<2 nm), mesoporosity (2–50 nm), or macroporosity (>50 nm)[9–16]. For example, Banerjee et al. employed an inside-out Ostwald ripening process to prepare hollow COF spheres with hierarchically mesoporous walls and macroscopic macropores, providing increased interaction sites for molecules and enzymes[17]. They also utilized the 3D printing technique to fabricate hierarchically porous

[1]College of Chemistry, Key Laboratory of Radiation Physics & Technology, Ministry of Education, Sichuan University, Chengdu 610064, PR China. [2]Institute of Materials, China Academy of Engineering Physics, Mianyang 621907, PR China. [3]Institute of Nuclear Science and Technology, Key Laboratory of Radiation Physics and Technology of the Ministry of Education, Sichuan University, Chengdu 610064, PR China. ✉e-mail: ly7701850@163.com; ma.lj@hotmail.com

COFs with interconnected macropores, enabling rapid and efficient pollutant adsorption[18]. Very recently, COF aerogels with hierarchic macropores, mesopores, and micropores have also been prepared to increase mass transfer speed[19–23]. Additionally, Liu and coworkers have developed a efficient approach for the synthesis of imine-based hollow COFs with meso- and macropores. This one-pot and one-step strategy, based on dynamic combinatorial chemistry, offers a versatile and straightforward method for the construction of hierarchical porous COFs[24]. However, for all the aforementioned methods, achieving ordered additional mesopores or macropores still remains challenging. The hard template method is an effective strategy to address this challenge by using pre-formed templates to create ordered macropores in COFs[11,16,25]. For instance, in 2023, Lu et al. synthesized COF-300 and COF-303 single crystals with ordered hierarchically microporous and macroporous structures using polystyrene microspheres as hard templates, achieving improved iodine adsorption rates[25]. Unlike the introduction of extrinsic pores, another alternative approach to address this limitation is to design organic structure monomers for the preparation of COFs with intrinsic ordered hierarchical micropores and mesopores. For example, Zhao et al. made a groundbreaking discovery by successfully constructing a COF with a distinctive dual-pore structure[26], following which a series of COFs with ordered hierarchical porosity were fabricated subsequently[27–30]. However, the use of hard template methods for the synthesis of ordered hierarchical pores is often associated with complex synthetic procedures and typically results in the formation of macropores. On the other hand, while monomer design-based approaches have been successful in introducing ordered hierarchical mesopores and micropores into COF structures, the synthesis of these monomers can be challenging. Therefore, developing simpler and more versatile methods for the preparation of ordered hierarchical porous COFs remains a challenge.

The soft-template method was initially proposed by researchers at Mobil for the synthesis of ordered hierarchical porous zeolites[31,32]. Subsequently, various types of amorphous or crystalline porous materials, including metal oxides[33–35], silica[36–39], and metal-organic frameworks (MOFs)[40–44], have been synthesized using this method to achieve ordered hierarchical porosity. In comparison with the design and synthesis of complex monomers or the use of hard templates, the soft-template method improves the scalability and flexibility of the synthesis process by employing readily available amphoteric surfactants as soft templates[45,46]. However, to the best of our knowledge, while surfactants have been employed in the synthesis of certain COFs, their utilization has primarily been limited to leveraging their amphiphilic properties or aiding in the formation of disordered pores[47–50], and there have been no published reports on the utilization of the soft-template method to construct ordered hierarchical porous COFs. We believe that extending this method to COF chemistry will bring a new and more convenient way to achieve faster mass transfer efficiency for COFs. The main challenge is that the soft-template method relies on the unique hydrophobic interactions of amphoteric surfactants in the aqueous phase to form micellar soft templates, whereas most of the solvents used for COF synthesis are organic solvents[51–55]. Additionally, the reversible covalent bonding nature of COFs also complicates the introduction and removal of templates.

Therefore, in this work, we used water as a solvent to introduce two amphoteric surfactants with different alkyl chain lengths, namely dodecyl trimethyl ammonium bromide (DTAB) and octadecyl trimethyl ammonium bromide (OTAB) as the soft templates in the synthesis of COFs with ordered hierarchical microporous/mesoporous structures. To avoid the template being expelled from the COF growth domain due to the strong driving force of crystallization, anionic COFs and ionic-type templates are employed, wherein the soft templates and COF backbone are interconnected through ionic bonds. The use of water as a solvent avoids the random distribution of amphoteric surfactants while ensuring their hydrophobic self-assembly. The

formation of ionic bonds facilitates the introduction and removal of templates. After detailed screening of the template removal method through ion exchange, we successfully removed the templates using hydrogen ions while preserving the crystallinity of the ordered hierarchical porous COFs. Anionic COFs with ordered hierarchical microporous/mesoporous structures (named OHMMCOFs) showed an obvious faster kinetic rate in the adsorption of U(VI) and Th(IV) through ion exchange. Specifically, the OHMMCOF-1 increased by 7.0 and 2.4 times within 5 min, respectively, while the OHMMCOF-2 increased by 19.3 and 4.6 times, respectively. By adjusting the concentration of the template, we were able to regulate the microporous/mesoporous ratio, further improving the adsorption rate of the OHMMCOF-1 to 15.0-fold and 3.5-fold within 5 min for U(VI) and Th(IV), respectively. Moreover, the presence of additional mesopores provided increased availability of functional groups, resulting in an improved adsorption capacity. Surprisingly, we found that, apart from adjusting the mesoporous pore size by changing the amphoteric surfactants with different chain lengths, the addition of different proportions of DTAB and OTAB to the same system allowed for precise control of adjustable mesopore size, which further simplifies the acquisition of additional ordered mesopores in hierarchical microporous/mesoporous COFs. Finally, we also constructed cationic guanidine-based COFs with ordered hierarchically micropores/mesopores (OHMMCOF-3) using the same method, which further demonstrates the universality of soft-template strategy.

## Results

### Synthesis and characterization of OHMMCOF-DTAB/OHMMCOF-OTAB

To obtain COFs with ordered hierarchical micropores/mesopores, we selected smaller building blocks that can construct about 1.8 nm micropores, and after optimization of solvent and catalyst conditions, we successfully constructed off-template pre-products of ordered hierarchical microporous/mesoporous COFs in an aqueous solution at 120 °C. The building blocks and templates used were 2-hydroxy-1,3,5-benzenetricarbaldehyde (Sa), 2,5-diaminobenzenesulfonic acid (DABA), and DTAB/OTAB, and the resulting COFs were named OHMMCOF-DTAB/OHMMCOF-OTAB (Supplementary Sections 1, 3). The crucial step in the synthesis involves dispersing DTAB/OTAB in an aqueous solution, allowing it to self-assemble into spherical micelles with amine bromide at the hydrophilic end (Micelles-spherical). Transmission electron microscopy (TEM) analysis revealed that the size distribution of the spherical micelles formed by the assembly of DTAB and OTAB is predominantly centered around 3.29 and 4.45 nm, while dynamic light scattering (DLS) measurements indicated sizes of 3.62 and 4.85 nm, respectively. As the temperature increased, the spherical micelles underwent further self-assembly to transition into columnar micelles (Micelles-columnar). Subsequently, the addition of DABA facilitated the formation of ammonium sulfonate as a micelle at the peripheral hydrophilic end through ion exchange, leading to the formation of DABA@micelles-columnar (Supplementary Figs. 5–8). The amino groups then underwent reversible condensation with the subsequently added aldehydes, resulting in the formation of COFs (Fig. 1). As a comparison, pristine microporous COFs (namely MPCOF) without any template were also constructed under the same conditions. The crystal structure of the synthesized materials was characterized using powder X-ray diffraction (PXRD), and the detailed spatial structure of template-free MPCOF was simulated with Materials Studio 8.0 (MS) software and Pawley refinement, as shown in Fig. 2A. The PXRD patterns of MPCOF exhibited a series of diffraction peaks at 4.8°, 8.3°, 9.5°, 12.6°, and 26.6°, corresponding to (100), (101), (200), (102), and (001) facets, respectively. MS simulation revealed that MPCOF adopted the eclipsed AA stacking mode. The optimized cell parameters are a = 22.51 Å, b = 3.48 Å, c = 22.73 Å and α = γ = 90°, β = 119.25° in a space group of $Pm$ after Pawley refinement, with

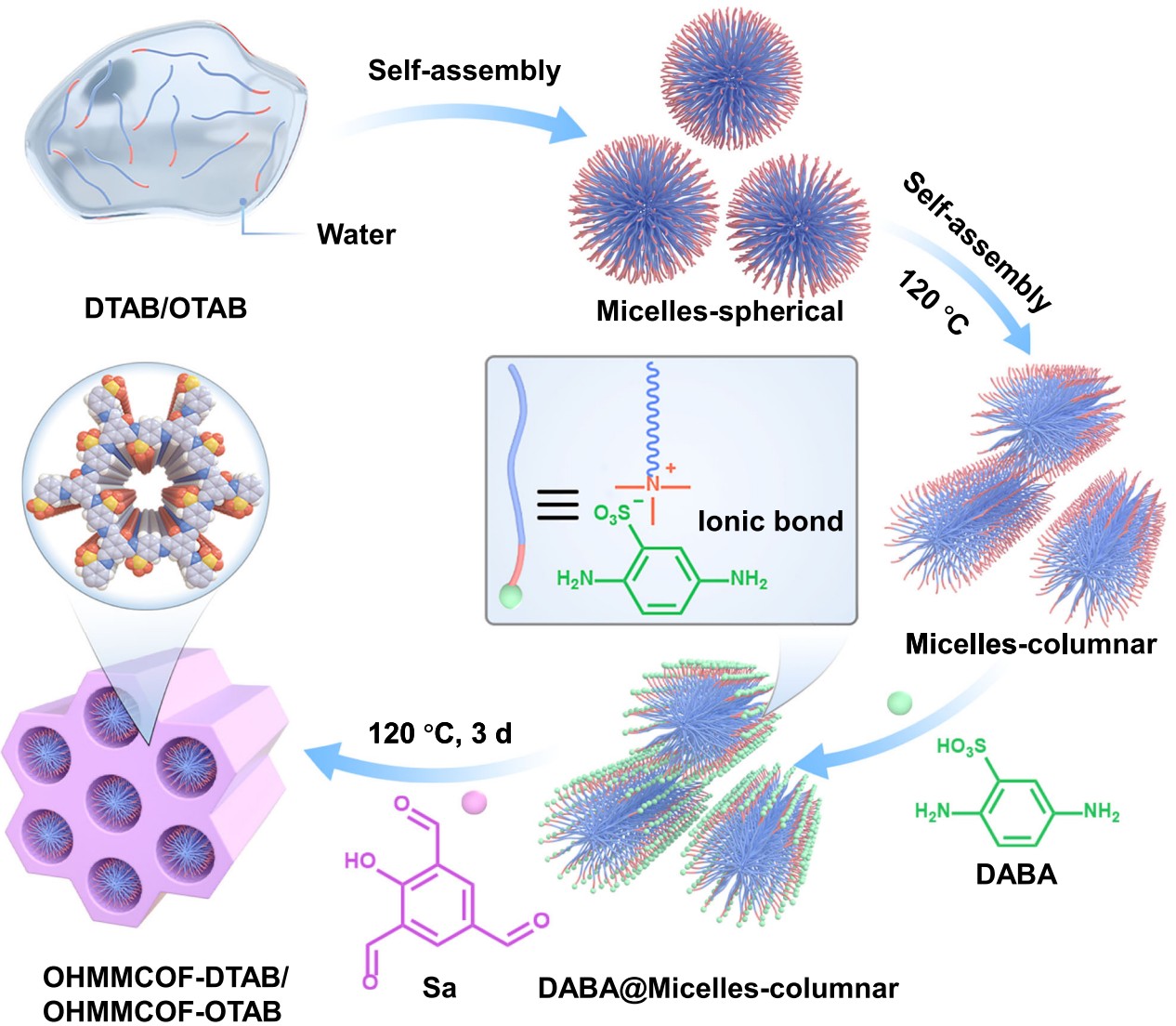

**Fig. 1 | Synthesis of OHMMCOF-DTAB/OHMMCOF-OTAB.** Illustration of the synthesis of off-template pre-products of ordered hierarchical microporous/mesoporous OHMMCOF-DTAB/OHMMCOF-OTAB.

reasonable residual factors of $R_{wp} = 1.37\%$ and $R_p = 1.78\%$. In comparison, the PXRD patterns of OHMMCOF-DTAB/OHMMCOF-OTAB as the template displayed additional sharp diffraction peaks at $2\theta = 2.5°$ and 2.0°, corresponding to d-spaces of 3.4 and 4.3 nm, respectively (Fig. 2B). The d-spaces coincide with the structural spacing formed by DTAB/OTAB in cylindrical micelles, revealing the orderly arrangement of the two templates in the synthesized OHMMCOF-DTAB/OHMM-COF-OTAB. The retention of the same Bragg diffraction peaks as MPCOF indicated that the introduction of templates did not affect the crystalline quality of COFs.

Fourier transform infrared spectroscopy (FT-IR) and $^{13}$C solid-state nuclear magnetic resonance spectroscopy ($^{13}$C SSNMR) were used to characterize the as-synthesized MPCOF and two other OHMMCOF-DTAB/OHMMCOF-OTAB (Fig. 2C, D). The appearance of C = N indicated the successful synthesis of three COFs (~1601 cm$^{-1}$ in FT-IR and ~156 ppm in $^{13}$C SSNMR), and the weak vibrations of C = C-N (-1290 cm$^{-1}$ in FT-IR and ~144 ppm in $^{13}$C SSNMR), -NH- (-3350 cm$^{-1}$ in FT-IR) and C = O (-1682 cm$^{-1}$ in FT-IR and ~197 ppm in $^{13}$C SSNMR) suggested the emergence of enol-ketone tautomeric effect in the structure[56,57]. In addition, the presence of methylene (~2920 cm$^{-1}$ and 2850 cm$^{-1}$ in FT-IR and ~31 ppm in $^{13}$C SSNMR) and S-OH (1020 cm$^{-1}$ in FT-IR and ~124 ppm in $^{13}$C SSNMR) in both OHMMCOF-DTAB/

OHMMCOF-OTAB indicated the successful introduction of the templates and partial reaction of sulfonate groups with the hydrophilic quaternary amine of the template[58–60]. On the other hand, the polar ends of cross-polarizable template molecules further demonstrated the presence of template molecules in micelles rather than "solution" species[32]. X-ray photoelectron spectroscopy (XPS) was also used to characterize the as-synthesized MPCOF and OHMMCOF-DTAB/OHMMCOF-OTAB. The survey XPS spectra showed that the three COFs prepared mainly contain peaks corresponding to C, N, O, and S (Supplementary Fig. 9). High-resolution spectra were used to further analyze the chemical structure and electronic state of COFs. The presence of C = N (398.5 eV, N 1s) indicated the successful synthesis of the three COFs, while the presence of C = C-N (399.9 eV, N 1s) and C = O (530.8 eV, O 1s) indicated the presence of enol-ketone tautomeric effect in the structures (Fig. 3A and Supplementary Fig. 10)[61,62]. In addition, the N 1s high-resolution spectra of OHMMCOF-DTAB/OHMMCOF-OTAB both exhibited additional peaks at 401.7 eV, attributed to the C-N$^+$ of the templates (Fig. 3A)[63]. The high-resolution spectra of S 2p (S 2p$_{3/2}$ = 168.3 eV and S 2p$_{1/2}$ = 167.1 eV) showed a negative shift of 0.2 eV due to the electron-withdrawing effect after the introduction of the templates, which further demonstrated the successful combination of the templates and COF backbone (Fig. 3B). The

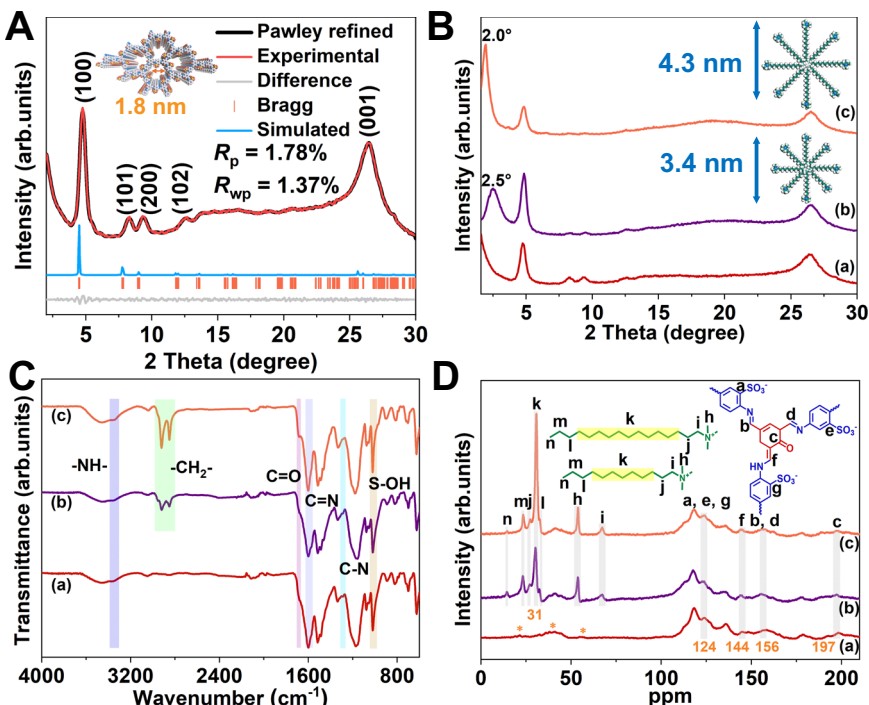

**Fig. 2 | Crystallinity and structural characterizations of MPCOF, OHMMCOF-DTAB, and OHMMCOF-OTAB. A** Simulated and experimental PXRD patterns of MPCOF, (**B**) PXRD patterns (inset: the simulated size of the templates), (**C**) FT-IR spectra, and (**D**) $^{13}$C SSNMR spectra (where the asterisks (*) represent the spinning sidebands). For pictures (**B**–**D**): (a) MPCOF, (b) OHMMCOF-DTAB, and (c) OHMMCOF-OTAB.

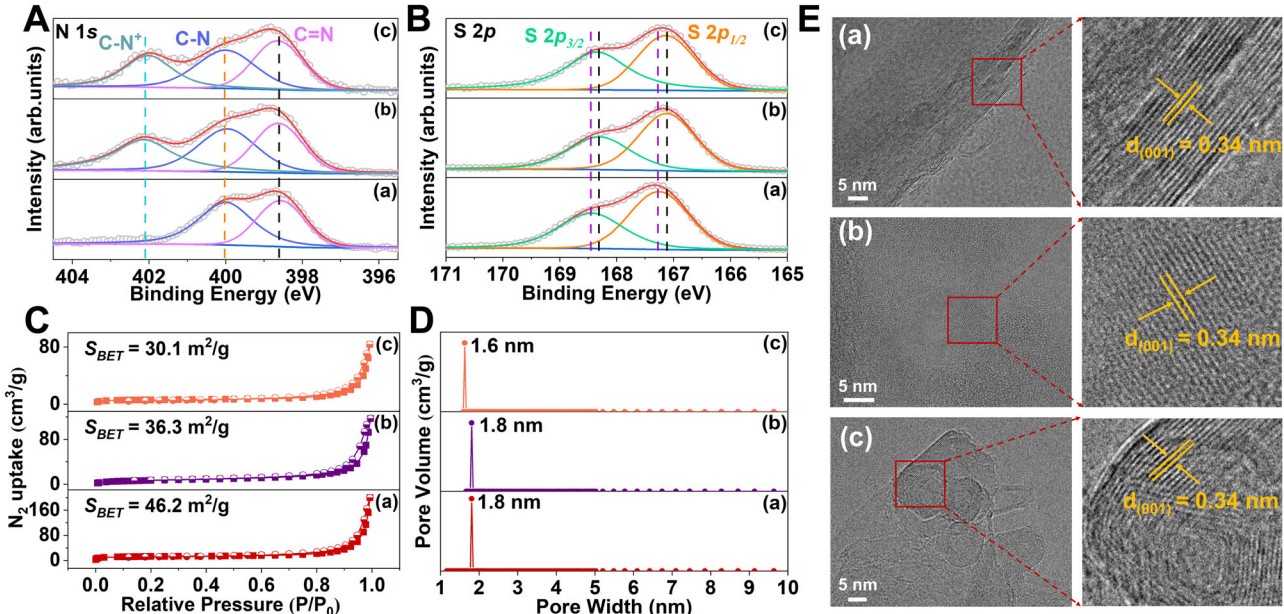

**Fig. 3 | Characterizations of the structure and porosity parameters of MPCOF, OHMMCOF-DTAB, and OHMMCOF-OTAB.** High-resolution XPS spectra of (**A**) N 1*s* and (**B**) S 2*p*, (**C**) nitrogen adsorption-desorption isotherms at 77 K and (**D**) the corresponding pore size distributions, and (**E**) the HRTEM images. For all pictures: (a) MPCOF, (b) OHMMCOF-DTAB, and (c) OHMMCOF-OTAB.

nitrogen adsorption and desorption isotherms at 77 K were used to evaluate the permanent porosity of the three COFs. As shown in Fig. 3C, the Brunauer-Emmet-Teller (BET) surface areas of the three COFs were relatively low, which could be attributed to the blockage of the internal pores of the COFs by sulfonate ions or long-chain alkyl quaternary amine ions in the pore (Supplementary Fig. 11 and Table 1). However, the pore size distribution derived from the non-local density functional theory method (NLDFT) demonstrated that the pore widths of the MPCOF and OHMMCOF-DTAB were centered at 1.8 nm, which is in good agreement with the theoretical value. The pore size of OHMMCOF-OTAB was reduced to 1.6 nm, which may be attributed to the extrusion of COF pores due to the presence of a large number of ordered arrangement templates in the structure (Fig. 3D). The elemental analysis (EA) showed that the C/N ratios of both OHMMCOF-

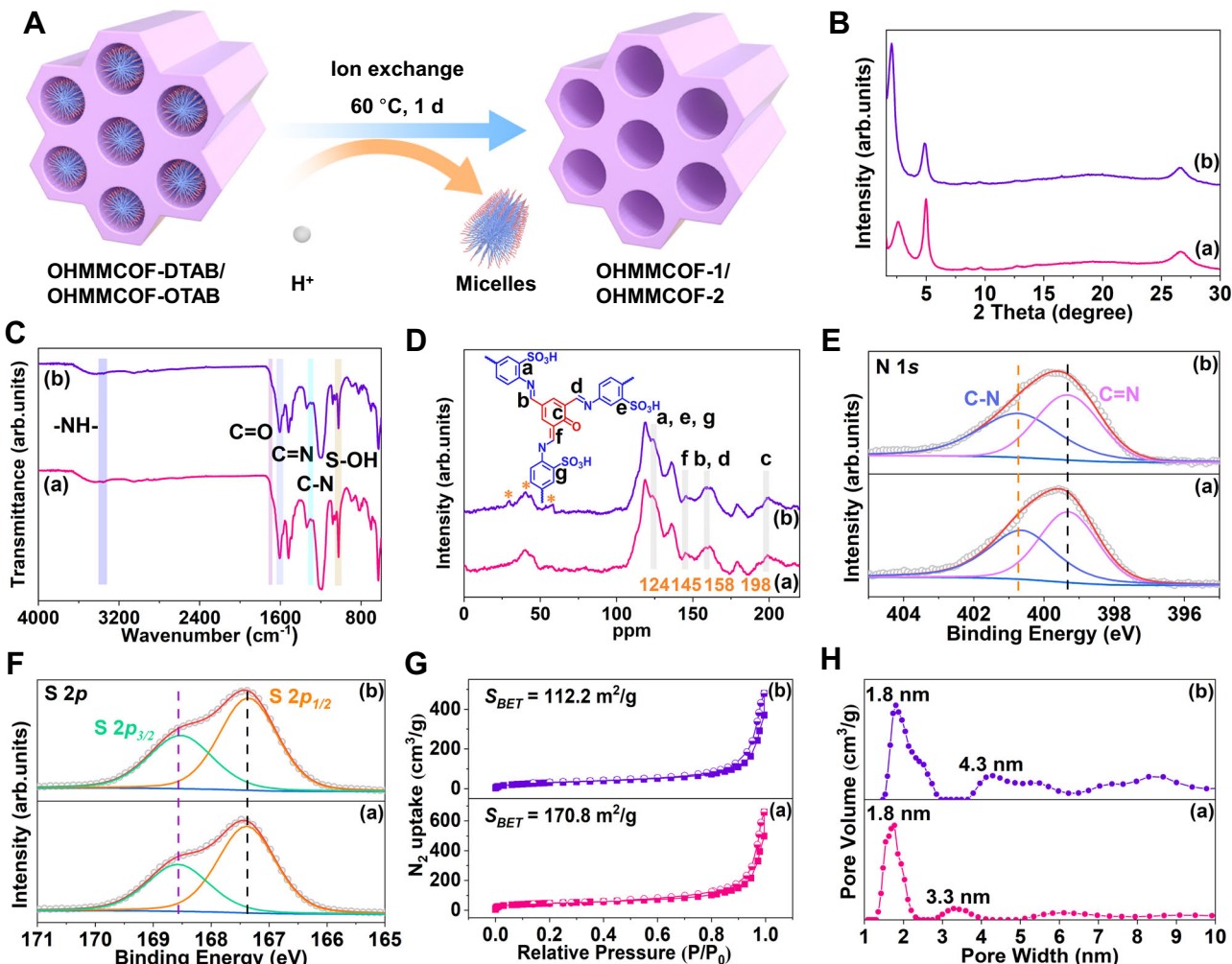

**Fig. 4 | Characterizations of crystallinity, structure, and porosity parameters of OHMMCOF−1 and OHMMCOF-2. A** Illustration of template removal using ion exchange method, (**B**) PXRD patterns, (**C**) FT-IR spectra, (**D**) $^{13}$C SSNMR spectra (where the asterisks (*) represent the spinning sidebands), high-resolution XPS spectra of (**E**) N 1$s$ and (**F**) S 2$p$, (**G**) nitrogen adsorption-desorption isotherms at 77 K and (**H**) the corresponding pore size distributions. For pictures (**B–H**): (a) OHMMCOF-1 and (b) OHMMCOF-2.

DTAB/OHMMCOF-OTAB (6.8 for OHMMCOF-DTAB and 7.2 for OHMMCOF-OTAB) were significantly increased compared with MPCOF (5.3) without templates, correlating with the increase of alkyl chain length of the introduced template (Supplementary Table 2). The hydrophobicity of long-chain alkyl micelles conferred a more hydrophobic surface to OHMMCOF-DTAB/OHMMCOF-OTAB compared to MPCOF (Supplementary Fig. 12). Thermogravimetric analysis (TGA) results showed that all three COFs are stable up to 270 °C, with the MPCOF materials exhibiting better stability than OHMMCOF-DTAB/OTAB. In addition, the TGA curves of both OHMMCOF-DTAB/OHMMCOF-OTAB showed an additional weightless plateau due to the long-chain alkyl disintegration at around 270−360 °C (Supplementary Fig. 13)[64].

Scanning electron microscopy (SEM) and TEM were used to characterize the microscopic morphology of the materials. The morphology of MPCOF and OHMMCOF-OTAB was similar nanotubular solid fiber structures, while OHMMCOF-DTAB displayed a tufted petal structure (Supplementary Figs. 14, 15). In addition, clear lattice fringes with a spacing of 0.34 nm could be observed in the high-resolution transmission electron microscopy (HRTEM) images of all three COFs, which were assigned to the (001) lattice planes (Fig. 3E), further demonstrating their high crystallinity. The TEM microregion energy dispersive X-ray spectroscopy (EDS) mappings demonstrated uniform elemental distributions within the COFs (Supplementary Fig. 16).

## Synthesis and characterization of OHMMCOF-1/OHMMCOF-2

After determining the successful introduction of ordered DTAB/OTAB templates in OHMMCOF-DTAB/OHMMCOF-OTAB, we conducted detailed template removal experiments. Maintaining the crystallinity of the material while removing the template has been a challenge in the past for hierarchical crystalline soft-template porous materials[42,51,65,66]. To address this issue, we established ionic bonds between the template and the COF material, allowing for template removal without breaking the covalent linkers using a simple ion exchange method. The resulting materials after template removal were named OHMMCOF-1/OHMMCOF-2 for OHMMCOF-DTAB/OHMMCOF-OTAB, respectively (Fig. 4A and Supplementary Section 14). Initially, we attempted to remove the DTAB template using a low-concentration hydrochloric acid aqueous solution, and after optimization, we successfully obtained ordered hierarchical microporous/mesoporous OHMMCOF-1 by stirring OHMMCOF-DTAB in a 0.5 M HCl aqueous solution at 60 °C for 1 day (Supplementary Fig. 17). However, this method was ineffective in removing the OTAB template, possibly due to the increased hydrophobicity of the material caused by longer alkyl chain micelles, which hindered the entry of the aqueous solution into the pores of OHMMCOF-OTAB. We then considered using an ethanol solution of hydrochloric acid to remove OTAB, and the results showed that OTAB could be removed with ethanol solution at a low concentration such as 0.5 M HCl, but the crystallinity of the entire material was also lost. This

may be because organic solvents disturb the internal order of COFs synthesized from the aqueous phase while removing the template, but aqueous solutions do not cause this result. Consequently, we decided to maintain the use of an aqueous solution of hydrochloric acid and increase the concentration to enhance the contact between H$^+$ and the template, enabling the successful removal of OTAB while preserving crystallinity. As expected, increasing the concentration of hydrochloric acid achieved the construction of ordered hierarchical microporous/mesoporous OHMMCOF-2 with OTAB as the template (Supplementary Fig. 18).

Characterizations of template removal and crystallinity retention were conducted using various methods. PXRD analysis confirmed the as-synthesized OHMMCOF-1/OHMMCOF-2 materials retain their crystallinity (Fig. 4B). FT-IR and $^{13}$C SSNMR were used to characterize the structural changes in OHMMCOF-1/OHMMCOF-2 after template removal (Fig. 4C, D). The C = N (~1606 cm$^{-1}$ in FT-IR and ~158 ppm in $^{13}$C SSNMR), C = O (-1683 cm$^{-1}$ in FT-IR and ~198 ppm in $^{13}$C SSNMR) and S-OH (1024 cm$^{-1}$ in FT-IR and ~124 ppm in $^{13}$C SSNMR) underwent a weak redshift after the removal of the templates, and methylene groups disappeared at the same time. In addition, the XPS high-resolution spectra also showed a faint redshift for C = N (399.3 eV, N 1$s$), C = O (530.9 eV, O 1$s$), and S 2$p$ (S 2$p_{3/2}$ = 168.6 eV and S 2$p_{1/2}$ = 167.4 eV), while C-N$^+$ disappeared at the same time (Fig. 4E, F and Supplementary Figs. 19, 20). Nitrogen adsorption-desorption isotherms at 77 K were used to characterize the pore parameters of OHMMCOF-1/OHMMCOF-2. As shown in Fig. 4G, both OHMMCOF-1/OHMMCOF-2 exhibited IV-type characteristic nitrogen adsorption isotherms as well as H3-type hysteresis loop, indicating the presence of mesopores in the materials[67]. The BET surface areas of OHMMCOF-1/OHMMCOF-2 after removal of DTAB and OTAB increased to 170.8 and 112.2 m$^2$/g compared to the OHMMCOF-DTAB/OHMMCOF-OTAB, respectively, which could be attributed to the removal of long-chain alkyl quaternary amine ions in the pores (Supplementary Fig. 21 and Table 3). The pore size distributions calculated by NLDFT revealed predominant pore sizes of 1.8 nm and additional mesoporous pore sizes of 3.3 nm for OHMMCOF-1, and 1.8 nm and 4.3 nm for OHMMCOF-2, which is very close to the theoretical values (Fig. 2B). Additionally, we observed that OHMMCOF-1/OHMMCOF-2 exhibited larger pore sizes in addition to the theoretical aperture constructed by cylindrical micelles, which may be due to some defects in the process of template removal (Fig. 4H). EA showed that the C/N ratios of the two OHMMCOF-1/OHMMCOF-2 were significantly reduced due to the removal of long-chain alkyl groups compared to OHMMCOF-DTAB/OHMMCOF-OTAB (Supplementary Table 4). Increased hydrophilicity and the absence of a weightless platform at around 270−360 °C in TGA further confirmed the removal of the templates (Supplementary Figs. 22, 23).

The morphological analysis of the OHMMCOF-1/OHMMCOF-2 after template removal revealed minimal changes in their structures, as observed in SEM and TEM images, and the TEM-EDS mappings showed that they still have a uniform distribution of elements (Supplementary Figs. 24−26). HRTEM images of both OHMMCOF-1/OHMMCOF-2 showed a lattice spacing of 1.8 nm corresponding to (100) lattice planes. Furthermore, since COFs are sensitive to high-energy electron beams, although XRD revealed an ordered arrangement of mesopores in OHMMCOF-1/OHMMCOF-2, only some disordered worm-like mesopores were shown in the HRTEM images (Supplementary Figs. 27, 28).

### Regulation of the extrinsic porosity of OHMMCOF-1/OHMMCOF-2

After determining that the strategy we designed could effectively construct ordered hierarchically microporous/mesoporous COFs, we studied the effect of template concentration and tried to construct mixed ordered hierarchically microporous/mesoporous COFs. We first changed the concentrations of DTAB/OTAB in the systems and selected 0.25, 0.5, 1.0, 2.0, and 3.0 eq for template concentration experiments. PXRD was performed on the synthesized materials, and the results were shown in Fig. 5A, B. For OHMMCOF-DTAB, the 2$\theta$ value of the corresponding self-assembly peak remained unchanged as the concentration of DTAB increased, but the intensity gradually increased compared with the (100) crystal plane of COFs, indicating an increase in the actual concentration of DTAB involved in the reaction (Supplementary Table 5). On the other hand, for the OHMMCOF-OTAB, the peak intensity ratios of the self-assembly peak to the (100) crystal plane of COFs increased with OTAB concentration up to 1.0 eq but did not change with further increase in OTAB concentration. These results showed that the template proportion in the final COFs increased with the increase of their concentration, implying that the mesoporous/microporous ratio can also be easily adjusted by this pathway, and this phenomenon was especially obvious when using the less hydrophobic DTAB as the template. The difference in solubility between DTAB and OTAB contributed to their distinct behavior in the reaction. The higher solubility of DTAB allowed for a continued increase in its actual reaction concentration, while the lower solubility of OTAB limited its participation in the reaction (Supplementary Fig. 29). OHMMCOF-DTAB-0.5 eq and OHMMCOF-DTAB-3.0 eq were selected for template removal experiments, followed by N$_2$ adsorption-desorption tests, and compared with OHMMCOF-DTAB-1.0 eq to verify the above conjectures. OHMMCOF-1-0.5 eq exhibited slightly lower specific surface area (159.0 m$^2$/g) and S$_{BET-meso}$/S$_{BET-micro}$ (5.5) than OHMMCOF-1-1.0 eq (170.8 m$^2$/g and 6.8), while OHMMCOF-1-3.0 eq exhibited higher specific surface area (178.1 m$^2$/g) and S$_{BET-meso}$/S$_{BET-micro}$ (7.5) (Fig. 5C, D and Supplementary Fig. 30, Tables 3, 6).

The change of template concentration affected the mesoporous/microporous ratio, and then we tried to add different proportions of DTAB/OTAB during the synthesis process, hoping to construct trimodal ordered hierarchically microporous/mesoporous COFs with two different mesopores. As shown in Fig. 5E, when OTAB accounted for more than 1/2 of the system, the 2$\theta$ value of the ordered structure in the XRD pattern is 2.0°, corresponding to a d-space of 4.3 nm. As the OTAB concentration gradually decreased in the system, both the 2$\theta$ value and the FWHM showed a controllable gradual decrease, indicating the size of the orderly arrangement micelles in the material structure is decreasing, and finally close to the size (3.3 nm) of the micelles formed by the DTAB template, which was also supported by the continuous enhancement of the relative strength of the (100) crystal plane peak from COFs. OTAB with longer alkyl chains has a stronger hydrophobicity during the formation of template micelles, which makes it dominant in mixed systems, so DTAB can only dominate when the proportion of OTAB gradually decreases. Although changing the ratio of DTAB/OTAB in the system could not synthesize ordered hierarchically microporous/mesoporous COFs with two different mesopores, it was unexpectedly found that the micelle size in the final COFs could be controlled, which means that this may be a new way to control the mesoporous size. Two materials with DTAB/OTAB of 14:1 and 1:2 were selected for the template removal experiment, and the resulting COFs were named OHMMCOF-M-14:1 and OHMMCOF-M-1:2 (while M represents the mixture), respectively, and then N$_2$ adsorption-desorption experiments were performed to verify the above conjecture (Fig. 5F and Supplementary Fig. 31, Table 7). The mesoporous pore size of OHMMCOF-M-14:1 was mainly 3.6 nm, which was relatively close to that of the mesopores constructed entirely with DTAB self-assembled micelles (3.3 nm), while the mesoporous pore size of OHMMCOF-M-1:2 was mainly 4.5 nm, which was also relatively close to the mesopores constructed entirely with OTAB self-assembled micelles (4.3 nm) (Fig. 5G).

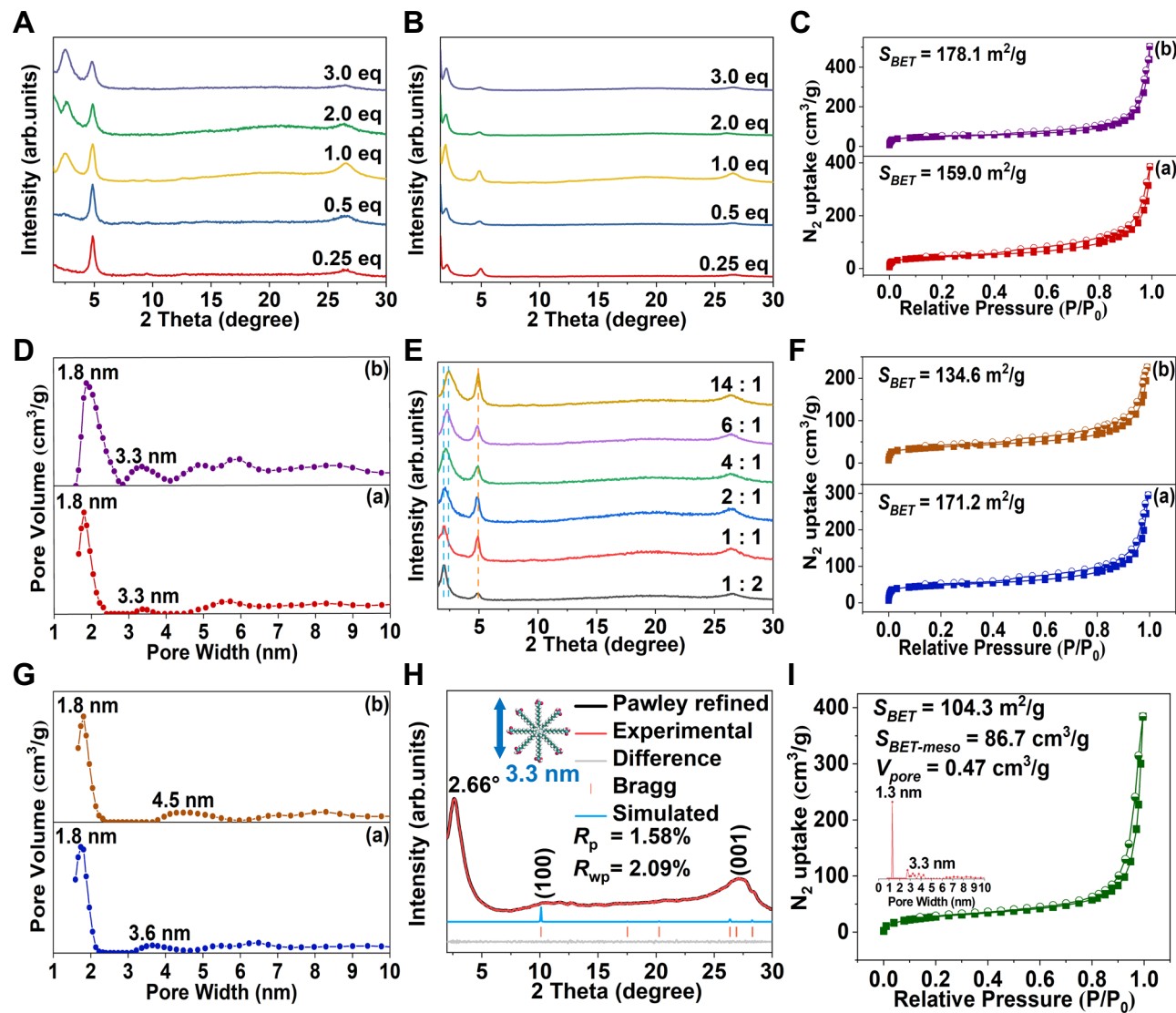

**Fig. 5 | Pore regulation and system expansion of OHMMCOFs.** PXRD patterns of (**A**) OHMMCOF-DTAB and (**B**) OHMMCOF-OTAB corresponding to different input concentration, (**C**) nitrogen adsorption-desorption isotherms at 77 K and (**D**) the corresponding pore size distributions, for (**C, D**): a) OHMMCOF-1-0.5 eq and b) OHMMCOF-1-3.0 eq, (**E**) PXRD patterns corresponding to templates with different concentration ratios of DTAB/OTAB, (**F**) nitrogen adsorption-desorption isotherms at 77 K and (**G**) the corresponding pore size distributions, for (**F, G**): a) OHMMCOF-M-14:1 and (b) OHMMCOF-M-1:2, (**H**) simulated and experimental PXRD patterns of OHMMCOF-3 (inset: the simulated size of the SLS micelles), and (**I**) nitrogen adsorption-desorption isotherms at 77 K of OHMMCOF-3 (inset: the corresponding pore size distributions).

## Validation of the scalability of the soft template strategy

To further validate our strategy, ordered hierarchically microporous/mesoporous cationic COFs were also constructed using sodium lauryl sulfonate (SLS) as the template, while triaminoguanidine chloride (TGCl) and 1,3,5-triformylphloroglucinol (TP) served as building blocks. The resulting COFs were named OHMMCOF-3. As shown in Fig. 5H, the PXRD pattern of OHMMCOF-3 showed three distinct Bragg diffraction peaks at 2.7, 10.3 and 27.2°, respectively. The latter two could correspond to the (100) and (001) crystal planes of COFs, respectively. MS simulation revealed that OHMMCOF-3 possessed an eclipsed AA stacking mode, and the optimized cell parameters were a = b = 10.13 Å, c = 3.38 Å and α = β = 90°, γ = 120° in a space group of P-6 after Pawley refinement, with reasonable final factors of $R_{wp}$ = 2.09% and $R_p$ = 1.58%. The additional former peak at 2.7° corresponded to a d-space of 3.3 nm, which was consistent with the theoretical diameter of the cylindrical micelles formed by SLS. Nitrogen adsorption and desorption at 77 K were also used to characterize the pore parameters. As shown in Fig. 5I and Supplementary Fig. 32, OHMMCOF-3 also

exhibited IV-type characteristic nitrogen adsorption isotherms as well as H3-type hysteresis loop, indicating the presence of mesopores in the material. The specific surface area was measured to be 104.3 m²/g, which was relatively low due to the large number of anions present in the framework. However, NLDFT calculation revealed the presence of micropores with a size of 1.3 nm, closer to the theoretical value. Additionally, the removal of cylindrical micelles constructed with SLS resulted in the formation of additional mesopores with a size of 3.3 nm[68].

## Study on the adsorption performance of U(VI)/Th(IV)

The extraction and separation of U(VI)/Th(IV) from rare earth or radioactive wastewater hold significant importance for rare earth hydrometallurgy, nuclear industry energy, and environmental protection[69–71]. Therefore, in this work, we used the solid-phase extraction method to compare and verify the ability of ordered hierarchical microporous/mesoporous OHMMCOF-1/OHMMCOF-2 compared with their parent microporous MPCOF in the adsorption

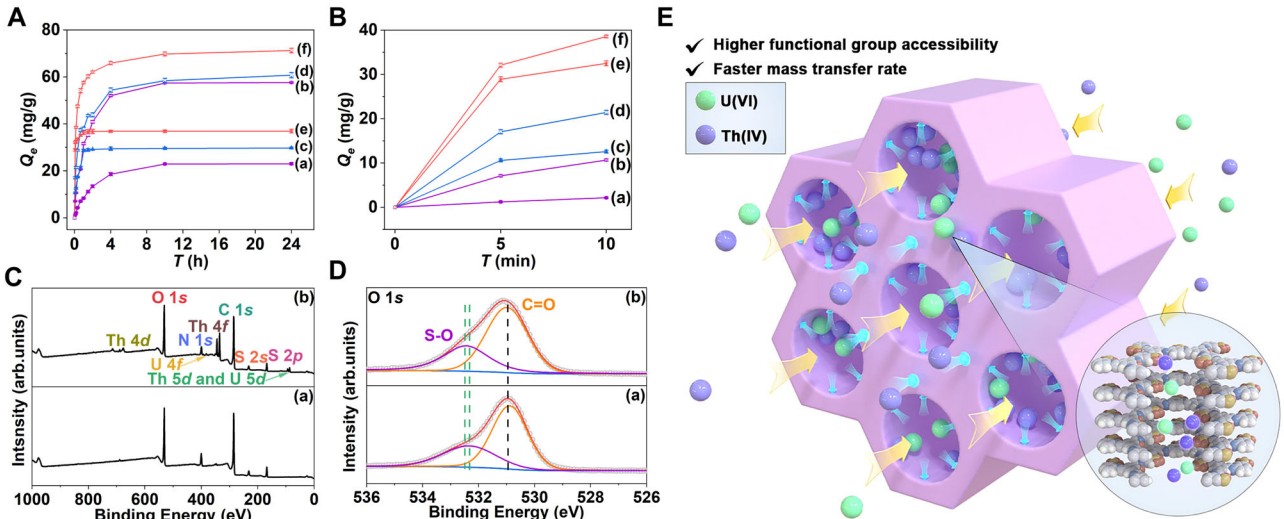

**Fig. 6 | U(VI)/Th(IV) adsorption experiments.** Adsorption kinetics at (**A**) 24 h and (**B**) 10 min, for pictures (**A**, **B**): (a) MPCOF-U, (b) MPCOF-Th, (c) OHMMCOF-1-U, (d) OHMMCOF-1-Th, (e) OHMMCOF-2-U, and (f) OHMMCOF-2-Th, (**C**) typical XPS survey spectra, (**D**) high-resolution XPS spectra of O 1$s$ for (a) OHMMCOF-2 and (b) OHMMCOF-2-U/Th, and (**E**) illustration of the process and mechanism of ordered hierarchical microporous/mesoporous OHMMCOFs with higher and faster adsorption of U(VI)/Th(IV). All the error bars represent the standard deviation of the experiments.

application of U(VI)/Th(IV) (Supplementary Section 2). Before application, we first tested the solvent and acid-base stability of the materials by stirring 10 mg of the target COF into 5 mL of different solutions for 24 h, and the results showed that both OHMMCOF-1/OHMMCOF-2 and their parent MPCOF had good stability, with only a slight loss of crystallinity under 1 M NaOH stirring condition, which ensured their potential applications under complex systems (Supplementary Fig. 33). After that, we performed 25 mg of activated material into 50 mL of U(VI)/Th(IV) mixed solution with a pH of about 3.0 for a time-dependent kinetic experiment. As shown in Fig. 6A, all three COFs had a higher adsorption capacity for Th(IV) than U(VI). The adsorption of U(VI)/Th(IV) by MPCOF reached saturation at around 10 h, and the adsorption kinetics followed the pseudo-second-order kinetic model, with correlation coefficient values ($R^2$) of 0.99319 and 0.99761, respectively. The saturated adsorption capacities of U(VI)/Th(IV) were calculated to be 24.6 and 59.7 mg/g, respectively, which is very close to the corresponding experimental equilibrium results of 23.0 and 57.6 mg/g, respectively (Supplementary Section 29, Figs. 34, 35, and Table 8). This kinetic model further proved that the adsorption of MPCOF was primarily governed by chemical adsorption through ion exchange[72]. Compared with the parent MPCOF, the two OHMMCOF-1/OHMMCOF-2 exhibited improved adsorption capacities and kinetics, especially in terms of kinetics. Both OHMMCOF-1/OHMMCOF-2 also followed the pseudo-second-order kinetic model with high correlation coefficient values (Supplementary Figs. 36–39 and Tables 9,10). The experimental equilibrium adsorption capacities of U(VI)/Th(IV) by OHMMCOF-1 were 29.7 and 60.8 mg/g, respectively, while for OHMMCOF-2, the adsorption capacities are 37.0 and 71.3 mg/g, respectively. The enhanced adsorption capacities could be attributed to the increased utilization of functional groups and the additional mesopores, which provide more opportunities for ions to meet the functional groups in the micropores. The saturation adsorption kinetic time of the two OHMMCOF-1/OHMMCOF-2 for U(VI) was accelerated to about 1 h, and there was almost no change in Th(IV), which may be due to the complex species distribution caused by hydrolysis of Th(IV) at pH = 3.0 (Supplementary Fig. 40)[73,74]. In more detail, the adsorption kinetics of the two OHMMCOF-1/OHMMCOF-2 for U(VI)/Th(IV) have been greatly improved according to the adsorption kinetic data in the first 5 min (Fig. 6B). The adsorption capacities of OHMMCOF-1 could reach 35.7% and 28.1% of the saturated adsorption capacity of U(VI)/

Th(IV) at 5 min, respectively, while the adsorption capacities of OHMMCOF-2 could reach 78.3% and 45.1% of the saturated adsorption capacity of U(VI)/Th(IV) at 5 min, which was greatly improved compared with 5.4% and 12.3% of the parent MPCOF. Their average adsorption rates within 5 minutes were calculated, revealing that OHMMCOF-1 has average adsorption rates of 2.1 and 3.4 mg/(g·min) for U(VI) and Th(IV), respectively. OHMMCOF-2 exhibited even higher average adsorption rates of 5.8 and 6.4 mg/(g·min) for U(VI) and Th(IV), respectively. Compared with the average adsorption rates of the parent MPCOF (0.3 mg/(g·min)-U(VI) and 1.4 mg/(g·min)-Th(IV)), their adsorption rates of U(VI) increased to 7.0 and 19.3 times, and the adsorption rates of Th(IV) increased to 2.4 and 4.6 times, respectively. At the same time, MPCOF-Organic was synthesized using conventional organic solvents and evaluated for crystallinity, porosity, and U(VI)/Th(IV) kinetic adsorption properties (Supplementary Section 31). The results showed that the porosity ($S_{BET}$ = 112.7 m²/g, $V_{pore}$ = 0.42 cm³/g) and the adsorption kinetics (0.7 mg/(g·min)-U(VI) and 1.7 mg/(g·min)-Th(IV) in the first 5 min) of MPCOF-Organic were slightly improved compared with MPCOF, but still much lower than those of OHMMCOF-1/OHMMCOF-2. This remarkable enhancement in kinetic performance could be attributed to the ordered hierarchical porous structure of the OHMMCOF-1/OHMMCOF-2 (Fig. 6E).

Furthermore, building upon our earlier hypothesis regarding the adjustability of the mesoporous/microporous ratio through variations in template concentration, we also delved deeper into this concept by elucidating its impact on the mass transfer capacity of U(VI)/Th(IV). We selected 0.5 and 3.0 eq of OHMMCOF-1 as the typical COF sorbents for U(VI)/Th(IV) time-dependent kinetics adsorption experiments. The adsorption results proved our conjecture that OHMMCOF-1 with 0.5 eq DTAB had a modest improvement in the adsorption capacity of U(VI)/Th(IV) (experimental equilibrium adsorption capacity: 29.4 mg/g-U(VI) and 59.2 mg/g-Th(IV)) compared with MPCOF, and the adsorption rate within 5 min could reach 1.3 and 2.7 mg/(g·min), respectively, corresponding to a 4.3 and 1.9 times increase. In contrast, OHMMCOF-1 with 3.0 eq DTAB had a significant improvement in the adsorption capacity of U(VI)/Th(IV) with experimental equilibrium adsorption capacities of 30.7 mg/g-U(VI) and 60.5 mg/g-Th(IV). The adsorption rate within 5 min could reach 4.5 and 4.9 mg/(g·min), respectively, increased to 15.0 and 3.5 times (Supplementary Figs. 42–46, and Tables 12, 13).

The adjustable microporous/mesoporous ratio will further facilitate the application of OHMMCOFs in controlled mass transport.

To gain a better understanding of the adsorption mechanism of U(VI)/Th(IV) by these materials, various characterization techniques were employed, including XPS, SEM, TEM, and EDS mapping. OHMMCOF-2 was selected as a representative material for these analyses. The material adsorbing U(VI)/Th(IV) showed additional double hump peaks attributable to U(VI) (U $4f_{5/2}$ = 392.5 eV and U $4f_{7/2}$ = 381.6 eV) and Th(IV) (Th $4f_{5/2}$ = 344.2 eV and Th $4f_{7/2}$ = 334.9 eV) on the XPS spectra, respectively (Fig. 6C and Supplementary Fig. 47c, d). In addition, the SEM and TEM images revealed minimal changes in the morphology, and the EDS mappings showed that U(VI)/Th(IV) are evenly distributed (Supplementary Fig. 48). The adsorption mechanism was further analyzed using XPS high-resolution spectroscopy. The high-resolution spectra of N 1 $s$ showed little change, while both S-O (532.4 eV) and S $2p$ (S $2p_{3/2}$ = 168.7 eV and S $2p_{1/2}$ = 167.5 eV) showed weak shifts to high binding energy direction, which were caused by the electrostatic attraction between the sulfonic acid groups and U(VI)/Th(IV) when ion exchange occurs (Fig. 6D and Supplementary Fig. 47a, b)[75].

After in-depth study of the pore structure regulation and U(VI)/Th(IV) mass transfer ability of OHMMCOFs, we further investigated the selective adsorption of OHMMCOF-2 on Th(IV)/U(VI) and Th(IV)/Ln at pH=3 (Supplementary Sections 35, 36). The high distribution coefficient ($K_d$) for Th ($4.6 \times 10^3$) and the high separation factor ($SF$) of Th for other metals ($SF_{Th/U}$ = 218 and $SF_{Th/Ln}$ > $3.2 \times 10^3$) indicate that OHMMCOF-2 has a unique selectivity for Th(IV) and is a potential material for Th(IV)/U(VI) and Th(IV)/Ln separation (Supplementary Fig. 49). OHMMCOF-2 also demonstrated excellent selective adsorption ability for Th(IV) when compared with other porous materials (Supplementary Table 14).

## Discussion

In summary, we successfully designed and synthesized ordered hierarchical microporous/mesoporous COFs (named OHMMCOF-1/OHMMCOF-2) using the soft-template method. In order to avoid the template being expelled from the COF growth domain due to the strong driving force of crystallization and facilitate subsequent template removal, the templates were combined with the backbone of COFs through ionic bonds, allowing for template removal without breaking the covalent linkers using a simple ion exchange method. Through various characterization techniques, we demonstrated the presence and organized arrangement of the templates in the precursor OHMMCOF-DTAB/OHMMCOF-OTAB, while maintaining the crystallinity of the COFs and orderly mesoporous arrangement through ion exchange and template removal. Due to the improvement of ordered hierarchical pores and functional group accessibility, the adsorption kinetics of OHMMCOF-2 for U(VI) and Th(IV) were greatly improved compared with the parent MPCOF (up to 19.3 and 4.6 times, respectively), with a slight improvement in adsorption capacity. In addition, by adjusting the template concentration, we successfully modulated the ratio of micropores and mesopores within the materials. Furthermore, we also realized the adjustment of mesoporous pore size by simultaneously mixing different proportions of templates in one system. Finally, we successfully constructed another cationic ordered hierarchical OHMMCOF-3 based on SLS templates using guanidine-based linkers, providing further evidence for the universality of soft-template strategy. Such ordered hierarchically microporous/mesoporous COFs have higher functional group accessibility and faster mass transfer capacity, making them highly promising for a wide range of applications.

## Methods
### Instruments
Powder X-ray diffraction (PXRD) data were collected on Bruker D2 PHASER X-ray diffractometer using Cu Kα radiation. Fourier transform infrared (FT-IR) spectra (KBr pellet) were measured on IR spectrometer (NEXUS 670) between the ranges of 4000 to 400 cm⁻¹. Dynamic light scattering (DLS) experiments were measured with a Malvern Panalytical Zetasizer Nano series instrument in a quartz cuvette, and the measurements were performed in water. ¹³C SSNMR experiments were carried out on a Bruker Avance III 400 MHz. X-ray photoelectron spectroscopy (XPS) data were collected using a Kratos ASAM800 spectrometer. Nitrogen sorption measurements were performed using a Micromeritics ASAP 2460 Version 3.01 surface area analyzer at 77 K. The pore-size distribution curves were obtained from the adsorption branches using the non-local density functional theory (NLDFT) method. Element analysis (EA) was performed on a CARLO ERBA 1106 analyzer. The water contact angle was measured on a German Lauda Scientific LSA100 analyzer, the test droplet volume was 2 μL and the tablet pressure was 10 MPa. Thermogravimetric analyses (TGA) were carried out on a Shimadzu DTG-60 (H) analyzer under N₂ atmosphere at a heating rate of 10 °C/min within a temperature range of r.t.–600 °C. Scanning electron microscopy (SEM) was carried out with a Zeiss Sigma 300 field-emission scanning electron microanalyzer. The samples were sputtered with Au before being tested. Transmission electron microscopy (TEM) images were recorded using JEM-F200 at an accelerating voltage of 200 kV.

### Chemicals
All reagents and solvents were commercially available and used as received unless otherwise stated. The 2-hydroxy-1,3,5-benzene-tricarbaldehyde (Sa) and 2,5-diaminobenzenesulfonic acid (DABA) were purchased from Shanghai Kylpharm Co. Ltd., China. Dodecyl trimethyl ammonium bromide (DTAB), octadecyl trimethyl ammonium bromide (OTAB), and sodium lauryl sulfonate (SLS) were purchased from Aladdin Chemistry Co. Ltd., China. $UO_2(NO_3)_2 \cdot 6H_2O$, $Th(NO_3)_4 \cdot 4H_2O$, $La(NO_3)_3 \cdot 6H_2O$, $Ce(NO_3)_3 \cdot 6H_2O$, $Nd(NO_3)_3 \cdot 6H_2O$, $Sm(NO_3)_3 \cdot 6H_2O$, $Gd(NO_3)_3 \cdot 6H_2O$, $Dy(NO_3)_3 \cdot 5H_2O$, $Ho(NO_3)_3 \cdot 5H_2O$, $Yb(NO_3)_3 \cdot 5H_2O$ and $Lu(NO_3)_3 \cdot 6H_2O$ were purchased from Hubei Chushengwei Chemical Co. Ltd., China.

### Synthesis of MPCOF
Sa (89.1 mg, 0.5 mmol), DABA (141.2 mg, 0.75 mmol), H₂O (5 mL), and 6 M acetic acid (0.5 mL) were placed into a 15 mL pressure-resistant tube and ultrasonicated for 5 min to evenly disperse. Subsequently, the reaction was shielded with nitrogen and the sealed tubes were placed in a heated oven at 120 °C for 3 days. After the reaction was completed, the samples were collected by the filter device and washed with DMF, methanol, and ethanol, respectively. Finally, the sample was vacuum-dried at 50 °C to obtain a black microporous parent MPCOF (156.4 mg, 77%).

### Synthesis of OHMMCOF-DTAB/OHMMCOF-OTAB
DTAB (231.3 mg, 0.75 mmol)/OTAB (294.4 mg, 0.75 mmol) was sonicated for 10 min into a 15 mL pressure-resistant tube containing H₂O (5 mL) to form micelles, followed by the addition of 2,5-diamino-benzenesulfonic acid (141.2 mg, 0.15 mmol) to continue sonicating for 5 min to fully react quaternary amine at the end of the micelles with the sulfonic acid group. Finally, 2-hydroxy-1,3,5-benzenetriacetaldehyde (89.1 mg, 0.5 mmol) and 6 M acetic acid (0.5 mL) were added and sonicated for 5 min to evenly disperse the mixture. The reaction was shielded with nitrogen and placed in a heated oven at 120 °C for 3 days. After the reaction was completed, the samples were collected by a filter device and washed with DMF, methanol, and ethanol, respectively. Finally, the samples were vacuum-dried at 50 °C to obtain black off-template pre-products OHMMCOF-DTAB/OHMMCOF-OTAB. The yield of OHMMCOF-DTAB was 73% (273.6 mg) and the actual concentration of DTAB involved in the reaction was calculated to be 0.38 mmol (51%), while the yield of OHMMCOF-OTAB was 75% (328.3 mg), and the actual concentration of OTAB involved in the

reaction was calculated to be 0.33 mmol (44%). When calculating, the monomers involved in the construction of the COF backbone in the default product are consistent with MPCOF.

## Synthesis of OHMMCOF-1/OHMMCOF-2

For OHMMCOF-DTAB, 500 mg of sample was put into a round bottom flask filled with 250 mL of 0.5 M HCl aqueous solution, then heated and stirred at 60 °C for 1 d to remove the template. After the reaction was completed, the samples were collected by a filter device and washed with water and methanol. Finally, the sample was vacuum-dried at 50 °C to obtain an ordered hierarchical microporous/mesoporous OHMMCOF-1 (271.0 mg). The calculated template removal rate was 107%, and the error may be due to partial damage to the sample by 0.5 M HCl aqueous solution.

For OHMMCOF-OTAB, 500 mg of the sample was put into a round bottom flask filled with 250 mL of 5 M HCl aqueous solution, then heated and stirred at 60 °C for 1 d to remove the template. After the reaction was completed, the samples were collected by a filter device and washed with water and methanol. Finally, the sample was vacuum-dried at 50 °C to obtain an ordered hierarchical microporous/mesoporous OHMMCOF-2 (202.1 mg). The calculated template removal rate was 129%, and the error may be due to partial damage to the sample by 5 M HCl aqueous solution.

## Synthesis of OHMMCOF-3

SLS (61.2 mg, 0.225 mmol) was sonicated for 10 min into a 15 mL pressure-resistant tube containing $H_2O$ (5 mL) to form micelles, followed by the addition of TGCl (31.5 mg, 0.225 mmol) to continue sonicating for 5 min to fully react the sulfonic acid group at the end of the micelle with the amino group. Finally, TP (47.4 mg, 0.225 mmol) and 6 M acetic acid (0.5 mL) were added and sonicated for 5 min to evenly disperse the mixture. Shield the reaction with nitrogen and place the sealed tube in a heated oven at 120 °C for 3 days. After the reaction was completed, the sample was collected by filtration and washed with DMF, methanol, and ethanol, respectively. Finally, the sample was vacuum-dried at 50 °C to obtain yellow off-template pre-products (80.4 mg, 70%). Subsequently, 80.4 mg of the sample was put into a round bottom flask filled with 100 mL of 0.5 M HCl aqueous solution, then heated and stirred at 60 °C for 1 d to remove the template. After the reaction was completed, the sample was collected by filtration and washed with water and methanol. Finally, the sample was vacuum-dried at 50 °C to obtain an ordered hierarchical microporous/mesoporous OHMMCOF-3 (48.9 mg).

## Data availability

All data supporting the findings of this work are available within this paper and its Supplementary Information. Other data are available from the corresponding author upon request.

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

## Acknowledgements

This work was financially supported by the National Natural Science Foundation of China (22125605 (L.M.), 22206137 (Y.L.), 22376150 (Y.L.) and U2067211 (N.L.)), the Postdoctoral Innovative Talent Support Program of China (BX20220218) (Y.L.), China Postdoctoral Science Foundation (2022M712230) (Y.L.). We are grateful to Shiyanjia Lab (www.shiyanjia.com) for the SEM, TEM and XPS analysis. We are also grateful to the Analytical & Testing Center, Sichuan University for the FT-IR, PXRD, and TGA analysis, especially Dr. Jing Zhou for her support of DLS testing. We thank Dr. Yue Qi from the Comprehensive Training Platform of the Specialized Laboratory in the College of Chemistry at Sichuan University for XRD and TGA testing. We also thank Dr. Feng Yang from the Comprehensive Training Platform of the Specialized Laboratory in the College of Chemistry at Sichuan University for TEM testing. The support from the Fundamental Research Funds for the Central Universities and the Comprehensive Training Platform Specialized Laboratory, College of Chemistry, Sichuan University, is gratefully acknowledged.

## Author contributions

L.M. and Y.L. were responsible for the overall design, direction, and supervision of the project. N.H. designed the material, performed the synthesis and characterizations, and completed the manuscript. Y. Zou drew the schematic diagram. C.C., M.T., Y. Zhang., X.Li., Z.J., J.Z., H.L., H.P., K.Y., B.J. helped with the PXRD, FT-IR, BET tests, and adsorption experiments. Z.H. helped with the TEM test. N. L participated in the discussion of the results of the study.

## Competing interests

The authors declare no competing interests.
