## [Peer Review File · Nature Communications]

REVIEWER COMMENTS

Reviewer #1 (Remarks to the Author):

In this work, Ma and coworkers reported the construction of a type of hierarchically porous organic materials. They have used water as the reaction medium to form the surfactant templates and facilitate their self-assembly, then link the soft templates to the COF backbone through ionic bonds between sulfonic sites and cationic ammonium sites. They attributed the formation of the porous structure to the strategy which avoids their crystallization incompatibilities thereby introducing the additional ordered arrangement of soft templates into the anionic microporous COFs. From my point of view, the report is interesting, which allows the formation of hierarchically porous structures of COFs. I recommend the manuscript be majorly revised before it can be recommended for publication, due to the following reasons.

- 1) The self-assembly mechanism is not clearly illustrated. The Scheme 1 provides some simple description of the self-assembly of the micelles, but it is confusing. It shows a sphere micelle. However, the final structure is tube like micelle. What is the relationship between the different kind of micelle structures? The current Scheme 1 is too simple and not sufficiently informative. Since the hierarchically structure is the main novelty of this work, I suggest the authors should clearly explain the formation mechanism and propose a new Scheme 1, which can better describe the process.
- 2) Another important issue is regarding the surface areas of these COFs. Although I really like this work, the highest surface area of the reported COFs is only 99 m²/g even after removal of the template, which is really not convincing to demonstrate them as a type of good crystalline materials. The BET surface areas of these COFs are too low to be called porous. Currently, the surface areas of the imine-linked COFs could reach very high levels. It is very easy for the imine-linked COFs to achieve high surface areas more than 1000 m²/g with the high crystalline structures. So, I think the surface area data of this work is too preliminary. Since this work highlights the hierarchically porous structures, I strongly suggest that it should be prominently improved in the revision.
- 3) In the solid-state ¹³C-NMR, some clear tiny peaks are not assigned. By the comparison of the ¹³-NMR before and after removal of the surfactant template, it seems that the templates are not sufficiently removed. Are the obvious peaks at around 50 ppm belong to the surfactant templates? If it is the remaining surfactant, how much content it was kept in the final materials?
- 4) It is suggested that the XRD patterns of the hierarchical COFs should be simulated to compare with the experimental data, so as to confirm the proposed structures.
- 5) The adsorption performance should be compared to the reported materials in details, to demonstrate the advances of the current adsorption materials.
- 6) Usually, the imine-linked COFs can be hardly stable in both acidic and basic conditions except some specific COFs structures. It is quite surprising that the COFs in this work could be stable in NaOH, HCl and even in HNO₃ and H₂SO₄.

7) The logic of the writing seems to be problematic. It may be better to establish the control strategy for the hierarchical structures entirely, before the discussion of the application of ion adsorption.

8) The format and the layout of this manuscript is rough, which should be carefully improved.

Reviewer #2 (Remarks to the Author):

COMMENTS TO AUTHOR

Recommendation: Reconsider after major revisions.

Comments: This manuscript from He and coworkers proposes a method to construct hierarchically porous COFs using amphoteric surfactants as soft templates, and investigate their adsorption properties for U(VI) and Th(IV). The authors conducted detailed experiments to explore the effect of the length, concentration, and ratio of the template on the hierarchical pore system, and illustrated the effective contribution of the hierarchical pore strategy to COF mass transfer through ion adsorption experiments. I like the approach of the authors and the design of the system. The enhancement of ion adsorption performance by hierarchically porous COFs is also impressive, and the overall experimental design and methodology are quite good. Although the observed performance is impressive, there are major concerns about the in-depth characterization of the hierarchically porous structure and the lack of research and discussion on the mechanism of the “water-triggered soft-template strategy”. All of the authors’ conclusions and insights into the system are based on the hierarchically porous structure, and uncertainties in these aspects undermine the whole study.

Overall, my recommendation is therefore to reconsider this work after major revisions.

Specific points:

1. What does “water-mediated” mean? What role does water play in addition to being a solvent for synthesizing dispersed surfactant templates? Additional rational discussion is necessary.
2. How did the sphere-like micelles allow the COF to evolve into through-circular mesoporous channels (Scheme 1) instead of a hollow sphere shape?
3. The authors simulated the dimensions of different micelles by theoretical calculations. Can the authors provide the actual sizes of the micelles determined by physical characterization such as dynamic light scattering (DLS) methods? Does the morphology of micelles change with the addition of acetic acid and the increase in temperature?
4. HR-TEM exhibits the lattice fringes of the (001) and (100) crystal planes attributed to OHMMCOF-DTAB/-OTAB and OHMMCOF-1/-2, respectively. Yet why are the newly formed ordered mesopores not observed? In fact, the diffraction peak signals corresponding to the ordered mesopores near $2.0-2.5^\circ$ are stronger than both the 100 and 001 peaks of the COF (Figures 1B and 3B), suggesting that these mesopores are more ordered and thus should be preferentially observed. These results appear to be

contradictory. In addition, does the PXRD diffraction signal corresponding to ordered mesopores shift when the template is removed?

5. All of the COFs in this manuscript exhibit low specific surface areas (less than 100 m²/g), and in particular the specific surface area of TpTGCl COF (26.9 m²/g) is much lower than those synthesized in the literature (200~300 m²/g) using conventional solvothermal methods (10.1021/jacs.5b13533; 10.1021/jacs.7b12292). These results make this approach uncompetitive. It is therefore unrepresentative to use only MPCOF (Section S2) as a comparison sample. This would probably better reflect the advanced nature of this strategy if MPCOF was constructed as a comparison sample using the currently commonly used solvothermal method.

6. The authors claim that “this work provides a brand new path for constructing ordered and tunable extrinsic porosity in COFs”. However, the manuscript only discusses one anionic COF and one cationic COF; is this strategy also valid for neutral COFs, which are currently the most widely used?

Minor Points:

1. The TGA in Sections S9 and S18 suggests incomplete sample activation. A weight loss of nearly 10% of the sample was observed before 100°C. In addition, the vertical scale of the TGA should be 0-100% instead of 0-1%.
2. Some more interpretation of the peaks at 25-50 ppm in the SSNMR (Figure 3D) would be helpful.
3. Can the authors add pore volume data for samples before and after template removal?
4. To demonstrate the superiority of the hierarchically porous COFs, it is necessary to compare the ion adsorption properties of OHMMCOF-1/-2 with those of porous materials reported in the literature.
5. Last but not least, some linguistic and writing errors need to be corrected and embellished. For example, the word “Anionic” on page 5, line 5, the sentence “showed a negative shift of -0.2 eV” on page 8, line 20, the labels “Figure S7” and “Figure S8” on page 22 (SI), and so on.

Reviewer #3 (Remarks to the Author):

In this work, the authors reported the synthesis and characterization of ordered hierarchical microporous/mesoporous COFs using the soft-template method. The ion exchange method is used to remove the templates while maintaining the high crystallinity of the COFs. Extrinsic porosity can be adjusted by changing template length, concentration, and ratio. Although the authors have carried out some experiments and collected significant data, they are common and ordinary, some of them are short of in-depth investigation. It lacks novelty as a research article for publication in Nature Communications.

1. The introduction section lacks a clear statement of the novelty and significance of the proposed soft-template strategy. Provide a more explicit explanation of how this approach differs from previous methods and its potential impact on COF research.

2. The authors stated that “there has been no published report on the utilization of the soft-template method to construct ordered hierarchical porous COFs”. However, numerous examples of soft template synthesis of COFs have been reported (J. Am. Chem. Soc. 2023, 145, 21974, Soft Matter, 2012, 8, 10801, Chem. Commun., 2022, 58, 9148). In this paper, the authors found that ordered microporous/mesoporous COFs can be synthesized based on the soft template method, which is only suitable for specific monomers and is not universal.
3. The authors should provide more detailed information about the synthesis of the COFs, including the specific reaction conditions, precursor materials, and purification methods. This will ensure reproducibility and facilitate further research in the field.
4. PXRD shows that the COFs synthesized by the soft template method show a small angle diffraction peak, which cannot indicate the existence of ordered microporous/mesoporous structure, and more experimental data should be provided to support it.
5. As the ordered crystal structure of COFs, microporous/mesoporous COFs can also be synthesized without soft template method. What are the advantages of introducing the soft template method to synthesize ordered hierarchical microporous/mesoporous COFs?
6. SEM and TEM characterizations of OHMMCOF and OHMMCOF do not show the ordered morphology, how to explain the ordered hierarchical structure of COFs synthesized by soft template method?
7. BET test of the COFs synthesized by soft template method showed relatively small specific surface area, is the template not removed clean? How to prove that the template has been removed clean? Does the residual template affect the XRD diffraction peak?
8. How to control the ordered hierarchical microporous/mesoporous of COF using soft template? As the main innovation point of this paper, this mechanism should be analyzed clearly.

Response to Reviewers

Reviewer: 1

In this work, Ma and coworkers reported the construction of a type of hierarchically porous organic materials. They have used water as the reaction medium to form the surfactant templates and facilitate their self-assembly, then link the soft templates to the COF backbone through ionic bonds between sulfonic sites and cationic ammonium sites. They attributed the formation of the porous structure to the strategy which avoids their crystallization incompatibilities thereby introducing the additional ordered arrangement of soft templates into the anionic microporous COFs. From my point of view, the report is interesting, which allows the formation of hierarchically porous structures of COFs. I recommend the manuscript be majorly revised before it can be recommended for publication, due to the following reasons.

Response: We are grateful to the reviewer for the helpful evaluation and comments as well as valuable suggestions. We have carefully revised the manuscript based on your professional and constructive comments and suggestions, please find our responses to your specific comments below.

1) The self-assembly mechanism is not clearly illustrated. The Scheme 1 provides some simple description of the self-assembly of the micelles, but it is confusing. It shows a sphere micelle. However, the final structure is tube like micelle. What is the relationship between the different kind of micelle structures? The current Scheme 1 is too simple and not sufficiently informative. Since the hierarchically structure is the main novelty of this work, I suggest the authors should clearly explain the formation mechanism and propose a new Scheme 1, which can better describe the process.

Response: We appreciate the reviewers' suggestions and apologize for the confusion caused by the unclear articulation of the self-assembly mechanism. To be honest, when drawing Scheme 1, we drew inspiration from the visual representation of mesoporous MOFs, mesoporous silica, mesoporous carbon, and other materials synthesized using

the soft template method, inadvertently overlooking the aspect of micellar shape changes (ACC. Chem. Res. 2022, 55, 2235-2247; ACS Nano 2022, 16, 13573-13594; Nat. Commun. 2023, 14, 8148). As you have mentioned, there is confusion between the spherical micelles and the tubular micelles in Scheme 1 of our original manuscript. To address this issue, we conducted an extensive literature review and then used TEM and DLS to offer a more detailed and intuitive characterization of micelles.

Professors Charles Tanford and Ninham et al. have extensively defined the relationship between the morphology of micelles formed by amphiphilic molecules and packing parameters (Tanford, C. Wiley-Interscience, New York, 1973; J. Chem. Soc. Faraday Transactions 2, 1976, 72, 1525-1568). The packing parameter (P) is defined as $P = v/A_{min}l_c$, where v represents the volume of the hydrophobic tail of surfactants, A_{min} is the optimal area of the hydrophilic head base, and l_c is the length of the hydrophobic chain tail. According to this definition, when $P < 1/3$, the micelles tend to assemble into a spherical shape, when $1/3 < P < 1/2$, the micelles tend to assemble into columnar, and when $P > 1/2$, the micelles tend to assemble into vesicles, as shown in Fig. R1-1. This definition has also been confirmed and cited in many subsequent literature (Langmuir 2001, 18, 31-38; Langmuir 2005, 21, 9426-9431; Macromol. Rapid. Comm. 2009, 30, 267-277; J. Mol. Liq. 2016, 222, 906-914, etc.). Therefore, the morphology of micelles in most of the literatures is defined by the packing parameter P . However, discrepancies arise when attempting to categorize micelle morphology solely based on the packing parameter P , as evidenced by studies such as J. Phy. Chem. 2002, 92, 774-783 (P (DTAB, 25°C) = 0.335) and J. Dispersion. Sci. Technol. 2019, 41, 856-862 (P (OTAB, 40 °C) = 0.374). Despite these P values slightly exceeding 1/3, the literature describes them as spherical micelles, indicating the complexity of directly defining micellar morphology using this formula, so we think it is not easy to directly define their specific morphology according to the formula.

Fig. R1-1 Various self-assembled structures formed by amphiphilic block copolymers in a block-selective solvent. The type of structure formed is due to the inherent curvature of the molecule, which can be estimated through calculation of its dimensionless packing parameter. (Macromol. Rapid. Comm. 2009, 30, 267-277)

To gain further insights into the micellar state within our system, we first tried to directly observe the morphology of the micelles by TEM at room temperature, and the results are shown in Fig. R1-2. According to Fig. R1-2, we can see that both DTAB/OTAB exhibit spherical micelles and the Gaussian statistics can be used to obtain their average sizes of 3.29 and 4.45 nm, respectively, which is very close to the observed d-spacing calculated from the surfactant sizes and XRD patterns (3.4 and 4.3 nm, respectively). The presence of larger micelles in the images may be due to the agglomeration of micelles during TEM sample drying (Fig. R1-2a). Subsequently, to mimic our experimental conditions more closely, we also investigated whether the addition of acetic acid affected the morphology of the micelles. The results showed that the addition of acetic acid did not affect the micelle morphology of OTAB and DTAB.

Fig. R1-2 Micellar morphology observed by TEM at room temperature: a) DTAB and e) OTAB, and corresponding size distributions of spherical micelles: b) DTAB and f) OTAB, and micelle morphology with the addition of acetic acid: c) DTAB and g) OTAB, and corresponding spherical micelle size distributions: d) DTAB and h) OTAB.

At the same time, we performed DLS tests on DTAB/OTAB, and the results showed that the particle sizes were distributed at 3.62/4.85 nm at 25 °C, respectively (Fig. R1-3). Since the DLS measurements yield a hydrodynamic diameter, it is larger than the TEM observations, but it is also very close to the d-spacing calculated from surfactant sizes and XRD patterns (3.4 and 4.3 nm, respectively). In addition, many larger-sized micelles resulting from disordered agglomeration were also observed.

Fig. R1-3 a) DTAB and b) OTAB micelles size distributions by DLS test.

The above results show that DTAB/OTAB will self-assemble into spherical micelles as shown in our Scheme 1 at room temperature. To further approximate the synthesis conditions, we raised the temperature of the micelle assembly to 120 °C for 3 d in a

sealed tube, after which the assembly product was collected for TEM observation, and the results are shown in Fig. R1-4. When the temperature increases, both DTAB and OTAB exhibit a small number of spherical shapes accompanied by a large number of tube-like micelle assemblies. The transverse dimensions of these tube-like micelles are consistent with the DTAB/OTAB spherical micelles, approaching 3.4 and 4.3 nm, respectively, suggesting that they are derived from the further assembly of spherical micelles. Therefore, we conclude that the increase in temperature will lead to the further transformation of spherical micelles into tube-like micelles. Similar examples of block copolymer assembly to form cylindrical micelles have been reported in the previous literature (ACS Cent. Sci. 2022, 8, 1196-1208; Nat. Commun. 2022, 13, 2170; Nat. Commun. 2023, 14, 8148, Fig. R1-5).

Fig. R1-4 Micellar morphology observed by TEM after heating assembly in a 120 °C water bath for 3 d: a) DTAB and b) OTAB.

Fig. R1-5 The formation process of the ordered mesoporous nanofibers. a) Optical photographs of

the pure water (left) and F127/Resol monomicelle systems (right) under red laser illumination. b) Cryo-TEM image and c) corresponding structural model of the monomicelles. SEM images of the nanofibers prepared by the kinetically driven monomicelle oriented self-assembly approach at different reaction times: d) 3 h, e) 6 h, and f) 24 h. (Nat. Commun. 2023, 14, 8148)

Once again, we thank the reviewers for their suggestions, as this has helped us gain insight into the assembly process of micelles. Combined with our newly obtained experimental results, we have added TEM and DLS results to the supplementary information (Sections 5-6) and revised Scheme 1 to further elucidate the synthesis process of hierarchical porous COFs. We have made corresponding modifications to the revised manuscript and supplementary information. To enhance readability, we have also compiled the key revisions in a table for easy comparison. Please check out the details of the changes as follows.

List of modifications:

Location	Original	Revised
Manuscript Page 6 Lines 15-24	The crucial step in the synthesis involves dispersing DTAB/OTAB in an aqueous solution, allowing it to self-assemble into spherical micelles with amine bromide at the hydrophilic end. Subsequently, the addition of DABA leads to the formation of ammonium sulfonate as a micelle at the peripheral hydrophilic end through ion exchange (DABA@micelles). The amino groups then undergo reversible condensation with the subsequently added aldehydes, resulting in the formation of COFs (Scheme 1).	The crucial step in the synthesis involves dispersing DTAB/OTAB in an aqueous solution, allowing it to self-assemble into spherical micelles with amine bromide at the hydrophilic end (Micelles-spherical). Transmission electron microscopy (TEM) analysis revealed that the size distribution of the spherical micelles formed by the assembly of DTAB and OTAB is predominantly centered around 3.29 and 4.45 nm, while dynamic light scattering (DLS) measurements indicated sizes of 3.62 and 4.85 nm, respectively. As the temperature increased, the spherical micelles underwent further self-assembly to transition into columnar micelles (Micelles-columnar). Subsequently, the addition of DABA facilitated the formation of ammonium sulfonate as a

		micelle at the peripheral hydrophilic end through ion exchange, leading to the formation of DABA@micelles-columnar (Supplementary Figs. 3-6). The amino groups then underwent reversible condensation with the subsequently added aldehydes, resulting in the formation of COFs (Scheme 1).
Manuscript Page 7 Lines 9-12	The d-spaces coincide with the structural spacing formed by DTAB/OTAB in spherical micelles, revealing the orderly arrangement of the two templates in the synthesized OHMMCOF-DTAB/OHMMCOF-OTAB. The retention of the same Bragg diffraction peaks as MPCOF indicated that the introduction of templates did not affect the crystalline quality of COFs.	The d-spaces coincide with the structural spacing formed by DTAB/OTAB in cylindrical micelles, revealing the orderly arrangement of the two templates in the synthesized OHMMCOF-DTAB/OHMMCOF-OTAB. The retention of the same Bragg diffraction peaks as MPCOF indicated that the introduction of templates did not affect the crystalline quality of COFs.
Manuscript Page 7 Line 15 and Page 8 Lines 1-3	 Scheme 1. Illustration of the synthesis of off-template pre-products of ordered hierarchical microporous/mesoporous OHMMCOF-DTAB/OHMMCOF-OTAB.	 Scheme 1. Scheme of OHMMCOF-DTAB/OHMMCOF-OTAB synthesis. Illustration of the synthesis of off-template pre-products of ordered hierarchical microporous/mesoporous OHMMCOF-DTAB/OHMMCOF-OTAB.

List of additions:

Location	Revised
----------	---------

Supplementary Section 5. TEM Images of DTAB/OTAB Self-Assembling into Micelles

To gain further insights into the micelles state within our system, we first tried to directly observe the morphology of the micelles by TEM at room temperature, and the results are shown in **Supplementary Fig. 3**. According to **Supplementary Fig. 3**, we can see that both DTAB/OTAB exhibit spherical micelles and the Gaussian statistics can be used to obtain their average sizes of 3.29 and 4.45 nm, respectively, which is very close to the observed d-spacing calculated from the surfactant sizes and XRD patterns (3.4 and 4.3 nm, respectively). The larger micelles in the figure may be due to the agglomeration of micelles during TEM sample drying (**Supplementary Fig. 3a**). Subsequently, to our experimental conditions more closely, we also investigated whether the addition of acetic acid affected the morphology of the micelles (**Supplementary Fig. 4**). The results showed that the addition of acetic acid did not affect the micelle morphology of OTAB and DTAB.

Supplementary information
Pages S10-12

Supplementary Fig. 3 Micellar morphology observed by TEM at room temperature: a) DTAB and c) OTAB, and corresponding size distributions of spherical micelles: b) DTAB and d) OTAB.

Location	Revised
	Supplementary Fig. 4 Micellar morphology observed by TEM with the addition of acetic acid at room temperature: a) DTAB and c) OTAB, and corresponding size distributions of spherical micelles: b) DTAB and d) OTAB. To further approximate the synthesis conditions, we raised the temperature of the micelle assembly to 120 °C for 3 days in a sealed tube, after which the assembly product was collected for TEM observation, and the results are shown in Supplementary Fig. 5. When the temperature increases, both DTAB and OTAB exhibit a small number of spherical shapes accompanied by a large number of cylindrical micelle assemblies. The transverse dimensions of these cylindrical micelles are consistent with the DTAB/OTAB spherical micelles, approaching 3.4 and 4.3 nm, respectively, suggesting that they are derived from the further assembly of spherical micelles. Therefore, we conclude that the increase in temperature will lead to the further transformation of spherical micelles into cylindrical micelles. Similar examples of block copolymer assembly to form cylindrical micelles have been reported in the previous literatures.²⁻⁴  Supplementary Fig. 5 Micellar morphology observed by TEM after heating assembly in a 120 °C water bath for 3 d: a) DTAB and b) OTAB.
Supplementary information Page S13	Supplementary Section 6. DLS Data of DTAB/OTAB Self-Assembling into Micelles To further clarify the size of the micelles formed by the self-assembly of DTAB/OTAB, we also observed them using DLS testing, and the results showed that the particle sizes were distributed at 3.62/4.85 nm at 25 °C, respectively (Supplementary Fig. 6). The main reason for this difference in particle size is that the DLS test yields a hydraulic diameter, which is larger than the actual size. In addition, many larger-sized micelles resulting from disordered agglomeration were also observed. Finally, to investigate whether the increase in temperature will affect the morphology of micelles, we looked at whether the increase in temperature would affect the size of the micelles. The results showed that the addition of acetic acid did not affect the micelle morphology of OTAB and DTAB.

Location	Revised
	 Supplementary Fig. 6 a) DTAB and b) OTAB micelles size distributions by DLS at different temperatures.

2) Another important issue is regarding the surface areas of these COFs. Although I really like this work, the highest surface area of the reported COFs is only 99 m²/g even after removal of the template, which is really not convincing to demonstrate them as a type of good crystalline materials. The BET surface areas of these COFs are too low to be called porous. Currently, the surface areas of the imine-linked COFs could reach very high levels. It is very easy for the imine-linked COFs to achieve high surface areas more than 1000 m²/g with the high crystalline structures. So, I think the surface area data of this work is too preliminary. Since this work highlights the hierarchically porous structures, I strongly suggest that it should be prominently improved in the revision.

Response: Many thanks to the reviewers for the kind concerns and suggestions to improve the quality of our work. As you mentioned, the reported COFs in our work have a relatively low specific surface areas compared to the reported imine COFs, which may be mainly due to the following two reasons. Firstly, it may be that the widely free anions/cations in the pores, which is also the key factor for the generally low BET of other reported ions COFs (ACS Appl. Mater. Interfaces 2016, 8, 18505-18512: 69 m² g⁻¹ for NUS-10(R); Chem. Mater. 2016, 28, 1489-1494: 215 m² g⁻¹ for TpPa-SO₃H; J. Am. Chem. Soc. 2018, 140, 896-899: 220 m² g⁻¹ for CON-Cl; Chem. Commun. 2023, 59, 14435-14438: 53, 103 and 113 m² g⁻¹ for C4-IL/ICOF, C6-IL/ICOF and C10-IL/ICOF, respectively; J. Am. Chem. Soc. 2024, 146, 2313-2318: 57 m² g⁻¹ for TpPa-SO₃H). On the other hand, it may be that our activation process is not complete, resulting in the presence of water or some solvents in the pores. Sulfonic acid is highly

hydrophilic, which leads to some pores being occupied by water molecules, thus reducing the specific surface area. Additionally, according to the research results of Zhao Dan et al., the introduction of guest molecules may promote the interlayer sliding of 2D COFs (J. Am. Chem. Soc. 2023, 145, 1359-1366; J. Am. Chem. Soc. 2020, 142, 12995-13002), increasing the disorder of the system and further reducing the material's specific surface area. Therefore, we resynthesized the COFs and pretreated the material under vacuum conditions with an activation temperature of 150 °C, and then tested their specific surface area, the results of which are shown in Fig. R1-6. All activated COFs showed higher BET values than the original manuscript, further verifying the significance of hierarchical porous COFs. In addition, using the same activation method, we supplemented the experimental information on nitrogen adsorption-desorption of representative COFs that adjusted the mesoporous/microporous ratio and mesoporous size in the manuscript. Thanks again to the reviewers for the valuable suggestions, which helped us to further strengthen our work. We have made changes to the corresponding positions in the revised manuscript and supplementary information.

Fig. R1-6 The nitrogen adsorption-desorption isotherms at 77 K: a) in the original manuscript after activation at 120 °C for 24 h and b) in the revised manuscript after high-temperature activation at 150 °C for 24 h.

List of modifications:

Location	Original	Revised
Manuscript Page 11 Lines 1-6	 Figure 2. High-resolution XPS spectra of A) N 1s and B) S 2p, C) nitrogen adsorption-desorption isotherms at 77 K and D) the corresponding pore size distributions, and E) the HRTEM images. For all pictures: (a) MPCOF, (b) OHMMCOF-DTAB, and (c) OHMMCOF-OTAB.	 Fig. 2 Characterizations of the structure and porosity parameters of MPCOF, OHMMCOF-DTAB, and OHMMCOF-OTAB. High-resolution XPS spectra of A) N 1s and B) S 2p, C) nitrogen adsorption-desorption isotherms at 77 K and D) the corresponding pore size distributions, and E) the HRTEM images. For all pictures: (a) MPCOF, (b) OHMMCOF-DTAB, and (c) OHMMCOF-OTAB.
Manuscript Page 12 Line 27 and Page 13 Line 1	The BET surface areas of OHMMCOF-1/OHMMCOF-2 after removal of DTAB and OTAB increased to 99.6 and 37.9 m²/g compared to the OHMMCOF-DTAB/OHMMCOF-OTAB, respectively, which could be attributed to the removal of long-chain alkyl quaternary amine ions in the pores. The pore size distributions calculated by NLDFT revealed predominant pore sizes of 1.8 nm and additional mesoporous pore sizes of 3.3 nm for OHMMCOF-1, and 1.7 nm and 4.2 nm for OHMMCOF-2, which is very close to the theoretical values (Figure 1b).	The BET surface areas of OHMMCOF-1/OHMMCOF-2 after removal of DTAB and OTAB increased to 170.8 and 112.2 m²/g compared to the OHMMCOF-DTAB/OHMMCOF-OTAB, respectively, which could be attributed to the removal of long-chain alkyl quaternary amine ions in the pores. The pore size distributions calculated by NLDFT revealed predominant pore sizes of 1.8 nm and additional mesoporous pore sizes of 3.3 nm for OHMMCOF-1, and 1.8 nm and 4.3 nm for OHMMCOF-2, which is very close to the theoretical values (Fig. 1b).
Manuscript Page 13 Lines 11-17		
Location	Original	Revised
	Figure 3. A) Illustration of template removal using ion exchange method, B) PXRD patterns, C) FT-IR spectra, D) ¹³C SSNMR spectra, high-resolution XPS spectra of E) N 1s and F) S 2p, G) nitrogen adsorption-desorption isotherms at 77 K and H) the corresponding pore size distributions. For pictures B-H: (a) OHMMCOF-1 and (b) OHMMCOF-2.	Fig. 3 Characterizations of crystallinity, structure, and porosity parameters of OHMMCOF-1 and OHMMCOF-2. A) Illustration of template removal using ion exchange method, B) PXRD patterns, C) FT-IR spectra, D) ¹³C SSNMR spectra (where the asterisks (*) represent the spinning sidebands), high-resolution XPS spectra of E) N 1s and F) S 2p, G) nitrogen adsorption-desorption isotherms at 77 K and H) the corresponding pore size distributions. For pictures B-H: (a) OHMMCOF-1 and (b) OHMMCOF-2.
Manuscript Page 14 Lines 23-30 and Page 15 Lines 1-7	These results showed that the template proportion in the final COFs increased with the increase of their concentration, and this phenomenon was especially obvious when using the less hydrophobic DTAB as the template. The difference in solubility between DTAB and OTAB contributed to their distinct behavior in the reaction. The higher solubility of DTAB allowed for a continued increase in its actual reaction concentration, while the lower solubility of OTAB limited its participation in the reaction (Figure S33).	These results showed that the template proportion in the final COFs increased with the increase of their concentration, implying that the mesoporous/microporous ratio can also be easily adjusted by this pathway, and this phenomenon was especially obvious when using the less hydrophobic DTAB as the template. The difference in solubility between DTAB and OTAB contributed to their distinct behavior in the reaction. The higher solubility of DTAB allowed for a continued increase in its actual reaction concentration, while the lower solubility of OTAB limited its participation in the reaction (Supplementary Fig. 27). OHMMCOF-DTAB-0.5 eq and OHMMCOF-DTAB-3.0 eq were selected for template removal experiments, followed by N₂ adsorption-desorption tests, and compared with OHMMCOF-DTAB-1.0 eq to verify the above conjectures. OHMMCOF-1-0.5 eq exhibited slightly lower specific surface area (159.0 m²/g) and S_{BET-meso}/S_{BET-micro} (5.5) than

Location	Original	Revised
		OHMMCOF-1-1.0 eq (170.8 m ² /g and 6.8), while OHMMCOF-1-3.0 eq exhibited higher specific surface area (178.1 m ² /g) and S _{BET-meso} /S _{BET-micro} (7.5) (Fig. 4C-D and Supplementary Fig. 28, Table 3, 6).
Manuscript Page 15 Lines 24-30 and Page 16 Lines 1-4	Although changing the ratio of DTAB/OTAB in the system could not synthesize ordered hierarchically microporous/mesoporous COFs with two different mesopores, it was unexpectedly found that the micelle size in the final COFs could be controlled, which opens up new possibilities for more straightforward changes in mesopore size.	Although changing the ratio of DTAB/OTAB in the system could not synthesize ordered hierarchically microporous/mesoporous COFs with two different mesopores, it was unexpectedly found that the micelle size in the final COFs could be controlled, which means that this may be a new way to control the mesoporous size. Two materials with DTAB/OTAB of 14:1 and 1:2 was selected for the template removal experiment, and the resulting COFs were named OHMMCOF-M-14:1 and OHMMCOF-M-1:2 (while M represents the mixture), respectively, and then N ₂ adsorption-desorption experiments were performed to verify the above conjecture (Fig. 4F and Supplementary Fig. 29, Table 7). The mesoporous pore size of OHMMCOF-M-14:1 was mainly 3.6 nm, which was relatively close to that of the mesopores constructed entirely with DTAB self-assembled micelles (3.3 nm), while the mesoporous pore size of OHMMCOF-M-1:2 was mainly 4.5 nm, which was also relatively close to the mesopores constructed entirely with OTAB self-assembled micelles (4.3 nm) (Fig. 4G).
Manuscript Page 16 Lines 20, 23	As shown in Figure S24 , OHMMCOF-3 also exhibited IV-type characteristic nitrogen adsorption isotherms as well as H3-type hysteresis loop, indicating the presence of mesopores in the material.	As shown in Fig. 4I and Supplementary Fig. 30 , OHMMCOF-3 also exhibited IV-type characteristic nitrogen adsorption isotherms as well as H3-type hysteresis loop, indicating the presence of mesopores in the material. The

Location	Original	Revised
	The specific surface area was measured to be 26.9 m²/g, which was relatively low due to the large number of anions present in the framework. However, NLDFT calculation revealed the presence of micropores with a size of 1.3 nm, closer to the theoretical value. Additionally, the removal of spherical micelles constructed with SLS resulted in the formation of additional mesopores with a size of 3.3 nm.⁷¹	specific surface area was measured to be 104.3 m²/g, which was relatively low due to the large number of anions present in the framework. However, NLDFT calculation revealed the presence of micropores with a size of 1.3 nm, closer to the theoretical value. Additionally, the removal of spherical micelles constructed with SLS resulted in the formation of additional mesopores with a size of 3.3 nm.⁷¹
Manuscript Page 17 Lines 1-10	 Figure 5. PXRD patterns of A) OHMMCOF-DTAB and B) OHMMCOF-OTAB corresponding to different input concentration, adsorption kinetics at C) 24 h and D) 10 min, for pictures C and D: (a) OHMMCOF-1-0.5 eq-U, (b) OHMMCOF-1-0.5 eq-Th, (c) OHMMCOF-1-3.0 eq-U, and (d) OHMMCOF-1-3.0 eq-Th, E) PXRD patterns corresponding to templates with different concentration ratios of DTAB/OTAB, F) simulated and experimental PXRD patterns (inset: the simulated size of the SLS micelles) of OHMMCOF-3.	 Fig. 4 Pore regulation and system expansion of OHMMCOFs. PXRD patterns of A) OHMMCOF-DTAB and B) OHMMCOF-OTAB corresponding to different input concentration, C) nitrogen adsorption-desorption isotherms at 77 K and D) the corresponding pore size distributions, for C and D: a) OHMMCOF-1-0.5 eq and b) OHMMCOF-1-3.0 eq, E) PXRD patterns corresponding to templates with different concentration ratios of DTAB/OTAB, F) nitrogen adsorption-desorption isotherms at 77 K and G) the corresponding pore size distributions, for F and G: a) OHMMCOF-M-14:1 and b) OHMMCOF-M-1:2, H) simulated and experimental PXRD patterns of OHMMCOF-3 (inset: the simulated size of the SLS micelles), and I)

Location	Original	Revised
		nitrogen adsorption-desorption isotherms at 77 K of OHMMCOF-3 (inset: the corresponding pore size distributions).

List of additions:

Location	Revised															
Supplementary information Pages S38	Supplementary Section 26. BET Surface Area Plots and Pore Volume Parameters of OHMMCOF-1-0.5/3.0 eq Supplementary Fig. 28 BET surface area plots of a) OHMMCOF-1-0.5 eq and b) OHMMCOF-1-3.0 eq. Supplementary Table 6. Pore volume parameters of OHMMCOF-1-0.5/3.0 eq.    COF V_{total} (cm^3/g) $S_{BET-micro}$ (m^2/g) $S_{BET-meso}$ (m^2/g) $S_{BET-meso}/S_{BET-micro}$     OHMMCOF-1-0.5 eq 0.59 24.5 134.5 5.5   OHMMCOF-1-3.0 eq 0.78 21.0 157.1 7.5   	COF	V_{total} (cm^3/g)	$S_{BET-micro}$ (m^2/g)	$S_{BET-meso}$ (m^2/g)	$S_{BET-meso}/S_{BET-micro}$	OHMMCOF-1-0.5 eq	0.59	24.5	134.5	5.5	OHMMCOF-1-3.0 eq	0.78	21.0	157.1	7.5
	COF	V_{total} (cm^3/g)	$S_{BET-micro}$ (m^2/g)	$S_{BET-meso}$ (m^2/g)	$S_{BET-meso}/S_{BET-micro}$											
OHMMCOF-1-0.5 eq	0.59	24.5	134.5	5.5												
OHMMCOF-1-3.0 eq	0.78	21.0	157.1	7.5												
Supplementary information Page S39	Supplementary Section 27. BET Surface Area Plots and Pore Volume Parameters of OHMMCOF-M-14:1/1:2															

3) In the solid-state ^{13}C -NMR, some clear tiny peaks are not assigned. By the comparison of the ^{13}C -NMR before and after removal of the surfactant template, it seems that the templates are not sufficiently removed. Are the obvious peaks at around 50 ppm belong to the surfactant templates? If it is the remaining surfactant, how much content it was kept in the final materials?

Response: Thanks for the reviewer's question. Based on your concern, we compared the ^{13}C SSNMR spectra of the parent MPCOF with the OHMMCOF-DTAB/OTAB before template removal and OHMMCOF-1/2 after template removal and combined the MPCOF with OHMMCOF-1/2 into Fig. R1-7. The results show that the apparent

peaks of around 50 ppm are not part of the surfactant template, as these peaks are also present in MPCOF. They may be rotational sideband peaks produced by the sample when it rotates at high speeds, and the same unknown peaks are found in similar structures in other literature, as shown in Fig. R1-8. In addition, the FT-IR (Fig. 3C), XPS (Fig. 3E), TGA (Supplementary Fig. 21), and water contact angles (Supplementary Fig. 20) data in our manuscript also show the complete removal of the template, so we believe that the template should be completely removed. To fully illustrate this result, we have supplemented the annotations of these unknown peaks in the corresponding figures and explained them in the legends, as shown in Figs. 1 and 3 in the manuscript.

Fig. R1-7 ^{13}C SSNMR spectra of MPCOF, OHMMCOF-1, and OHMMCOF-2.

Fig. R1-8 (a) Probable chemical structures of TpPa-(SO₃H-Py). (b) N 1s X-ray photoelectron spectroscopy (XPS) of hybrid COF TpPa-(SO₃HPy). (c) S 2p XPS of hybrid COF TpPa-(SO₃H-Py). (d) FESEM micrographs of TpPa-SO₃H (top row) and TpPa-(SO₃H-Py) (bottom row), before

(left) and after (right) treatment with phytic acid. (e) Solid-state ^{13}C CP MAS of phytic acid-loaded COFs (phytic@COFs), where the asterisks (*) represent the spinning sidebands. (Chem. Mater., 2016, 28, 1489-1494)

List of modifications:

Location	Original	Revised
Manuscript Page 10 Lines 1-6	 Figure 1. A) Simulated and experimental PXRD patterns of MPCOF, B) PXRD patterns (inset: the simulated size of the templates), C) FT-IR spectra, and D) ^{13}C SSNMR spectra. For pictures B-D: (a) MPCOF, (b) OHMMCOF-DTAB, and (c) OHMMCOF-OTAB.	 Fig. 1 Crystallinity and structural characterizations of MPCOF, OHMMCOF-DTAB, and OHMMCOF-OTAB. A) Simulated and experimental PXRD patterns of MPCOF, B) PXRD patterns (inset: the simulated size of the templates), C) FT-IR spectra, and D) ^{13}C SSNMR spectra (where the asterisks (*) represent the spinning sidebands). For pictures B-D: (a) MPCOF, (b) OHMMCOF-DTAB, and (c) OHMMCOF-OTAB.
Manuscript Page 13 Lines 11-17	 Figure 3. A) Illustration of template removal using ion exchange method, B) PXRD patterns, C) FT-IR spectra, D) ^{13}C SSNMR spectra, high-resolution XPS spectra of E) N 1s and F) S 2p, G) nitrogen	 Fig. 3 Characterizations of crystallinity, structure, and porosity parameters of OHMMCOF-1 and OHMMCOF-2. A) Illustration of template removal using ion exchange method, B) PXRD patterns, C) FT-IR

Location	Original	Revised
	adsorption-desorption isotherms at 77 K and H) the corresponding pore size distributions. For pictures B-H: (a) OHMMCOF-1 and (b) OHMMCOF-2.	spectra, D) ¹³ C SSNMR spectra (where the asterisks (*) represent the spinning sidebands), high-resolution XPS spectra of E) N 1s and F) S 2p, G) nitrogen adsorption-desorption isotherms at 77 K and H) the corresponding pore size distributions. For pictures B-H: (a) OHMMCOF-1 and (b) OHMMCOF-2.

4) It is suggested that the XRD patterns of the hierarchical COFs should be simulated to compare with the experimental data, so as to confirm the proposed structures.

Response: We would like to thank the reviewers for their suggestions to improve the quality of our work. Since the mesopores of hierarchical porous COFs are derived from the removal of additional orderly micelles, the micelles are linked to the COF backbone as ionic bonds. Therefore, it is difficult to simulate the XRD of hierarchical COFs by Materials Studio. The location of the Bragg diffraction peak based on the micelle size reaction has been reported in some literature, such as Nat. Mater. 2003, 2, 801-805, Angew. Chem. Int. Ed. 2011, 50, 636-639, and Angew. Chem. Int. Ed. 2018, 57, 3439-3443. Essentially, the additional ordered mesopores can be viewed as periodic defects present in the perfect COF structures. We believe that simulating hierarchical COF structures from the perspective of defect creation may be achievable. To further prove the reliability of this strategy, we try to construct additionally ordered mesopores in the COF structure using the ‘defect construction method’, and the results are shown in Fig. R1-9. As shown in the figure, we can see that the additional ordered mesopores lead to a change in the topology, as well as changes in the a and b values of the newly constructed unit cells. The result is that a mesoporous size of 3.0 nm corresponds to a Bragg diffraction peak of about 2.3 °, which is very much in line with the description in our manuscript. It is worth mentioning that due to the non-uniform size and non-unique arrangement of the soft template micelles, the modeling presented in Fig. R1-9 represents just one of many potential structures and may deviate significantly from the

actual scenario.

Fig. R1-9 Simulated COFs structure and XRD patterns of the additional ordered mesopores constructed by the ‘defect construction method’.

5) The adsorption performance should be compared to the reported materials in details, to demonstrate the advances of the current adsorption materials.

Response: Thanks to the reviewer’s valuable suggestion. Combined with our subsequent application system, we first supplemented the Th(IV)/U(VI) and Th(IV)/Ln separation experiments, the specific calculation formulas and experiment results have been supplemented in the Supplementary information (Supplementary Sections 3, 36). Through these experiments, we obtained the distribution coefficient (K_d) and separation factor $SF_{(Th/U, Ln)}$ of OHMMCOF-2 for different ions (Supplementary Fig. 47). We then compared it with the reported porous materials (Supplementary Section 37) and found that OHMMCOF-2 exhibited good Th(IV) adsorption capacity, K_d , and $SF_{(Th/U, Ln)}$. It is important to note that due to the lack of specific adsorption rate data within the initial 5-minute timeframe in much of the literature, a comprehensive comparison could not be made in the table. For a detailed summary, please refer to Supplementary Table 14.

List of modifications:

Location	Original	Revised
Manuscript Page 20 Lines 21-29	However, NLDFT calculation revealed the presence of micropores with a size of 1.3 nm, closer to the theoretical value. Additionally, the removal of spherical micelles constructed with SLS resulted in the formation of additional mesopores with a size of 3.3 nm.⁷¹	The high-resolution spectra of N 1s showed little change, while both S-O (532.4 eV) and S 2p (S 2p_{3/2} = 168.7 eV and S 2p_{1/2} = 167.5 eV) showed weak shifts to high binding energy direction, which were caused by the electrostatic attraction between the sulfonic acid groups and U(VI)/Th(IV) when ion exchange occurs (Fig. 5D and Supplementary Fig. 45a-b).⁷⁵ After in-depth study of the pore structure regulation and U(VI)/Th(IV) mass transfer ability of OHMMCOFs, we further investigated the selective adsorption of OHMMCOF-2 on Th(IV)/U(VI) and Th(IV)/Ln at pH=3 (Supplementary Sections 36, 37). The high distribution coefficient (K_d) for Th (4.6×10^3) and the high separation factor (SF) of Th(IV) for other metals ($SF_{Th/U} = 218$ and $SF_{Th/Ln} > 3.2 \times 10^3$) indicate that OHMMCOF-2 has a unique selectivity for Th and is a potential material for Th(IV)/U(VI) and Th(IV)/Ln separation (Supplementary Fig. 47). OHMMCOF-2 also demonstrated excellent selective adsorption ability for Th(IV) when compared with other porous materials (Supplementary Table 14).
Supplementary information Page S7	Section 3. Adsorption Experiments of U(VI)/Th(IV) All adsorption experiments were performed using a thermostatic shaking chamber at 25 °C with a vibration frequency of 200 rpm. The experiments were repeated more than three times and then averaged to ensure the reproducibility of the experimental results. Metal ion concentrations in solutions before	Section 3. Adsorption Experiments All adsorption experiments were performed using a thermostatic shaking chamber at 25 °C with a vibration frequency of 200 rpm. The experiments were repeated more than three times and then averaged to ensure the reproducibility of the experimental results. Metal ion concentrations in solutions before and after sorption were determined using

Location	Original	Revised
	and after sorption were determined using an ICP-OES PE OPTIMA 8000DV spectrometer. The equilibrium adsorption capacity (Q_e, mg/g) of U(VI)/Th(IV) was calculated using the following equation: $Q_e = \frac{(C_0 - C_e)}{m} \times V(1)$ where C_0 (ppm) and C_e (ppm) are the initial and equilibrium concentrations of U(VI)/Th(IV), respectively; V (L) is the volume of the solution; and m (g) is the mass of the adsorbent.	an ICP-OES OPTIMA8000 spectrometer. The equilibrium adsorption capacity (Q_e, mg/g), distribution coefficient (K_d, mL/g), and separation factor (SF) were calculated using the following equations: $Q_e = \frac{(C_0 - C_e)}{m} \times V(1)$ $K_d = \frac{(C_0 - C_e)}{C_e} \times \frac{V}{m}(2)$ $SF_{Th/M} = \frac{K_{d,Th}}{K_{d,M}}(3)$ where C_0 (ppm) and C_e (ppm) are the initial and equilibrium concentrations, respectively; V (L) is the volume of the solution; and m (g) is the mass of the adsorbent.

List of additions:

Location	Revised												
Supplementary information Page S62	Supplementary Section 36. Selective Adsorption Experiment of OHMMCOF-2 Supplementary Fig. 47 a) Distribution coefficients (K_d) and b) separation factors (SF) for thorium and other metals on OHMMCOF-2.												
Supplementary information Page S63	Supplementary Section 37. Comparison of Th(IV) Selective Adsorption Properties of OHMMCOF-2 and Other Porous Materials Supplementary Table 14. Comparison of Th(IV) selective adsorption properties of OHMMCOF-2 and other porous materials*.    Porous materials Conditions Capacity (mg/g) K_d(Th) SF (Th/U) Ref.             	Porous materials	Conditions	Capacity (mg/g)	K_d (Th)	SF (Th/U)	Ref.						
Porous materials	Conditions	Capacity (mg/g)	K_d (Th)	SF (Th/U)	Ref.								

Location	Revised					
GO	pH = 3.0, $C_0 \approx 23$ ppm	74.2	N.A.	36.3	6	
Salen-containing MOF	pH = N.A., $C_0 = 87$ ppm	46.4	N.A.	N.A.	7	
PPAC	pH = 4.0, $C_0 \approx 10$ ppm	2.8	0.33	2.1	8	
WMC-O	pH = 1.7, $C_0 = 10$ ppm	8.4	1.3×10^5	~ 200	9	
M-DGA	pH = 3.0, $C_0 = 90$ ppm	55.6	N.A.	N.A.	10	
SBA-15-O-DMAP	pH = 1.0, $C_0 = 100$ ppm	37.5	2.5×10^3	50	11	
KMS-5	pH = 1.0, $C_0 = 50$ ppm	50.0	1.04×10^6	N.A.	12	
PEI-GO-AG	pH = 3.0, $C_0 = 50$ ppm	12.0	N.A.	N.A.	13	
HPSM-PGAH	pH = 3.0, $C_0 = 20$ ppm	52.4	2.0×10^4	140	14	
Py-TFImI-25	pH = 4.0, $C_0 = 25$ ppm	80	5.5×10^3	~ 200	15	
Py-TFIm-25	pH = 4.0, $C_0 = 25$ ppm	140	1.2×10^4	~ 400	15	
OHMMCOF-2	pH = 3.0, $C_0 = 50$ ppm	71.3	4.6×10^3	218	This work	
* Since many literatures do not give specific adsorption rate of the first 5 min, a comprehensive comparison could not be made in the table.						

6) Usually, the imine-linked COFs can be hardly stable in both acidic and basic conditions except some specific COFs structures. It is quite surprising that the COFs in this work could be stable in NaOH, HCl and even in HNO₃ and H₂SO₄.

Response: Thank the reviewers for their keen attention to this phenomenon, and we

are also amazed at the excellent stability of the COFs. This high stability may be attributed to the prevalent enol-ketone tautomeric effect within the structure of our materials. This phenomenon leads to the replacement of numerous C=N bonds with more stable C=C-N bonds, surpassing the stability offered by imine bonds (J. Am. Chem. Soc. 2012, 134, 19524–19527; J. Am. Chem. Soc. 2013, 135, 5328–5331; Nat. Commun. 2020, 13, 1020). The enol-ketone tautomeric effect was also described in our manuscript (The appearance of C=N indicated the successful synthesis of three COFs ($\sim 1601\text{ cm}^{-1}$ in FT-IR and $\sim 156\text{ ppm}$ in ^{13}C SSNMR), and the weak vibrations of C=C-N ($\sim 1290\text{ cm}^{-1}$ in FT-IR and $\sim 144\text{ ppm}$ in ^{13}C SSNMR), -NH- ($\sim 3350\text{ cm}^{-1}$ in FT-IR) and C=O ($\sim 1682\text{ cm}^{-1}$ in FT-IR and $\sim 197\text{ ppm}$ in ^{13}C SSNMR) suggested the emergence of enol-ketone tautomeric effect in the structure.).

To delve deeper into the stability boundaries of COFs, we conducted additional acid-base stability experiments on three materials utilized in the adsorption studies at elevated concentrations (Fig. R1-10 and Supplementary Fig. 15, 16). The results showed that the three materials were highly resistant to HCl. However, their resistance to HNO₃, H₂SO₄, and NaOH was relatively weak, and exposure to higher concentrations of these substances led to the partial breakdown of the hierarchical porous structure of the COFs.

Fig. R1-10 PXRD patterns of a) MPCOF, b) OHMMCOF-1, and c) OHMMCOF-2 upon 1-day treatment under higher acid-base concentrations.

7) The logic of the writing seems to be problematic. It may be better to establish the control strategy for the hierarchical structures entirely, before the discussion of the application of ion adsorption.

Response: Thank you very much for your suggestion. Based on your suggestion, we have made appropriate changes to the revised manuscript.

List of modifications:

Location	Original	Revised
Manuscript Page 14 Lines 10-11	After determining that the strategy we designed could effectively construct ordered hierarchically microporous/mesoporous COFs and enhance their sorption capacity, we studied the effect of template concentration and tried to construct mixed ordered hierarchically microporous/mesoporous COFs.	After determining that the strategy we designed could effectively construct ordered hierarchically microporous/mesoporous COFs, we studied the effect of template concentration and tried to construct mixed ordered hierarchically microporous/mesoporous COFs.
Manuscript Page 14 Lines 23-27	These results showed that the template proportion in the final COFs increased with the increase of their concentration, and this phenomenon was especially obvious when using the less hydrophobic DTAB as the template.	These results showed that the template proportion in the final COFs increased with the increase of their concentration, implying that the mesoporous/microporous ratio can also be easily adjusted by this pathway, and this phenomenon was especially obvious when using the less hydrophobic DTAB as the template.
Manuscript Page 15 Lines 8-11	The great advantage of ordered hierarchical microporous/mesoporous COFs in terms of transport capability stimulated our interest in constructing multi-hierarchical mesopores in one system, so we tried to add different proportions of DTAB/OTAB during the synthesis process, hoping to construct trimodal ordered hierarchically microporous/mesoporous COFs with two different mesopores.	The change of template concentration affected the mesoporous/microporous ratio, and then we tried to add different proportions of DTAB/OTAB during the synthesis process, hoping to construct trimodal ordered hierarchically microporous/mesoporous COFs with two different mesopores.
Manuscript Page 19 Lines 20-23	Motivated by these findings, we investigated whether the mesoporous/microporous ratio in the final COFs could be regulated by adjusting the concentration of the template, thereby changing the transport ability of the COFs. So, we selected 0.5 and 3.0 eq of OHMMCOF-1 as the typical COF sorbents for	Furthermore, building upon our earlier hypothesis regarding the adjustability of the mesoporous/microporous ratio through variations in template concentration, we also delved deeper into this concept by elucidating its impact on the mass transfer capacity of U(VI)/Th(IV). We selected 0.5 and 3.0 eq of OHMMCOF-1 as the typical COF

Location	Original	Revised
	U(VI)/Th(IV) time-dependent kinetics adsorption experiments.	sorbents for U(VI)/Th(IV) time-dependent kinetics adsorption experiments.
Manuscript Page 20 Lines 4-5	The adjustable microporous/mesoporous ratio will further facilitate the application of OHMMCOFs in controlled mass transport.	The adjustable microporous/mesoporous ratio will further facilitate the application of OHMMCOFs in controlled mass transport.

8) The format and the layout of this manuscript is rough, which should be carefully improved.

Response: Many thanks for your kind reminder! We have made detailed changes to the article in accordance with the formatting requirements of *Nature Communications*, including moving the Chemicals and Materials Synthesis section from the supplementary information to the main text, adding a Data availability and Author Contributions statement, and changing the figures, legends, and the format of the references, etc., which have been highlighted in YELLOW background in the revised manuscript.

Reviewer: 2

This manuscript from He and coworkers proposes a method to construct hierarchically porous COFs using amphoteric surfactants as soft templates, and investigate their adsorption properties for U(VI) and Th(IV). The authors conducted detailed experiments to explore the effect of the length, concentration, and ratio of the template on the hierarchical pore system, and illustrated the effective contribution of the hierarchical pore strategy to COF mass transfer through ion adsorption experiments. I like the approach of the authors and the design of the system. The enhancement of ion adsorption performance by hierarchically porous COFs is also impressive, and the overall experimental design and methodology are quite good. Although the observed performance is impressive, there are major concerns about the in-depth characterization of the hierarchically porous structure and the lack of research and discussion on the mechanism of the “water-triggered soft-template strategy”. All of the authors’ conclusions and insights into the system are based on the hierarchically porous structure, and uncertainties in these aspects undermine the whole study.

Overall, my recommendation is therefore to reconsider this work after major revisions.

Response: We are very grateful to the reviewers for their recognition of the methods proposed by our work and the design of the experiments, and we are also grateful for the valuable comments of the reviewers to improve the quality of our work. We have made detailed changes based on your suggestions, please check them out.

1. What does “water-mediated” mean? What role does water play in addition to being a solvent for synthesizing dispersed surfactant templates? Additional rational discussion is necessary.

Response: Thank you very much for your questions and suggestions. We propose a ‘water-mediated’ soft template strategy for two main reasons. On the one hand, as you can see, water provides the necessary conditions for the self-assembly of surfactants and facilitates the further molding of micelles, and on the other hand, the reason why

we use the term ‘water-mediated’ in the title is that organic solvents are usually used for the synthesis of conventional COFs, which are not green and environmentally friendly. In this work, we chose water to avoid the use of organic solvents, and at the same time synthesized very interesting hierarchical porous COFs, which is both environmentally friendly and meaningful.

Besides, to be candid, the original title we used was ‘Constructing Ordered and Tunable Extrinsic Porosity in Covalent Organic Frameworks *via* Soft-Template Strategy in Water’. Throughout the submission process, after multiple revisions and refinements, we thought that the term ‘water-mediated’ might be better. Hence, we ultimately opted to include this term in the title.

2. How did the sphere-like micelles allow the COF to evolve into through-circular mesoporous channels (Scheme 1) instead of a hollow sphere shape?

Response: We are very grateful for the questions raised by the reviewers’ keen instincts, and we apologize for the confusion caused by the imperfection of the previous work. In our previous version of the manuscript, we focused more on the structure and properties of the subsequent COFs and therefore ignored the specific assembly process of micelles. So, there’s a lot of confusion for you and both other reviewers. In this modification, we take an in-depth look at the micelle assembly process. We used TEM to observe actual morphology of the micelles. As shown in Fig. R2-1, we can see that both DTAB/OTAB exhibit spherical micelles at room temperature and the Gaussian statistics can be used to obtain their average sizes of 3.29 and 4.45 nm, respectively, which is very close to the d-spacing calculated from surfactant sizes and XRD patterns (3.4 and 4.3 nm, respectively). The larger micelles in the figure may be due to the agglomeration of micelles during TEM sample drying.

Fig. R2-1 Micellar morphology observed by TEM at room temperature: a) DTAB and c) OTAB, and corresponding size distributions of spherical micelles: b) DTAB and d) OTAB.

When the temperature increased to 120 °C for 3 d, both DTAB and OTAB exhibit a small number of spherical shapes accompanied by a large number of cylindrical assemblies. The transverse dimensions of these cylindrical assemblies are consistent with the DTAB/OTAB spherical micelles, approaching 3.4 and 4.3 nm, respectively, suggesting that they are derived from the further assembly of spherical micelles (Fig. R2-2). Therefore, we conclude that the increase in temperature will lead to the further transformation of spherical micelles into cylindrical micelles. Similar examples of block copolymer assembly to form cylindrical micelles have been reported in the previous literature (ACS Cent. Sci. 2022, 8, 1196-1208; Nat. Commun. 2022, 13, 2170; Nat. Commun. 2023, 14, 8148, Fig. R2-3). Therefore, the formation of through-circular mesoporous channels is due to the removal of cylindrical micelles, which also means that no hollow sphere shape channels are formed.

Fig. R2-2 Micellar morphology observed by TEM after heating assembly in a 120 °C water bath for 3 d: a) DTAB and b) OTAB.

Fig. R2-3 The formation process of the ordered mesoporous nanofibers. a) Optical photographs of the pure water (left) and F127/Resol monomicelle systems (right) under red laser illumination. b) Cryo-TEM image and c) corresponding structural model of the monomicelles. SEM images of the nanofibers prepared by the kinetically driven monomicelle oriented self-assembly approach at different reaction times: d) 3 h, e) 6 h, and f) 24 h. (Nat. Commun. 2023, 14, 8148)

Combined with our newly obtained experimental results, we have added TEM results to the supplementary information (Supplementary Section 5) and changed Scheme 1 to further elucidate the synthesis process of hierarchical porous COFs. Please check out the details of the changes as follows.

List of modifications:

Location	Original	Revised
Manuscript Page 6 Lines 15-24	The crucial step in the synthesis involves dispersing DTAB/OTAB in an aqueous solution, allowing it to self-assemble into spherical micelles with amine bromide at the hydrophilic end. Subsequently, the addition of DABA leads to the formation of ammonium sulfonate as a micelle at the peripheral hydrophilic end through ion exchange (DABA@micelles). The amino groups then undergo reversible condensation with the subsequently added aldehydes, resulting in the formation of COFs (Scheme 1).	The crucial step in the synthesis involves dispersing DTAB/OTAB in an aqueous solution, allowing it to self-assemble into spherical micelles with amine bromide at the hydrophilic end (Micelles-spherical). Transmission electron microscopy (TEM) analysis revealed that the size distribution of the spherical micelles formed by the assembly of DTAB and OTAB is predominantly centered around 3.29 and 4.45 nm, while dynamic light scattering (DLS) measurements indicated sizes of 3.62 and 4.85 nm, respectively. As the temperature increased, the spherical micelles underwent further self-assembly to transition into columnar micelles (Micelles-columnar). Subsequently, the addition of DABA facilitated the formation of ammonium sulfonate as a micelle at the peripheral hydrophilic end through ion exchange, leading to the formation of DABA@micelles-columnar (Supplementary Figs. 3-6). The amino groups then undergo reversible condensation with the subsequently added aldehydes, resulting in the formation of COFs (Scheme 1).
Manuscript Page 7 Lines 9-12	The d-spaces coincide with the structural spacing formed by DTAB/OTAB in spherical micelles, revealing the orderly arrangement of the two templates in the synthesized OHMMCOF-DTAB/OHMMCOF-OTAB. The retention of the same Bragg diffraction peaks as MPCOF indicated that the introduction of templates did not affect the crystalline quality of COFs.	The d-spaces coincide with the structural spacing formed by DTAB/OTAB in cylindrical micelles, revealing the orderly arrangement of the two templates in the synthesized OHMMCOF-DTAB/OHMMCOF-OTAB. The retention of the same Bragg diffraction peaks as MPCOF indicated that the introduction of templates did not affect the crystalline quality of COFs.

Manuscript Page 7 Line 14 and Page 8 Lines 1-3	 Scheme 1. Illustration of the synthesis of off-template pre-products of ordered hierarchical microporous/mesoporous OHMMCOF-DTAB/OHMMCOF-OTAB.	 Scheme 1. Scheme of OHMMCOF-DTAB/OHMMCOF-OTAB synthesis. Illustration of the synthesis of off-template pre-products of ordered hierarchical microporous/mesoporous OHMMCOF-DTAB/OHMMCOF-OTAB.
---	--	--

List of additions:

Location	Revised
Supplementary information Pages S10-12	Supplementary Section 5. TEM Images of DTAB/OTAB Self-Assembling into Micelles To gain further insights into the micelles state within our system, we first tried to directly observe the morphology of the micelles by TEM at room temperature, and the results are shown in Supplementary Fig. 3. According to Supplementary Fig. 3, we can see that both DTAB/OTAB exhibit spherical micelles and the Gaussian statistics can be used to obtain their average sizes of 3.29 and 4.45 nm, respectively, which is very close to the observed d-spacing calculated from the surfactant sizes and XRD patterns (3.4 and 4.3 nm, respectively). The larger micelles in the figure may be due to the agglomeration of micelles during TEM sample drying (Supplementary Fig. 3a). Subsequently, to our experimental conditions more closely, we also investigated whether the addition of acetic acid affected the morphology of the micelles (Supplementary Fig. 4). The results showed that the addition of acetic acid did not affect the micelle morphology of OTAB and DTAB.

Location	Revised
----------	---------

Supplementary Fig. 3 Micellar morphology observed by TEM at room temperature: a) DTAB and c) OTAB, and corresponding size distributions of spherical micelles: b) DTAB and d) OTAB.

Supplementary Fig. 4 Micellar morphology observed by TEM with the addition of acetic acid at room temperature: a) DTAB and c) OTAB, and corresponding size distributions of spherical micelles: b) DTAB and d) OTAB.

To further approximate the synthesis conditions, we raised the temperature of the micelle assembly to 120 °C for 3 days in a sealed tube, after which the assembly product was collected for TEM observation, and the results are shown in **Supplementary Fig. 5**. When the temperature increases, both DTAB and OTAB exhibit a small number of spherical shapes accompanied by a large number of cylindrical micelle assemblies. The transverse dimensions of these cylindrical micelles are consistent with the DTAB/OTAB spherical micelles, approaching 3.4 and 4.3 nm, respectively, suggesting that they are derived from the further assembly of spherical micelles. Therefore, we conclude that the increase in temperature will lead to the further transformation of spherical micelles into cylindrical micelles. Similar examples of block copolymer assembly to form cylindrical micelles have been reported in the previous

Location	Revised
	literatures. ²⁻⁴  Supplementary Fig. 5 Micellar morphology observed by TEM after heating assembly in a 120 °C water bath for 3 d: a) DTAB and b) OTAB.

3. The authors simulated the dimensions of different micelles by theoretical calculations. Can the authors provide the actual sizes of the micelles determined by physical characterization such as dynamic light scattering (DLS) methods? Does the morphology of micelles change with the addition of acetic acid and the increase in temperature?

Response: Thank you very much for the reviewers' suggestions. Based on your suggestion, we performed DLS and TEM tests on the micelles formed by the surfactant to further determine the actual size and morphology of the micelles. The DLS results showed that at 25 °C, the particle sizes of DTAB/OTAB were 3.62/4.85 nm, respectively, which was relatively larger compared to the d-spacing of 3.4/4.3 nm, while the results of TEM were more similar (3.29/4.45 nm) (Fig. R2-4). The main reason for this difference in particle size is that the DLS test yields a hydraulic diameter, which is larger than the actual size. The larger micelles observed in the DLS and TEM results are likely due to the formation of larger agglomerates resulting from disorder or surfactant drying.

Subsequently, to investigate whether the addition of acetic acid would affect the morphology of micelles, we performed TEM tests on the micelles after the addition of acetic acid. The results showed that the addition of acetic acid did not affect the morphology of the OTAB and DTAB (Fig. R2-5).

Furthermore, to investigate whether the increase in temperature will affect the

morphology of micelles, we first increased the temperature from room temperature to the upper limit of 40 °C in the DLS test. The results showed that the increase in temperature does not affect the size at 40 °C. Subsequently, we also used TEM to observe the micelles maintained for 3 days at 120 °C, and the results are shown in Fig. R2-6. The higher temperature results in the micelles forming cylindrical assemblies as described in our previous response to your concerns.

Fig. R2-4 a) DTAB and b) OTAB micelles size distributions by DLS at different temperatures.

Fig. R2-5 Micellar morphology observed by TEM at room temperature: a) DTAB and c) OTAB, and corresponding size distributions of spherical micelles: b) DTAB and d) OTAB, and micelle morphology with the addition of acetic acid: e) DTAB and g) OTAB, and corresponding spherical micelle size distributions: f) DTAB and h) OTAB.

Fig. R2-6 Micellar morphology observed by TEM after heating assembly in a 120 °C water bath for 3 d: a) DTAB and b) OTAB.

List of additions:

Location	Revised
Supplementary information Pages S10-12	Supplementary Section 5. TEM Images of DTAB/OTAB Self-Assembling into Micelles To gain further insights into the micelles state within our system, we first tried to directly observe the morphology of the micelles by TEM at room temperature, and the results are shown in Supplementary Fig. 3. According to Supplementary Fig. 3, we can see that both DTAB/OTAB exhibit spherical micelles and the Gaussian statistics can be used to obtain their average sizes of 3.29 and 4.45 nm, respectively, which is very close to the observed d-spacing calculated from the surfactant sizes and XRD patterns (3.4 and 4.3 nm, respectively). The larger micelles in the figure may be due to the agglomeration of micelles during TEM sample drying (Supplementary Fig. 3a). Subsequently, to our experimental conditions more closely, we also investigated whether the addition of acetic acid affected the morphology of the micelles (Supplementary Fig. 4). The results showed that the addition of acetic acid did not affect the micelle morphology of OTAB and DTAB.

Location	Revised
----------	---------

Supplementary Fig. 3 Micellar morphology observed by TEM at room temperature: a) DTAB and c) OTAB, and corresponding size distributions of spherical micelles: b) DTAB and d) OTAB.

Supplementary Fig. 4 Micellar morphology observed by TEM with the addition of acetic acid at room temperature: a) DTAB and c) OTAB, and corresponding size distributions of spherical micelles: b) DTAB and d) OTAB.

To further approximate the synthesis conditions, we raised the temperature of the micelle assembly to 120 °C for 3 days in a sealed tube, after which the assembly product was collected for TEM observation, and the results are shown in **Supplementary Fig. 5**. When the temperature increases, both DTAB and OTAB exhibit a small number of spherical shapes accompanied by a large number of cylindrical micelle assemblies. The transverse dimensions of these cylindrical micelles are consistent with the DTAB/OTAB spherical micelles, approaching 3.4 and 4.3 nm, respectively, suggesting that they are derived from the further assembly of spherical micelles. Therefore, we conclude that the increase in temperature will lead to the further transformation of spherical micelles into cylindrical micelles. Similar examples of block copolymer assembly to form cylindrical micelles have been reported in the previous

Location	Revised
	literatures.²⁻⁴    Supplementary Fig. 5 Micellar morphology observed by TEM after heating assembly in a 120 °C water bath for 3 d: a) DTAB and b) OTAB.
Supplementary information Page S13	Supplementary Section 6. DLS Data of DTAB/OTAB Self-Assembling into Micelles To further clarify the size of the micelles formed by the self-assembly of DTAB/OTAB, we also observed them using DLS testing, and the results showed that the particle sizes were distributed at 3.62/4.85 nm at 25 °C, respectively (Supplementary Fig. 6). The main reason for this difference in particle size is that the DLS test yields a hydraulic diameter, which is larger than the actual size. In addition, many larger-sized micelles resulting from disordered agglomeration were also observed. Finally, to investigate whether the increase in temperature will affect the morphology of micelles, we looked at whether the increase in temperature would affect the size of the micelles. The results showed that the addition of acetic acid did not affect the micelle morphology of OTAB and DTAB.    Supplementary Fig. 6 a) DTAB and b) OTAB micelles size distributions by DLS at different temperatures.

4. HR-TEM exhibits the lattice fringes of the (001) and (100) crystal planes attributed to OHMMCOF-DTAB/-OTAB and OHMMCOF-1/-2, respectively. Yet why are the newly formed ordered mesopores not observed? In fact, the diffraction peak signals corresponding to the ordered mesopores near 2.0-2.5° are stronger than both the 100 and 001 peaks of the COF (Figures 1B and 3B), suggesting that these mesopores are more ordered and thus should be preferentially observed. These results appear to be contradictory. In addition, does the PXRD diffraction signal corresponding to ordered

mesopores shift when the template is removed?

Response: Thank you very much for your questions. In fact, we are also very interested in observing the presence of ordered mesopores through HR-TEM, as this is more direct evidence. In our previous experiments, we thought that the main reason why no mesopores were observed was that the pretreatment of our HR-TEM test was to preferentially use a cell pulverizer to make the material smaller and thinner, but the results showed that the material was still large and thick, which led to the fact that we did not observe the presence of ordered mesopores in multiple attempts. As a result, we resynthesized the ordered hierarchical porous COFs and changed the pretreatment for HR-TEM testing. We tried to observe the mesopores by slicing and thinning the sample, but the incompatibility of the sample with the encapsulant epoxy resin made the experiment impossible. Subsequently, we planned to further thin the sample and expose the ordered mesopores by comminuting the ultrasound for a long time, but the sample was still very thick after tens of hours of crushing, so we still did not observe the ordered mesopores. On the other hand, we believe that COFs are very sensitive to electron beams, and the ordered arrangement of micelles or mesopores is 'defective' compared to COFs, so they are more sensitive to electron beams, which is one of the reasons why we do not observe their orderly arrangement. We regret that we did not observe the ordered arrangement of micelles or mesopores under TEM, but this does not affect the construction of ordered hierarchical micropores/mesoporous COFs by the soft template method proposed in this work, because other experiments including nitrogen adsorption-desorption have demonstrated the ordered micropores/mesopores of the materials.

We then compared the PXRD of the samples before and after template removal in detail. As shown in Fig. R2-7, the Bragg diffraction peaks corresponding to the ordered mesopores on the PXRD spectrum are shifted 0.1° to the right of the OHMMCOF-1/2 after template removal compared to the OHMMCOF-DTAB/OTAB with template. This deviation is very small, it could be a measurement error or may be due to the flexible compression of the COF backbone after template removal, resulting in a slight

reduction of the pore.

Fig. R2-7 Comparison of PXRD patterns of samples before and after template removal.

5. All of the COFs in this manuscript exhibit low specific surface areas (less than 100 m²/g), and in particular the specific surface area of TpTGCl COF (26.9 m²/g) is much lower than those synthesized in the literature (200~300 m²/g) using conventional solvothermal methods (10.1021/jacs.5b13533; 10.1021/jacs.7b12292). These results make this approach uncompetitive. It is therefore unrepresentative to use only MPCOF (Section 2) as a comparison sample. This would probably better reflect the advanced nature of this strategy if MPCOF was constructed as a comparison sample using the currently commonly used solvothermal method.

Response: We sincerely thank the reviewer for their suggestions. As you mentioned, the reported COFs in our work have a relatively low specific surface areas compared to the reported imine COFs, which may be mainly due to the following two reasons. Firstly, it may be that the widely free anions/cations in the pores, which is also the key factor for the generally low BET of other reported ions COFs (ACS Appl. Mater. Interfaces 2016, 8, 18505-18512: 69 m² g⁻¹ for NUS-10(R); Chem. Mater. 2016, 28, 1489-1494: 215 m² g⁻¹ for TpPa-SO₃H; J. Am. Chem. Soc. 2018, 140, 896-899: 220 m² g⁻¹ for CON-Cl; Chem. Commun. 2023, 59, 14435-14438: 53, 103 and 113 m² g⁻¹ for C4-IL/ICOF, C6-IL/ICOF and C10-IL/ICOF, respectively; J. Am. Chem. Soc. 2024,

146, 2313-2318: $57 \text{ m}^2 \text{ g}^{-1}$ for TpPa-SO₃H). On the other hand, it may be that our activation process is not complete, resulting in the presence of water or some solvents in the pores. Sulfonic acid is highly hydrophilic, which leads to some pores being occupied by water molecules, thus reducing the specific surface area. Additionally, according to the research results of Zhao Dan et al., the introduction of guest molecules may promote the interlayer sliding of 2D COFs (J. Am. Chem. Soc. 2023, 145, 1359-1366; J. Am. Chem. Soc. 2020, 142, 12995-13002), increasing the disorder of the system and further reducing the material's specific surface area. Therefore, we resynthesized the COFs and pretreated the material under vacuum conditions with an activation temperature of 150 °C, and then tested their specific surface area, the results of which are shown in Fig. R2-8. All activated COFs showed higher BET values than the original manuscript, further verifying the significance of hierarchical porous COFs. In addition, using the same activation method, we supplemented the experimental information on nitrogen adsorption-desorption of representative COFs that adjusted the mesoporous/microporous ratio and mesoporous size in the manuscript. Thanks again to the reviewers for the valuable suggestions, which helped us to further strengthen our work. We have made changes to the corresponding positions in the revised manuscript and supplementary information.

Fig. R2-8 The nitrogen adsorption-desorption isotherms at 77 K: a) in the original manuscript after activation at 120 °C for 24 h and b) in the revised manuscript after high-temperature activation at 150 °C for 24 h.

At the same time, we synthesized MPCOF-Organic by solvothermal method, and its specific surface area can reach $112.7 \text{ m}^2/\text{g}$ after the same high-temperature activation treatment, which is higher than that of the material synthesized with water as a solvent.

We speculate that this may be because water as a solvent reduces the reversibility of the reaction and thus the crystallinity. However, water promotes the self-assembly of surfactants and is irreplaceable in our entire synthesis system. Therefore, based on your suggestion and our experimental design, we first performed U/Th kinetic adsorption experiments on MPCOF synthesized in the aqueous phase and activated at high temperatures, and the results showed that their kinetics and adsorption capacity changed very little, indicating that the increase of specific surface area would not affect the results of the series of experiments in the manuscript (Fig. R2-9). After that, we performed U/Th kinetic adsorption experiments on MPCOF-Organic. The results showed that the adsorption kinetics and adsorption capacity of MPCOF-Organic were slightly improved compared with the MPCOF, but they were still much lower than those of OHMMCOFs. We have added this data in the Supplementary information Section 32 and made corresponding descriptions and modifications in the manuscript, please review it.

Fig. R2-9 Adsorption kinetics at A) 24 h and B) 10 min, for pictures A and B: (a) MPCOF-U, (b) MPCOF-Th, (c) MPCOF-150 °C-U, (d) MPCOF-150 °C-Th, (e) MPCOF-Organic-U, and (f) MPCOF-Organic-Th.

List of modifications:

Location	Original	Revised
Manuscript Page 11 Lines 1-6	 Figure 2. High-resolution XPS spectra of A) N 1s and B) S 2p, C) nitrogen adsorption-desorption isotherms at 77 K and D) the corresponding pore size distributions, and E) the HRTEM images. For all pictures: (a) MPCOF, (b) OHMMCOF-DTAB, and (c) OHMMCOF-OTAB.	 Fig. 2 Characterizations of the structure and porosity parameters of MPCOF, OHMMCOF-DTAB, and OHMMCOF-OTAB. High-resolution XPS spectra of A) N 1s and B) S 2p, C) nitrogen adsorption-desorption isotherms at 77 K and D) the corresponding pore size distributions, and E) the HRTEM images. For all pictures: (a) MPCOF, (b) OHMMCOF-DTAB, and (c) OHMMCOF-OTAB.
Manuscript Page 12 Line 27 and Page 13 Line 1	The BET surface areas of OHMMCOF-1/OHMMCOF-2 after removal of DTAB and OTAB increased to 99.6 and 37.9 m²/g compared to the OHMMCOF-DTAB/OHMMCOF-OTAB, respectively, which could be attributed to the removal of long-chain alkyl quaternary amine ions in the pores. The pore size distributions calculated by NLDFT revealed predominant pore sizes of 1.8 nm and additional mesoporous pore sizes of 3.3 nm for OHMMCOF-1, and 1.7 nm and 4.2 nm for OHMMCOF-2, which is very close to the theoretical values (Figure 1b).	The BET surface areas of OHMMCOF-1/OHMMCOF-2 after removal of DTAB and OTAB increased to 170.8 and 112.2 m²/g compared to the OHMMCOF-DTAB/OHMMCOF-OTAB, respectively, which could be attributed to the removal of long-chain alkyl quaternary amine ions in the pores. The pore size distributions calculated by NLDFT revealed predominant pore sizes of 1.8 nm and additional mesoporous pore sizes of 3.3 nm for OHMMCOF-1, and 1.8 nm and 4.3 nm for OHMMCOF-2, which is very close to the theoretical values (Fig. 1b).
Manuscript Page 13 Lines 11-17		
Location	Original	Revised
	Figure 3. A) Illustration of template removal using ion exchange method, B) PXRD patterns, C) FT-IR spectra, D) ¹³C SSNMR spectra, high-resolution XPS spectra of E) N 1s and F) S 2p, G) nitrogen adsorption-desorption isotherms at 77 K and H) the corresponding pore size distributions. For pictures B-H: (a) OHMMCOF-1 and (b) OHMMCOF-2.	Fig. 3 Characterizations of crystallinity, structure, and porosity parameters of OHMMCOF-1 and OHMMCOF-2. A) Illustration of template removal using ion exchange method, B) PXRD patterns, C) FT-IR spectra, D) ¹³C SSNMR spectra (where the asterisks (*) represent the spinning sidebands), high-resolution XPS spectra of E) N 1s and F) S 2p, G) nitrogen adsorption-desorption isotherms at 77 K and H) the corresponding pore size distributions. For pictures B-H: (a) OHMMCOF-1 and (b) OHMMCOF-2.
Manuscript Page 14 Lines 23-30 and Page 15 Lines 1-7	These results showed that the template proportion in the final COFs increased with the increase of their concentration, and this phenomenon was especially obvious when using the less hydrophobic DTAB as the template. The difference in solubility between DTAB and OTAB contributed to their distinct behavior in the reaction. The higher solubility of DTAB allowed for a continued increase in its actual reaction concentration, while the lower solubility of OTAB limited its participation in the reaction (Figure S33).	These results showed that the template proportion in the final COFs increased with the increase of their concentration, implying that the mesoporous/microporous ratio can also be easily adjusted by this pathway, and this phenomenon was especially obvious when using the less hydrophobic DTAB as the template. The difference in solubility between DTAB and OTAB contributed to their distinct behavior in the reaction. The higher solubility of DTAB allowed for a continued increase in its actual reaction concentration, while the lower solubility of OTAB limited its participation in the reaction (Supplementary Fig. 27). OHMMCOF-DTAB-0.5 eq and OHMMCOF-DTAB-3.0 eq were selected for template removal experiments, followed by N₂ adsorption-desorption tests, and compared with OHMMCOF-DTAB-1.0 eq to verify the above conjectures. OHMMCOF-1-0.5 eq exhibited slightly lower specific surface area (159.0 m²/g) and S_{BET-meso}/S_{BET-micro} (5.5) than

Location	Original	Revised
		OHMMCOF-1-1.0 eq (170.8 m ² /g and 6.8), while OHMMCOF-1-3.0 eq exhibited higher specific surface area (178.1 m ² /g) and S _{BET-meso} /S _{BET-micro} (7.5) (Fig. 4C-D and Supplementary Fig. 28, Table 3, 6).
Manuscript Page 15 Lines 24-30 and Page 16 Lines 1-4	Although changing the ratio of DTAB/OTAB in the system could not synthesize ordered hierarchically microporous/mesoporous COFs with two different mesopores, it was unexpectedly found that the micelle size in the final COFs could be controlled, which opens up new possibilities for more straightforward changes in mesopore size.	Although changing the ratio of DTAB/OTAB in the system could not synthesize ordered hierarchically microporous/mesoporous COFs with two different mesopores, it was unexpectedly found that the micelle size in the final COFs could be controlled, which means that this may be a new way to control the mesoporous size. Two materials with DTAB/OTAB of 14:1 and 1:2 was selected for the template removal experiment, and the resulting COFs were named OHMMCOF-M-14:1 and OHMMCOF-M-1:2 (while M represents the mixture), respectively, and then N ₂ adsorption-desorption experiments were performed to verify the above conjecture (Fig. 4F and Supplementary Fig. 29, Table 7). The mesoporous pore size of OHMMCOF-M-14:1 was mainly 3.6 nm, which was relatively close to that of the mesopores constructed entirely with DTAB self-assembled micelles (3.3 nm), while the mesoporous pore size of OHMMCOF-M-1:2 was mainly 4.5 nm, which was also relatively close to the mesopores constructed entirely with OTAB self-assembled micelles (4.3 nm) (Fig. 4G).
Manuscript Page 16 Lines 20, 23	As shown in Figure S24 , OHMMCOF-3 also exhibited IV-type characteristic nitrogen adsorption isotherms as well as H3-type hysteresis loop, indicating the presence of mesopores in the material.	As shown in Fig. 4I and Supplementary Fig. 30 , OHMMCOF-3 also exhibited IV-type characteristic nitrogen adsorption isotherms as well as H3-type hysteresis loop, indicating the presence of mesopores in the material. The

Location	Original	Revised
	The specific surface area was measured to be 26.9 m²/g, which was relatively low due to the large number of anions present in the framework. However, NLDFT calculation revealed the presence of micropores with a size of 1.3 nm, closer to the theoretical value. Additionally, the removal of spherical micelles constructed with SLS resulted in the formation of additional mesopores with a size of 3.3 nm.⁷¹	specific surface area was measured to be 104.3 m²/g, which was relatively low due to the large number of anions present in the framework. However, NLDFT calculation revealed the presence of micropores with a size of 1.3 nm, closer to the theoretical value. Additionally, the removal of spherical micelles constructed with SLS resulted in the formation of additional mesopores with a size of 3.3 nm.⁷¹
Manuscript Page 17 Lines 1-10	 Figure 5. PXRD patterns of A) OHMMCOF-DTAB and B) OHMMCOF-OTAB corresponding to different input concentration, adsorption kinetics at C) 24 h and D) 10 min, for pictures C and D: (a) OHMMCOF-1-0.5 eq-U, (b) OHMMCOF-1-0.5 eq-Th, (c) OHMMCOF-1-3.0 eq-U, and (d) OHMMCOF-1-3.0 eq-Th, E) PXRD patterns corresponding to templates with different concentration ratios of DTAB/OTAB, F) simulated and experimental PXRD patterns (inset: the simulated size of the SLS micelles) of OHMMCOF-3.	 Fig. 4 Pore regulation and system expansion of OHMMCOFs. PXRD patterns of A) OHMMCOF-DTAB and B) OHMMCOF-OTAB corresponding to different input concentration, C) nitrogen adsorption-desorption isotherms at 77 K and D) the corresponding pore size distributions, for C and D: a) OHMMCOF-1-0.5 eq and b) OHMMCOF-1-3.0 eq, E) PXRD patterns corresponding to templates with different concentration ratios of DTAB/OTAB, F) nitrogen adsorption-desorption isotherms at 77 K and G) the corresponding pore size distributions, for F and G: a) OHMMCOF-M-14:1 and b) OHMMCOF-M-1:2, H) simulated and experimental PXRD patterns of OHMMCOF-3 (inset: the simulated size of the SLS micelles), and I)

Location	Original	Revised
		nitrogen adsorption-desorption isotherms at 77 K of OHMMCOF-3 (inset: the corresponding pore size distributions).

List of additions:

Location	Revised
Supplementary information Pages S51, S52	Supplementary Section 32. Experimental Results of MPCOF-Organic Synthesized by Organic Solvothermal Method The solvothermal method was used to select trimethylbenzene and dioxane with a volume ratio of 1:1 as the synthesis solvents, and the parent COF was synthesized and named MPCOF-Organic. Subsequently, PXRD and N₂ adsorption-desorption were used to characterize the crystallinity and porosity of the material, respectively. Finally, the U(VI)/Th(IV) kinetic adsorption properties of the material were evaluated (Supplementary Fig. 39). MPCOF-Organic exhibited good crystallinity and higher BET and pore volume than aqueous MPCOF, which may be because water as a solvent reduces the partial reversibility of the synthesis process. The results of U(VI)/Th(IV) kinetic adsorption experiments showed that MPCOF-Organic (23.2 mg/g-U(VI) and 58.3 mg/g-Th(IV)) and MPCOF exhibited almost the same U(VI)/Th(IV) adsorption capacity, and the U(VI)/Th(IV) kinetics are relatively weakly increased (0.7 mg/(g·min)-U(VI) and 1.7 mg/(g·min)-Th(IV) in the first 5 minutes).
	a PXRD pattern showing Intensity (a.u.) vs 2 Theta (degree) with peaks at approximately 5, 10, 15, 20, 25, and 30 degrees. b N₂ uptake (cm³/g) vs Relative Pressure (P/P₀) showing a hysteresis loop. Text in plot: $S_{BET} = 112.7 \text{ m}^2/\text{g}$, $V_{pore} = 0.42 \text{ cm}^3/\text{g}$. c Pore volume $V^{-1}[(P_0/P)]^{-1}$ vs P/P_0 showing a linear relationship. Regression statistics: $y = 0.07x$, Intercept: $-1.81205E-4 \pm 1.28245E-4$, Slope: $0.04179 \pm 7.31117E-4$, Residual Sum of Squares: 2.97737E-7, Pearson's r: 0.99878, R-Square (COD): 0.99726, Adj. R-Square: 0.99725. d Pore size distribution showing a sharp peak at 1.8 nm vs Pore Width (nm).

6. The authors claim that “this work provides a brand new path for constructing ordered and tunable extrinsic porosity in COFs”. However, the manuscript only discusses one

anionic COF and one cationic COF; is this strategy also valid for neutral COFs, which are currently the most widely used?

Response: Thank you very much for your question. In our manuscript, we propose a novel approach for constructing hierarchical porous COFs based on a soft template strategy. The key to the success of this method lies in the binding of the surfactant to the COF backbone through ionic bonds, which avoids the incompatibility of surfactant self-assembly and COF crystallization. It is important to note that the strategy proposed in this study serves as a starting point to inspire further developments in this field. We are exploring the potential extension of this strategy to a broader range of neutral COFs and the compatibility with non-ionic surfactants, including those with higher molecular weights. The central focus remains on enabling the self-assembly of surfactants/micelles and the crystallization of COFs while circumventing their inherent incompatibility. Therefore, while this work was going on, we were already thinking and starting to try. Our ongoing experimentation has revealed that without ionic bonding, direct mixing of COFs and surfactants/micelles results in separate heteronucleated products, as observed in our experiments. We experimented with mixed crystallization of neutral monomers and ionic surfactants using the synthetic pathway shown in Scheme R2-1 using a method consistent with our manuscript. The results show that due to the weak reversibility of imine in the aqueous phase, the product cannot be obtained by Pathway 2. Although the product of pathway 1 has good COF crystallization, it does not contain the self-assembly of surfactants, and the yield is still very low.

Scheme R2-1 Crystallization and synthesis of neutral monomer and ionic surfactant DTAB.

To address this challenge, we are considering two main avenues of exploration. Firstly, we are investigating the feasibility of establishing a linkage between

surfactants/micelles, metal ions, and the COF skeleton through coordination bonds, and utilization of good coordination ability of COF structures. Additionally, we are exploring the concept of inducing charge in neutral COFs/surfactants/micelles, such as through local protonation, a strategy previously demonstrated in Metal-Organic Frameworks (MOFs) literature (Chem. Mater. 2012, 24, 2253-2255), to assess its applicability to COFs.

We've tried a strategy with local protonation. In this attempt, we dissolved P123 (polyethylene oxide–polypropylene oxide–polyethylene oxide, an amphiphilic triblock neutral polymer) in an aqueous solution, and then adjusted the pH to 2-3 to achieve local protonation of micelles, inspired by the principles outlined in the aforementioned literature (Chem. Mater. 2012, 24, 2253-2255) (Fig. R2-10). Subsequently, we used the same synthesis method as in the present manuscript to mix and react them with monomers, to construct microporous COFs with an ordered micelle arrangement using the ionic bonds (Fig. R2-11). We found the presence of methylene in the corresponding FT-IR spectra, but we did not find an ordered micelle arrangement in the corresponding small-angle XRD, which means that we did not achieve our goal. The possible reasons are as follows: according to the literature, the size of the ordered micelle arrangement formed by P123 will be close to 14 nm, and the SEM images we have observed show that the material is heterogeneous and difficult to form a large size, which may hinder the formation of a large, ordered micelle arrangement. COFs that are larger and easier to crystallize have the potential to achieve our approach. In conclusion, we believe that this work will open a new chapter and provide a new and more universal way to construct ordered and tunable extrinsic porosity in COFs.

Fig. R2-10 Schematic model for the formation of mesoporous MOFs. The coordination numbers in

the right figure did not reflect the real situation. (Chem. Mater. 2012, 24, 2253-2255)

Fig. R2-11 COFs constructed using locally protonated P123 as a template: a) 0.5-10° small-angle PXRD spectra, b) 2-10° magnified PXRD spectra, c) FT-IR spectra, and d) SEM image.

7. The TGA in Sections S9 and S18 suggests incomplete sample activation. A weight loss of nearly 10% of the sample was observed before 100°C. In addition, the vertical scale of the TGA should be 0-100% instead of 0-1%.

Response: Thank you very much for the reviewers' suggestions. In our manuscript, the samples were preconditioned at 50 °C in a vacuum before the TGA experiment, but the results showed that there was still a lot of water in the sample. This may be because the material has a good ability to absorb water, and despite after vacuum activation, some of the water is still confined to the pores, leading to the observed low BET surface area. Therefore, we resynthesized the samples and vacuum-dried them overnight at a much higher temperature of 150 °C, and the results showed almost complete removal of water. In addition, we apologize for the error in the vertical scale and have made corresponding changes, all changes have been highlighted in the manuscript, please check them out.

List of modifications:

Location	Original	Revised
Manuscript Page 9 Lines 18-19	Thermogravimetric analysis (TGA) results showed that all three COFs are stable up to 270 °C, with the OHMMCOF-DTAB/OTAB materials exhibiting better stability than MPCOF. In addition, the TGA curves of both OHMMCOF-DTAB/OHMMCOF-OTAB showed an additional weightless plateau due to the long-chain alkyl disintegration at around 270-360 °C (Fig. S10).	Thermogravimetric analysis (TGA) results showed that all three COFs are stable up to 270 °C, with the MPCOF materials exhibiting better stability than OHMMCOF-DTAB/OTAB. In addition, the TGA curves of both OHMMCOF-DTAB/OHMMCOF-OTAB showed an additional weightless plateau due to the long-chain alkyl disintegration at around 270-360 °C (Supplementary Fig. 11).⁶⁴
Supplementary information Page S19	 Figure S10. TGA curves of a) MPCOF, b) OHMMCOF-DTAB, and c) OHMMCOF-OTAB.	 Supplementary Fig. 11 TGA curves of a) MPCOF, b) OHMMCOF-DTAB, and c) OHMMCOF-OTAB.
Supplementary information Page S31	 Figure S20. TGA curves of OHMMCOF-DTAB/OHMMCOF-OTAB and OHMMCOF-1/OHMMCOF-2.	 Supplementary Fig. 21 TGA curves of OHMMCOF-DTAB/OHMMCOF-OTAB and OHMMCOF-1/OHMMCOF-2.

8. Some more interpretation of the peaks at 25-50 ppm in the SSNMR (Figure 3D) would be helpful.

Response: Thanks for the reviewer's question. Based on your concern, we compared

the ^{13}C SSNMR spectra of the parent MPCOF with the OHMMCOF-DTAB/OTAB before template removal and OHMMCOF-1/2 after template removal and combined the MPCOF with OHMMCOF-1/2 into Fig. R2-12. The results show that the apparent peaks of around 50 ppm are not part of the surfactant template, as these peaks are also present in MPCOF. They may be rotational sideband peaks produced by the sample when it rotates at high speeds. To fully illustrate this result, we have supplemented the annotations of these unknown peaks in the corresponding figures and explained them in the legends, as shown in Figs. 1 and 3 in the manuscript.

Fig. R2-12 ^{13}C SSNMR spectra of MPCOF, OHMMCOF-1, and OHMMCOF-2.

List of modifications:

Location	Original	Revised
Manuscript Page 10 Lines 1-6	 Figure 1. A) Simulated and experimental PXRD patterns of MPCOF, B) PXRD patterns (inset: the simulated size of the templates), C) FT-IR spectra, and D) ^{13}C SSNMR spectra. For pictures B-D:	 Fig. 1 Crystallinity and structural characterizations of MPCOF, OHMMCOF-DTAB, and OHMMCOF-OTAB. A) Simulated and experimental PXRD patterns of MPCOF, B) PXRD patterns (inset: the

Location	Original	Revised
	(a) MPCOF, (b) OHMMCOF-DTAB, and (c) OHMMCOF-OTAB.	simulated size of the templates), C) FT-IR spectra, and D) ¹³ C SSNMR spectra (where the asterisks (*) represent the spinning sidebands). For pictures B-D: (a) MPCOF, (b) OHMMCOF-DTAB, and (c) OHMMCOF-OTAB.
Manuscript Page 13 Lines 11-17	 Figure 3. A) Illustration of template removal using ion exchange method, B) PXRD patterns, C) FT-IR spectra, D) ¹³C SSNMR spectra, high-resolution XPS spectra of E) N 1s and F) S 2p, G) nitrogen adsorption-desorption isotherms at 77 K and H) the corresponding pore size distributions. For pictures B-H: (a) OHMMCOF-1 and (b) OHMMCOF-2.	 Fig. 3 Characterizations of crystallinity, structure, and porosity parameters of OHMMCOF-1 and OHMMCOF-2. A) Illustration of template removal using ion exchange method, B) PXRD patterns, C) FT-IR spectra, D) ¹³C SSNMR spectra (where the asterisks (*) represent the spinning sidebands), high-resolution XPS spectra of E) N 1s and F) S 2p, G) nitrogen adsorption-desorption isotherms at 77 K and H) the corresponding pore size distributions. For pictures B-H: (a) OHMMCOF-1 and (b) OHMMCOF-2.

9. Can the authors add pore volume data for samples before and after template removal?

Response: Thanks very much to you for the helpful suggestions to improve our work.

We have supplemented the pore volume parameters of all materials in the revised supporting information, which are modified in detail as follows.

List of additions:

Supplementary Table 1. Pore volume parameters of MPCOF and OHMMCOF-DTAB/OHMMCOF-OTAB.

COF	V_{total} (cm ³ /g)	$S_{\text{BET-micro}}$ (m ² /g)	$S_{\text{BET-meso}}$ (m ² /g)	$S_{\text{BET-meso}}/S_{\text{BET-micro}}$
MPCOF	0.31	19.3	26.9	1.4
OHMMCOF-DTAB	0.17	15.6	20.7	1.3
OHMMCOF-OTAB	0.13	12.0	18.1	1.5

Supplementary Table 3. Pore volume parameters of MPCOF and OHMMCOF-DTAB/OHMMCOF-OTAB.

COF	V_{total} (cm ³ /g)	$S_{\text{BET-micro}}$ (m ² /g)	$S_{\text{BET-meso}}$ (m ² /g)	$S_{\text{BET-meso}}/S_{\text{BET-micro}}$
OHMMCOF-1	0.61	21.8	149.0	6.8
OHMMCOF-2	0.45	10.8	101.4	9.4

Supplementary Table 6. Pore volume parameters of OHMMCOF-1-0.5/3.0 eq.

COF	V_{total} (cm ³ /g)	$S_{\text{BET-micro}}$ (m ² /g)	$S_{\text{BET-meso}}$ (m ² /g)	$S_{\text{BET-meso}}/S_{\text{BET-micro}}$
OHMMCOF-1-0.5 eq	0.59	24.5	134.5	5.5
OHMMCOF-1-3.0 eq	0.78	21.0	157.1	7.5

Supplementary Table 7. Pore volume parameters of OHMMCOF-M-14:1/1:2.

COF	V_{total} (cm ³ /g)	$S_{\text{BET-micro}}$ (m ² /g)	$S_{\text{BET-meso}}$ (m ² /g)	$S_{\text{BET-meso}}/S_{\text{BET-micro}}$
OHMMCOF-M-14:1	0.44	23.5	147.7	6.3
OHMMCOF-M-1:2	0.35	14.5	120.1	8.3

10. To demonstrate the superiority of the hierarchically porous COFs, it is necessary to compare the ion adsorption properties of OHMMCOF-1/-2 with those of porous materials reported in the literature.

Response: Thanks to the reviewer's valuable suggestion. Combined with our subsequent application system, we first supplemented the Th(IV)/U(VI) and Th(IV)/Ln

separation experiments, the specific calculation formulas and experiment results have been supplemented in the Supplementary information (Supplementary Sections 3, 36). Through these experiments, we obtained the distribution coefficient (K_d) and separation factor $SF_{(Th/U, Ln)}$ of OHMMCOF-2 for different ions (Supplementary Fig. 47). We then compared it with the reported porous materials (Supplementary Section 37) and found that OHMMCOF-2 exhibited good Th(IV) adsorption capacity, K_d , and $SF_{(Th/U, Ln)}$. It is important to note that due to the lack of specific adsorption rate data within the initial 5-minute timeframe in much of the literature, a comprehensive comparison could not be made in the table. For a detailed summary, please refer to Supplementary Table 14.

List of modifications:

Location	Original	Revised
Manuscript Page 20 Lines 21-29	However, NLDFT calculation revealed the presence of micropores with a size of 1.3 nm, closer to the theoretical value. Additionally, the removal of spherical micelles constructed with SLS resulted in the formation of additional mesopores with a size of 3.3 nm.⁷¹	The high-resolution spectra of N 1s showed little change, while both S-O (532.4 eV) and S 2p (S 2p_{3/2} = 168.7 eV and S 2p_{1/2} = 167.5 eV) showed weak shifts to high binding energy direction, which were caused by the electrostatic attraction between the sulfonic acid groups and U(VI)/Th(IV) when ion exchange occurs (Fig. 5D and Supplementary Fig. 45a-b).⁷⁵ After in-depth study of the pore structure regulation and U(VI)/Th(IV) mass transfer ability of OHMMCOFs, we further investigated the selective adsorption of OHMMCOF-2 on Th(IV)/U(VI) and Th(IV)/Ln at pH=3 (Supplementary Sections 36, 37). The high distribution coefficient (K_d) for Th (4.6×10^3) and the high separation factor (SF) of Th(IV) for other metals ($SF_{Th/U} = 218$ and $SF_{Th/Ln} > 3.2 \times 10^3$) indicate that OHMMCOF-2 has a unique selectivity for Th and is a potential material for Th(IV)/U(VI) and Th(IV)/Ln separation (Supplementary Fig. 47).

Location	Original	Revised
		OHMMCOF-2 also demonstrated excellent selective adsorption ability for Th(IV) when compared with other porous materials (Supplementary Table 14).
Supplementary information Page S7	Section 3. Adsorption Experiments of U(VI)/Th(IV) All adsorption experiments were performed using a thermostatic shaking chamber at 25 °C with a vibration frequency of 200 rpm. The experiments were repeated more than three times and then averaged to ensure the reproducibility of the experimental results. Metal ion concentrations in solutions before and after sorption were determined using an ICP-OES PE OPTIMA 8000DV spectrometer. The equilibrium adsorption capacity (Q_e, mg/g) of U(VI)/Th(IV) was calculated using the following equation: $Q_e = \frac{(C_0 - C_e)}{m} \times V \text{ (S1)}$ where C_0 (ppm) and C_e (ppm) are the initial and equilibrium concentrations of U(VI)/Th(IV), respectively; V (L) is the volume of the solution; and m (g) is the mass of the adsorbent.	Section 3. Adsorption Experiments All adsorption experiments were performed using a thermostatic shaking chamber at 25 °C with a vibration frequency of 200 rpm. The experiments were repeated more than three times and then averaged to ensure the reproducibility of the experimental results. Metal ion concentrations in solutions before and after sorption were determined using an ICP-OES OPTIMA8000 spectrometer. The equilibrium adsorption capacity (Q_e, mg/g), distribution coefficient (K_d, mL/g), and separation factor (SF) were calculated using the following equations: $Q_e = \frac{(C_0 - C_e)}{m} \times V \text{ (1)}$ $K_d = \frac{(C_0 - C_e)}{C_e} \times \frac{V}{m} \text{ (2)}$ $SF_{Th/M} = \frac{K_{d,Th}}{K_{d,M}} \text{ (3)}$ where C_0 (ppm) and C_e (ppm) are the initial and equilibrium concentrations, respectively; V (L) is the volume of the solution; and m (g) is the mass of the adsorbent.

List of additions:

Location	Revised
Supplementary information Page S62	Supplementary Section 36. Selective Adsorption Experiment of OHMMCOF-2

Location	Revised																																																												
	 Supplementary Fig. 47 a) Distribution coefficients (K_d) and b) separation factors (SF) for thorium and other metals on OHMMCOF-2.																																																												
Supplementary information Page S63	Supplementary Section 37. Comparison of Th(IV) Selective Adsorption Properties of OHMMCOF-2 and Other Porous Materials Supplementary Table 14. Comparison of Th(IV) selective adsorption properties of OHMMCOF-2 and other porous materials*.																																																												
	   Porous materials Conditions Capacity (mg/g) K_d(Th) SF (Th/U) Ref.     GO pH = 3.0, $C_0 \approx 23$ ppm 74.2 N.A. 36.3 6   Salen-containing MOF pH = N.A., $C_0 = 87$ ppm 46.4 N.A. N.A. 7   PPAC pH = 4.0, $C_0 \approx 10$ ppm 2.8 0.33 2.1 8   WMC-O pH = 1.7, $C_0 = 10$ ppm 8.4 1.3×10^5 ~ 200 9   M-DGA pH = 3.0, $C_0 = 90$ ppm 55.6 N.A. N.A. 10   SBA-15-O-DMAP pH = 1.0, $C_0 = 100$ ppm 37.5 2.5×10^3 50 11   KMS-5 pH = 1.0, $C_0 = 50$ ppm 50.0 1.04×10^6 N.A. 12   PEI-GO-AG pH = 3.0, $C_0 = 50$ ppm 12.0 N.A. N.A. 13   HPSM-PGAH pH = 3.0, $C_0 = 20$ ppm 52.4 2.0×10^4 140 14   	Porous materials	Conditions	Capacity (mg/g)	K_d (Th)	SF (Th/U)	Ref.	GO	pH = 3.0, $C_0 \approx 23$ ppm	74.2	N.A.	36.3	6	Salen-containing MOF	pH = N.A., $C_0 = 87$ ppm	46.4	N.A.	N.A.	7	PPAC	pH = 4.0, $C_0 \approx 10$ ppm	2.8	0.33	2.1	8	WMC-O	pH = 1.7, $C_0 = 10$ ppm	8.4	1.3×10^5	~ 200	9	M-DGA	pH = 3.0, $C_0 = 90$ ppm	55.6	N.A.	N.A.	10	SBA-15-O-DMAP	pH = 1.0, $C_0 = 100$ ppm	37.5	2.5×10^3	50	11	KMS-5	pH = 1.0, $C_0 = 50$ ppm	50.0	1.04×10^6	N.A.	12	PEI-GO-AG	pH = 3.0, $C_0 = 50$ ppm	12.0	N.A.	N.A.	13	HPSM-PGAH	pH = 3.0, $C_0 = 20$ ppm	52.4	2.0×10^4	140	14
	Porous materials	Conditions	Capacity (mg/g)	K_d (Th)	SF (Th/U)	Ref.																																																							
	GO	pH = 3.0, $C_0 \approx 23$ ppm	74.2	N.A.	36.3	6																																																							
	Salen-containing MOF	pH = N.A., $C_0 = 87$ ppm	46.4	N.A.	N.A.	7																																																							
	PPAC	pH = 4.0, $C_0 \approx 10$ ppm	2.8	0.33	2.1	8																																																							
	WMC-O	pH = 1.7, $C_0 = 10$ ppm	8.4	1.3×10^5	~ 200	9																																																							
	M-DGA	pH = 3.0, $C_0 = 90$ ppm	55.6	N.A.	N.A.	10																																																							
	SBA-15-O-DMAP	pH = 1.0, $C_0 = 100$ ppm	37.5	2.5×10^3	50	11																																																							
	KMS-5	pH = 1.0, $C_0 = 50$ ppm	50.0	1.04×10^6	N.A.	12																																																							
PEI-GO-AG	pH = 3.0, $C_0 = 50$ ppm	12.0	N.A.	N.A.	13																																																								
HPSM-PGAH	pH = 3.0, $C_0 = 20$ ppm	52.4	2.0×10^4	140	14																																																								

Location	Revised					
	Py-TFImI-25	pH = 4.0, C ₀ = 25 ppm	80	5.5 × 10 ³	~ 200	15
	Py-TFIm-25	pH = 4.0, C ₀ = 25 ppm	140	1.2 × 10 ⁴	~ 400	15
	OHMMCOF-2	pH = 3.0, C ₀ = 50 ppm	71.3	4.6 × 10 ³	218	This work
* Since many literatures do not give specific adsorption rate of the first 5 min, a comprehensive comparison could not be made in the table.						

11. Last but not least, some linguistic and writing errors need to be corrected and embellished. For example, the word “Anionic” on page 5, line 5, the sentence “showed a negative shift of -0.2 eV” on page 8, line 20, the labels “Figure S7” and “Figure S8” on page 22 (SI), and so on.

Response: We are very sorry that there are some linguistic and writing errors in our manuscript, and we apologize for any trouble this may cause. We have changed the description and read and checked the full text again, and all relevant changes have been marked for review. Please check it out.

List of modifications:

Location	Original	Revised
Manuscript Page 5 Line 8	To avoid the template being expelled from the COF growth domain due to the strong driving force of crystallization, Anionic COFs and ionic-type templates are employed, wherein the soft templates and COF backbone are interconnected through ionic bonds.	To avoid the template being expelled from the COF growth domain due to the strong driving force of crystallization, anionic COFs and ionic-type templates are employed, wherein the soft templates and COF backbone are interconnected through ionic bonds.
Manuscript Page 8 Line 29	The high-resolution spectra of S 2p (S 2p _{3/2} = 168.3 eV and S 2p _{1/2} = 167.1 eV) showed a negative shift of -0.2 eV due to the electron-withdrawing effect	The high-resolution spectra of S 2p (S 2p _{3/2} = 168.3 eV and S 2p _{1/2} = 167.1 eV) showed a negative shift of 0.2 eV due to the electron-withdrawing

Location	Original	Revised
	after the introduction of the templates, which further demonstrated the successful combination of the templates and COF backbone (Figure 2B).	effect after the introduction of the templates, which further demonstrated the successful combination of the templates and COF backbone (Fig. 2B).
Manuscript Page 21 Line 19	Through various characterization techniques, we demonstrated the presence and organized arrangement of the templates in the precursor OHMMCOF-DTAB/OHMMCOF-OTAB, while maintaining the crystallinity of the COFs and orderly template arrangement through ion exchange and template removal.	Through various characterization techniques, we demonstrated the presence and organized arrangement of the templates in the precursor OHMMCOF-DTAB/OHMMCOF-OTAB, while maintaining the crystallinity of the COFs and orderly mesoporous arrangement through ion exchange and template removal.
Supplementary information Page S24 Line 12-14	After vacuum drying, OHMMCOF-1/OHMMCOF-2 with ordered hierarchical micropores/mesopores was obtained. Firstly, a detailed conditional optimization experiment was carried out for the template removal experiment of OHMMCOF-DTAB (Figure S7), and then the template removal experimental parameters of OHMMCOF-OTAB was further optimized at 60 °C for 1 d (Figure S8), and PXRD and FT-IR were used to monitor the retention of crystallinity and the removal of the template, respectively.	After vacuum drying, OHMMCOF-1/OHMMCOF-2 with ordered hierarchical micropores/mesopores was obtained. Firstly, a detailed conditional optimization experiment was carried out for the template removal experiment of OHMMCOF-DTAB (Supplementary Fig. 15), and then the template removal experimental parameters of OHMMCOF-OTAB was further optimized at 60 °C for 1 d (Supplementary Fig. 16), and PXRD and FT-IR were used to monitor the retention of crystallinity and the removal of the template, respectively.

Reviewer: 3

In this work, the authors reported the synthesis and characterization of ordered hierarchical microporous/mesoporous COFs using the soft-template method. The ion exchange method is used to remove the templates while maintaining the high crystallinity of the COFs. Extrinsic porosity can be adjusted by changing template length, concentration, and ratio. Although the authors have carried out some experiments and collected significant data, they are common and ordinary, some of them are short of in-depth investigation. It lacks novelty as a research article for publication in Nature Communications.

Response: We appreciate the comments of the reviewers. We understand the reviewers' concerns about the innovation and depth of our work, and we have endeavored to address these by providing expanded explanations of our key contributions and presenting additional detailed findings. In response to the reviewers' comments, we have made significant revisions to our manuscript. We believe that these revisions have significantly improved the quality and depth of our manuscript, aligning it more closely with the high standards of Nature Communications.

1. The introduction section lacks a clear statement of the novelty and significance of the proposed soft-template strategy. Provide a more explicit explanation of how this approach differs from previous methods and its potential impact on COF research.

Response: We thank the reviewer for the valuable suggestions on our manuscripts. At the same time, we apologize for the lack of clarity in the introduction about the novelty and importance of the soft template strategy. Based on the reviewers' suggestions, we have added a clear description of the novelty and importance of the soft template strategy in more detail, and further compared this method with other methods to clarify the potential implications for COF research. The relevant revisions have been brightly colored in the revised manuscript.

List of modifications:

Located	Original	Revised
Manuscript Page 4 Lines 23-28	However, to the best of our knowledge, there has been no published reports on the utilization of the soft-template method to construct ordered hierarchical porous COFs. The main challenge is that the soft-template method relies on the unique hydrophobic interactions of amphoteric surfactants in the aqueous phase to form micellar soft templates, whereas most of the solvents used for COF synthesis are organic solvents. ⁴⁷⁻⁵¹	However, to the best of our knowledge, while surfactants have been employed in the synthesis of certain COFs, their utilization has primarily been limited to leveraging their amphiphilic properties or aiding in the formation of disordered pores, ⁴⁷⁻⁵⁰ and there have been no published reports on the utilization of the soft-template method to construct ordered hierarchical porous COFs. We believe that extending this method to COF chemistry will bring a new and more convenient way to achieve faster mass transfer efficiency for COFs. The main challenge is that the soft-template method relies on the unique hydrophobic interactions of amphoteric surfactants in the aqueous phase to form micellar soft templates, whereas most of the solvents used for COF synthesis are organic solvents. ⁵¹⁻⁵⁵

2. The authors stated that “there has been no published report on the utilization of the soft-template method to construct ordered hierarchical porous COFs”. However, numerous examples of soft template synthesis of COFs have been reported (J. Am. Chem. Soc. 2023, 145, 21974, Soft Matter, 2012, 8, 10801, Chem. Commun., 2022, 58, 9148). In this paper, the authors found that ordered microporous/mesoporous COFs can be synthesized based on the soft template method, which is only suitable for specific monomers and is not universal.

Response: We appreciate the issues pointed out by the reviewers, and we agree that surfactants/soft templates are already being used to construct COFs. For example, Jin et al. used the emulsion template method to use the amphiphilic properties of pyridine

surfactants as emulsifiers for emulsion polymerization, and controlled and synthesized different morphologies of TpPa-COFs (J. Am. Chem. Soc. 2023, 145, 21974); Zheng et al. used the amphiphilic of amino acid derivatives to delay the crystallization of COFs through a dynamic hydrophobic compartment formed by their self-assembled micelles, thereby obtaining multi-gram single crystal COFs (Nat. Chem. 2023, 15, 841-847); Feng et al. and Liu et al. also used the amphiphilic of surfactants to make COFs monomers well preorganized at the water-surfactant interface in advance, to construct COFs with high crystallinity (Nat. Chem. 2019, 11, 994-1000; Adv. Mater. 2023, 35, 2300975). Although Ordonsky et al. achieved the construction of additional pores in the construction of COF-LZU1-NT, the pores were disordered. In addition, the construction monomer of the soft template is a special monomer, which is not universal and adjustable (Chem. Commun. 2022, 58, 9148-9151). Therefore, although surfactants have been used to construct COFs, their utilization has primarily been limited to leveraging their amphiphilic properties or aiding in the formation of disordered pores, while the use of surfactant self-assembled micelles as soft templates to construct hierarchical ordered COFs has not been reported.

Regarding the generality of the reviewers' considerations, we apologize for not giving you a detailed description of the manuscript. First, our approach does not just work for a specific monomer. In our manuscript, the soft template is linked to COFs by ionic bonds and finally removed by the ion exchange method, so theoretically the proposed method can be achieved by ionic monomers and ionic surfactants. Our manuscript also demonstrates the universality of this approach, first selecting two different cationic surfactants DTAB/OTAB to react with the anionic monomer DABA, demonstrating that changing the surfactant can still construct hierarchical ordered COFs. Later, we further used the anionic surfactant SLS to react with the cationic monomer TpPa-Cl and proved that the change of COF monomer and charge did not affect the applicability of the method.

The key to the success of this method lies in the binding of the surfactant to the COF backbone through ionic bonds, which avoids the incompatibility of surfactant self-assembly and COF crystallization. It is important to note that the strategy proposed in

this study serves as a starting point to inspire further developments in this field. We are exploring the potential extension of this strategy to a broader range of neutral COFs and the compatibility with non-ionic surfactants, including those with higher molecular weights. The central focus remains on enabling the self-assembly of surfactants/micelles and the crystallization of COFs while circumventing their inherent incompatibility. Therefore, while this work was going on, we were already thinking and starting to try. Our ongoing experimentation has revealed that without ionic bonding, direct mixing of COFs and surfactants/micelles results in separate heteronucleated products, as observed in our experiments. We experimented with mixed crystallization of neutral monomers and ionic surfactants using the synthetic pathway shown in Scheme R3-1 using a method consistent with our manuscript. The results show that due to the weak reversibility of imine in the aqueous phase, the product cannot be obtained by Pathway 2. Although the product of pathway 1 has good COF crystallization, it does not contain the self-assembly of surfactants, and the yield is still very low.

Scheme R3-1 Crystallization and synthesis of neutral monomer and ionic surfactant DTAB.

To address this challenge, we are considering two main avenues of exploration. Firstly, we are investigating the feasibility of establishing a linkage between surfactants/micelles, metal ions, and the COF skeleton through coordination bonds, and utilization of good coordination ability of COF structures. Additionally, we are exploring the concept of inducing charge in neutral COFs/surfactants/micelles, such as through local protonation, a strategy previously demonstrated in Metal-Organic Frameworks (MOFs) literature (Chem. Mater. 2012, 24, 2253-2255), to assess its applicability to COFs.

We've tried a strategy with local protonation. In this attempt, we dissolved P123 (polyethylene oxide–polypropylene oxide–polyethylene oxide, an amphiphilic triblock

neutral polymer) in an aqueous solution, and then adjusted the pH to 2-3 to achieve local protonation of micelles, inspired by the principles outlined in the aforementioned literature (Chem. Mater. 2012, 24, 2253-2255) (Fig. R3-1). Subsequently, we used the same synthesis method as in the present manuscript to mix and react them with monomers, to construct microporous COFs with an ordered micelle arrangement using the ionic bonds (Fig. R3-2). We found the presence of methylene in the corresponding FT-IR spectra, but we did not find an ordered micelle arrangement in the corresponding small-angle XRD, which means that we did not achieve our goal. The possible reasons are as follows: according to the literature, the size of the ordered micelle arrangement formed by P123 will be close to 14 nm, and the SEM images we have observed show that the material is heterogeneous and difficult to form a large size, which may hinder the formation of a large, ordered micelle arrangement. COFs that are larger and easier to crystallize have the potential to achieve our approach. In conclusion, we believe that this work will open a new chapter and provide a new and more universal way to construct ordered and tunable extrinsic porosity in COFs.

Fig. R3-1 Schematic model for the formation of mesoporous MOFs. The coordination numbers in the right figure did not reflect the real situation. (Chem. Mater. 2012, 24, 2253-2255)

Fig. R3-2 COFs constructed using locally protonated P123 as a template: a) 0.5-10° small-angle PXRD spectra, b) 2-10° magnified PXRD spectra, c) FT-IR spectra, and d) SEM image.

3. The authors should provide more detailed information about the synthesis of the COFs, including the specific reaction conditions, precursor materials, and purification methods. This will ensure reproducibility and facilitate further research in the field.

Response: We would like to thank the reviewer for the valuable suggestions to improve the quality of our manuscripts. Based on this recommendation, we have added detailed information on the synthesis of COFs to the manuscript, including specific reaction conditions, precursor materials, and purification methods (Pages 22-24).

List of additions:

Location	Revised
Manuscript Pages 22-24	Methods Chemicals All reagents and solvents were commercially available and used as received unless otherwise stated. The 2-hydroxy-1,3,5-benzenetricarbaldehyde (Sa) and 2,5-diaminobenzenesulfonic acid (DABA) were purchased from Shanghai Kylpharm Co. Ltd., China. Dodecyl trimethyl ammonium bromide (DTAB), octadecyl trimethyl ammonium bromide (OTAB), and sodium lauryl sulfonate (SLS) were purchased from Aladdin Chemistry Co. Ltd., China. $\text{UO}_2(\text{NO}_3)_2 \cdot 6\text{H}_2\text{O}$, $\text{Th}(\text{NO}_3)_4 \cdot 4\text{H}_2\text{O}$, $\text{La}(\text{NO}_3)_3 \cdot 6\text{H}_2\text{O}$, $\text{Ce}(\text{NO}_3)_3 \cdot 6\text{H}_2\text{O}$,

Location	Revised
	Nd(NO₃)₃·6H₂O, Sm(NO₃)₃·6H₂O, Gd(NO₃)₃·6H₂O, Dy(NO₃)₃·5H₂O, Ho(NO₃)₃·5H₂O, Yb(NO₃)₃·5H₂O and Lu(NO₃)₃·6H₂O were purchased from Hubei Chushengwei Chemical Co. Ltd., China. Synthesis of MPCOF Sa (89.1 mg, 0.5 mmol), DABA (141.2 mg, 0.75 mmol), H₂O (5 mL), and 6 M acetic acid (0.5 mL) were placed into a 15 mL pressure-resistant tube and ultrasonicated for 5 min to evenly disperse. Subsequently, the reaction was shielded with nitrogen and the sealed tubes were placed in a heated oven at 120 °C for 3 days. After the reaction was completed, the samples were collected by the filter device and washed with DMF, methanol, and ethanol, respectively. Finally, the sample was vacuum-dried at 50 °C to obtain a black microporous parent MPCOF (156.4 mg, 77%). Synthesis of OHMMCOF-DTAB/OHMMCOF-OTAB DTAB (231.3 mg, 0.75 mmol)/OTAB (294.4 mg, 0.75 mmol) was sonicated for 10 min into a 15 mL pressure-resistant tube containing H₂O (5 mL) to form micelles, followed by the addition of 2,5-diaminobenzenesulfonic acid (141.2 mg, 0.15 mmol) to continue sonicating for 5 min to fully react quaternary amine at the end of the micelles with the sulfonic acid group. Finally, 2-hydroxy-1,3,5-benzenetriacetaldehyde (89.1 mg, 0.5 mmol) and 6 M acetic acid (0.5 mL) were added and sonicated for 5 min to evenly disperse the mixture. The reaction was shielded with nitrogen and placed in a heated oven at 120 °C for 3 days. After the reaction was completed, the samples were collected by a filter device and washed with DMF, methanol, and ethanol, respectively. Finally, the samples were vacuum-dried at 50 °C to obtain black off-template pre-products OHMMCOF-DTAB/OHMMCOF-OTAB. The yield of OHMMCOF-DTAB was 73% (273.6 mg) and the actual concentration of DTAB involved in the reaction was calculated to be 0.38 mmol (51%), while the yield of OHMMCOF-OTAB was 75% (328.3 mg), and the actual concentration of OTAB involved in the reaction was calculated to be 0.33 mmol (44%). When calculating, the monomers involved in the construction of the COF backbone in the default product are consistent with MPCOF. Synthesis of OHMMCOF-1/OHMMCOF-2 For OHMMCOF-DTAB, 500 mg of sample was put into a round bottom flask filled with 250 mL of 0.5 M HCl aqueous solution, then heated and stirred at 60 °C for 1 d to remove the template. After the reaction was completed, the samples were collected by a filter device and washed with water and methanol. Finally, the sample was vacuum-dried at 50 °C to obtain an ordered hierarchical microporous/mesoporous OHMMCOF-1 (271.0 mg). The calculated template removal rate was 107%, and the error may be due to partial

Location	Revised
	damage to the sample by 0.5 M HCl aqueous solution. For OHMMCOF-OTAB, 500 mg of the sample was put into a round bottom flask filled with 250 mL of 5 M HCl aqueous solution, then heated and stirred at 60 °C for 1 d to remove the template. After the reaction was completed, the samples were collected by a filter device and washed with water and methanol. Finally, the sample was vacuum-dried at 50 °C to obtain an ordered hierarchical microporous/mesoporous OHMMCOF-2 (202.1 mg). The calculated template removal rate was 129%, and the error may be due to partial damage to the sample by 5 M HCl aqueous solution. Synthesis of OHMMCOF-3 SLS (61.2 mg, 0.225 mmol) was sonicated for 10 min into a 15 mL pressure-resistant tube containing H₂O (5 mL) to form micelles, followed by the addition of TGCl (31.5 mg, 0.225 mmol) to continue sonicating for 5 min to fully react the sulfonic acid group at the end of the micelle with the amino group. Finally, TP (47.4 mg, 0.225 mmol) and 6 M acetic acid (0.5 mL) were added and sonicated for 5 min to evenly disperse the mixture. Shield the reaction with nitrogen and place the sealed tube in a heated oven at 120 °C for 3 days. After the reaction was completed, the sample was collected by filtration and washed with DMF, methanol, and ethanol, respectively. Finally, the sample was vacuum-dried at 50 °C to obtain yellow off-template pre-products (80.4 mg, 70%). Subsequently, 80.4 mg of the sample was put into a round bottom flask filled with 100 mL of 0.5 M HCl aqueous solution, then heated and stirred at 60 °C for 1 d to remove the template. After the reaction was completed, the sample was collected by filtration and washed with water and methanol. Finally, the sample was vacuum-dried at 50 °C to obtain an ordered hierarchical microporous/mesoporous OHMMCOF-3 (48.9 mg).

4. PXRD shows that the COFs synthesized by the soft template method show a small angle diffraction peak, which cannot indicate the existence of ordered microporous/mesoporous structure, and more experimental data should be provided to support it.

Response: Many thanks to the reviewer for the advice. We agree that relying solely on the small-angle diffraction peaks on PXRD does not indicate the presence of ordered microporous/mesoporous structures. The most direct way to demonstrate ordered micropore/mesoporous structures is to analyze the nitrogen adsorption-desorption of

pores. Given the low specific surface area provided by our previous manuscript, in this revised version, we have removed as much water as possible from the pores by high-temperature activation at 150 °C, resulting in a higher specific surface area and more accurate pore parameters. The results are shown in Fig. R3-3. The newly synthesized COFs exhibited IV-type characteristic nitrogen adsorption isotherms consistent with previous manuscripts and obtained higher specific surface area. The presence of the H3-type hysteresis loop and the pore size distribution containing both micropores and mesopores indicate that they are ordered hierarchical microporous/mesoporous materials. In addition, the detailed microporous/mesoporous ratio of surface area further demonstrates the microporous/mesoporous structure of the material (Table R3-1). Detailed data updates have been highlighted in the manuscript.

Fig. R3-3 The nitrogen adsorption-desorption isotherms at 77 K: a) in the original manuscript after activation at 120 °C for 24 h and b) in the revised manuscript after high-temperature activation at 150 °C for 24 h, the corresponding pore size distributions: c) in the original manuscript after activation at 120 °C for 24 h and d) in the revised manuscript after high-temperature activation at 150 °C for 24 h.

Table R3-1 Pore volume parameters of MPCOF, OHMMCOF-DTAB/OTAB, and OHMMCOF-1/2/3.

COF	V_{total} (cm ³ /g)	$S_{\text{BET-micro}}$ (m ² /g)	$S_{\text{BET-meso}}$ (m ² /g)	$S_{\text{BET-meso}}/S_{\text{BET-micro}}$
-----	---	--	---	--

MPCOF	0.31	19.3	26.9	1.4
OHMMCOF-DTAB	0.17	15.6	20.7	1.3
OHMMCOF-OTAB	0.13	12.0	18.1	1.5
OHMMCOF-1	0.61	21.8	149.0	6.8
OHMMCOF-2	0.45	10.8	101.4	9.4
OHMMCOF-3	0.47	17.6	86.7	4.9

In addition, the structural characterization before and after template removal, including FT-IR, ^{13}C SSNMR, and XPS, also confirmed the formation of ordered hierarchical pores. At the same time, the increase of OHMMCOF-1/2 in U(VI)/Th(IV) adsorption capacity and adsorption rate compared with MPCOF also confirms the formation of ordered hierarchical pores. At last, we are also very interested in observing the presence of ordered mesopores through HR-TEM, as this is more direct evidence. In our previous experiments, we thought that the main reason why no mesopores were observed was that the pretreatment of our HR-TEM test was to preferentially use a cell pulverizer to make the material smaller and thinner, but the results showed that the material was still large and thick, which led to the fact that we did not observe the presence of ordered mesopores in multiple attempts. As a result, we resynthesized the ordered hierarchical porous COFs and changed the pretreatment for HR-TEM testing. We tried to observe the mesopores by slicing and thinning the sample, but the incompatibility of the sample with the encapsulant epoxy resin made the experiment impossible. Subsequently, we planned to further thin the sample and expose the ordered mesopores by comminuting the ultrasound for a long time, but the sample was still very thick after tens of hours of crushing, so we still did not observe the ordered mesopores. On the other hand, we believe that COFs are very sensitive to electron beams, and the ordered arrangement of micelles or mesopores is 'defective' compared to COFs, so they are more sensitive to electron beams, which is one of the reasons why we do not observe their orderly arrangement. We regret that we did not observe the ordered arrangement of micelles or mesopores under TEM, but this does not affect the construction of ordered hierarchical micropores/mesoporous COFs by the soft template method proposed in this work, because other experiments have demonstrated the ordered micropores/mesopores of the materials.

List of modifications:

Location	Original	Revised
Manuscript Page 11 Lines 1-6	 Figure 2. High-resolution XPS spectra of A) N 1s and B) S 2p, C) nitrogen adsorption-desorption isotherms at 77 K and D) the corresponding pore size distributions, and E) the HRTEM images. For all pictures: (a) MPCOF, (b) OHMMCOF-DTAB, and (c) OHMMCOF-OTAB.	 Fig. 2 Characterizations of the structure and porosity parameters of MPCOF, OHMMCOF-DTAB, and OHMMCOF-OTAB. High-resolution XPS spectra of A) N 1s and B) S 2p, C) nitrogen adsorption-desorption isotherms at 77 K and D) the corresponding pore size distributions, and E) the HRTEM images. For all pictures: (a) MPCOF, (b) OHMMCOF-DTAB, and (c) OHMMCOF-OTAB.
Manuscript Page 12 Line 27 and Page 13 Line 1	The BET surface areas of OHMMCOF-1/OHMMCOF-2 after removal of DTAB and OTAB increased to 99.6 and 37.9 m²/g compared to the OHMMCOF-DTAB/OHMMCOF-OTAB, respectively, which could be attributed to the removal of long-chain alkyl quaternary amine ions in the pores. The pore size distributions calculated by NLDFT revealed predominant pore sizes of 1.8 nm and additional mesoporous pore sizes of 3.3 nm for OHMMCOF-1, and 1.7 nm and 4.2 nm for OHMMCOF-2, which is very close to the theoretical values (Figure 1b).	The BET surface areas of OHMMCOF-1/OHMMCOF-2 after removal of DTAB and OTAB increased to 170.8 and 112.2 m²/g compared to the OHMMCOF-DTAB/OHMMCOF-OTAB, respectively, which could be attributed to the removal of long-chain alkyl quaternary amine ions in the pores. The pore size distributions calculated by NLDFT revealed predominant pore sizes of 1.8 nm and additional mesoporous pore sizes of 3.3 nm for OHMMCOF-1, and 1.8 nm and 4.3 nm for OHMMCOF-2, which is very close to the theoretical values (Fig. 1b).

Location	Original	Revised
Manuscript Page 13 Lines 11-17	 Figure 3. A) Illustration of template removal using ion exchange method, B) PXRD patterns, C) FT-IR spectra, D) ¹³C SSNMR spectra, high-resolution XPS spectra of E) N 1s and F) S 2p, G) nitrogen adsorption-desorption isotherms at 77 K and H) the corresponding pore size distributions. For pictures B-H: (a) OHMMCOF-1 and (b) OHMMCOF-2.	 Fig. 3 Characterizations of crystallinity, structure, and porosity parameters of OHMMCOF-1 and OHMMCOF-2. A) Illustration of template removal using ion exchange method, B) PXRD patterns, C) FT-IR spectra, D) ¹³C SSNMR spectra (where the asterisks (*) represent the spinning sidebands), high-resolution XPS spectra of E) N 1s and F) S 2p, G) nitrogen adsorption-desorption isotherms at 77 K and H) the corresponding pore size distributions. For pictures B-H: (a) OHMMCOF-1 and (b) OHMMCOF-2.
Manuscript Page 14 Lines 23-30 and Page 15 Lines 1-7	These results showed that the template proportion in the final COFs increased with the increase of their concentration, and this phenomenon was especially obvious when using the less hydrophobic DTAB as the template. The difference in solubility between DTAB and OTAB contributed to their distinct behavior in the reaction. The higher solubility of DTAB allowed for a continued increase in its actual reaction concentration, while the lower solubility of OTAB limited its participation in the reaction (Figure S33).	These results showed that the template proportion in the final COFs increased with the increase of their concentration, implying that the mesoporous/microporous ratio can also be easily adjusted by this pathway, and this phenomenon was especially obvious when using the less hydrophobic DTAB as the template. The difference in solubility between DTAB and OTAB contributed to their distinct behavior in the reaction. The higher solubility of DTAB allowed for a continued increase in its actual reaction concentration, while the lower solubility of OTAB limited its participation in the reaction (Supplementary Fig. 27). OHMMCOF-DTAB-0.5 eq and OHMMCOF-DTAB-

Location	Original	Revised
		3.0 eq were selected for template removal experiments, followed by N₂ adsorption-desorption tests, and compared with OHMMCOF-DTAB-1.0 eq to verify the above conjectures. OHMMCOF-1-0.5 eq exhibited slightly lower specific surface area (159.0 m²/g) and $S_{\text{BET-meso}}/S_{\text{BET-micro}}$ (5.5) than OHMMCOF-1-1.0 eq (170.8 m²/g and 6.8), while OHMMCOF-1-3.0 eq exhibited higher specific surface area (178.1 m²/g) and $S_{\text{BET-meso}}/S_{\text{BET-micro}}$ (7.5) (Fig. 4C-D and Supplementary Fig. 28, Table 3, 6).
Manuscript Page 15 Lines 24-30 and Page 16 Lines 1-4	Although changing the ratio of DTAB/OTAB in the system could not synthesize ordered hierarchically microporous/mesoporous COFs with two different mesopores, it was unexpectedly found that the micelle size in the final COFs could be controlled, which opens up new possibilities for more straightforward changes in mesopore size.	Although changing the ratio of DTAB/OTAB in the system could not synthesize ordered hierarchically microporous/mesoporous COFs with two different mesopores, it was unexpectedly found that the micelle size in the final COFs could be controlled, which means that this may be a new way to control the mesoporous size. Two materials with DTAB/OTAB of 14:1 and 1:2 was selected for the template removal experiment, and the resulting COFs were named OHMMCOF-M-14:1 and OHMMCOF-M-1:2 (while M represents the mixture), respectively, and then N₂ adsorption-desorption experiments were performed to verify the above conjecture (Fig. 4F and Supplementary Fig. 29, Table 7). The mesoporous pore size of OHMMCOF-M-14:1 was mainly 3.6 nm, which was relatively close to that of the mesopores constructed entirely with DTAB self-assembled micelles (3.3 nm), while the mesoporous pore size of OHMMCOF-M-1:2 was mainly 4.5 nm, which was also relatively close to the mesopores

Location	Original	Revised
		constructed entirely with OTAB self-assembled micelles (4.3 nm) (Fig. 4G).
Manuscript Page 16 Lines 20, 23	As shown in Figure S24, OHMMCOF-3 also exhibited IV-type characteristic nitrogen adsorption isotherms as well as H3-type hysteresis loop, indicating the presence of mesopores in the material. The specific surface area was measured to be 26.9 m²/g, which was relatively low due to the large number of anions present in the framework. However, NLDFT calculation revealed the presence of micropores with a size of 1.3 nm, closer to the theoretical value. Additionally, the removal of spherical micelles constructed with SLS resulted in the formation of additional mesopores with a size of 3.3 nm.⁷¹	As shown in Fig. 4I and Supplementary Fig. 30, OHMMCOF-3 also exhibited IV-type characteristic nitrogen adsorption isotherms as well as H3-type hysteresis loop, indicating the presence of mesopores in the material. The specific surface area was measured to be 104.3 m²/g, which was relatively low due to the large number of anions present in the framework. However, NLDFT calculation revealed the presence of micropores with a size of 1.3 nm, closer to the theoretical value. Additionally, the removal of spherical micelles constructed with SLS resulted in the formation of additional mesopores with a size of 3.3 nm.⁷¹
Manuscript Page 17 Lines 1-10	 Figure 5. PXRD patterns of A) OHMMCOF-DTAB and B) OHMMCOF-OTAB corresponding to different input concentration, adsorption kinetics at C) 24 h and D) 10 min, for pictures C and D: (a) OHMMCOF-1-0.5 eq-U, (b) OHMMCOF-1-0.5 eq-Th, (c) OHMMCOF-1-3.0 eq-U, and (d) OHMMCOF-1-3.0 eq-Th, E) PXRD patterns corresponding to templates with different concentration ratios of DTAB/OTAB, F) simulated and experimental PXRD patterns (inset: the simulated size of the SLS micelles) of OHMMCOF-3.	 Fig. 4 Pore regulation and system expansion of OHMMCOFs. PXRD patterns of A) OHMMCOF-DTAB and B) OHMMCOF-OTAB corresponding to different input concentration, C) nitrogen adsorption-desorption isotherms at 77 K and D) the corresponding pore size distributions, for C and D: a) OHMMCOF-1-0.5 eq and b) OHMMCOF-1-3.0 eq, E) PXRD patterns corresponding to templates with different concentration ratios of DTAB/OTAB, F) nitrogen

Location	Original	Revised
		adsorption-desorption isotherms at 77 K and G) the corresponding pore size distributions, for F and G: a) OHMMCOF-M-14:1 and b) OHMMCOF-M-1:2, H) simulated and experimental PXRD patterns of OHMMCOF-3 (inset: the simulated size of the SLS micelles), and I) nitrogen adsorption-desorption isotherms at 77 K of OHMMCOF-3 (inset: the corresponding pore size distributions).

5. As the ordered crystal structure of COFs, microporous/mesoporous COFs can also be synthesized without soft template method. What are the advantages of introducing the soft template method to synthesize ordered hierarchical microporous/mesoporous COFs?

Response: Thank you very much for the question from the reviewer. We agree with your statement that ordered microporous/mesoporous or microporous/macroporous COFs can be synthesized without the need for a soft template method, as we mentioned in the introduction of our manuscript, the use of the hard template method usually requires the preparation of the hard template in advance, and the synthesis process is more complex. While the monomer design-based method has successfully introduced ordered hierarchical mesopores and micropores into the COF structure, the synthesis of these monomers is still challenging. Furthermore, the relative size and proportion of micropores and mesopores are constrained by the monomers, making them challenging to control. Therefore, although the above methods can complete the construction of ordered additional pores, their adjustability and versatility are still insufficient.

In our work, a soft template method was developed to construct ordered microporous/mesoporous COFs. The soft template method, which relies on ionic interactions to avoid the incompatibility of surfactant self-assembly with COF

crystallization, theoretically allows any charged monomer and surfactant to construct ordered microporous/mesoporous structures by this method. In addition, in response to your first question, we also mentioned that neutral COFs/surfactants (even high molecular weight micelles) are also expected to achieve the construction of ordered additional pores through our proposed soft template method, so this method is more generalizable. On the other hand, when constructing additional pores, we can easily control the additional porosity by changing the type of surfactant, the input concentration, and the ratio of different surfactants, which is much simpler and faster than the previously mentioned methods. Finally, it is worth mentioning that surfactants are very inexpensive, and their assembly relies on hydrophobic interactions, so the use of solvent water makes this more environmentally friendly method a promising engineering application.

6. SEM and TEM characterizations of OHMMCOF and OHMMCOF do not show the ordered morphology, how to explain the ordered hierarchical structure of COFs synthesized by soft template method?

Response: We appreciate the questions raised by the reviewers and apologize that we did not answer your questions about the content of our previous manuscript. We understand your concerns, as ordered hierarchies have been observed in SEM or TEM in the literature. For example, Su et al. and Zhao et al. used the soft template method to synthesize a large number of inorganic or organic materials that can observe ordered structures in SEM or TEM images, respectively (Chem. Soc. Rev. 2017, 46, 481-558; Chem. Rev. 2021, 121, 14349-14429), and Gu et al. also used this method to synthesize a large number of microporous/mesoporous MOFs that can observe ordered structures in SEM or TEM images (Acc. Chem. Res. 2022, 55, 2235-2247). It was the work of these predecessors that inspired and prompted us to investigate whether the soft template method could also be used to construct ordered hierarchical microporous/mesoporous COFs, which is the origin of this work.

However, unlike conventional inorganic materials and MOFs, the crystallization of

COFs is completely composed of organic monomers, so its morphology is easily affected by weak interactions between monomers and monomers/solvents, so it is very difficult to show ordered COFs in SEM images. This can be seen from the current development of single-crystalline and morphology control of COFs. In addition, the ordered mesoporous or macroporous structures constructed by previous predecessors that can be observed in SEM often mean that the pore size has reached tens of nanometers, while the additional mesopores constructed in this work are only 3-5 nm, so it cannot be observed in SEM images. On the other hand, it is theoretically possible to see the ordered microporous/mesoporous structure in TEM. As described in the main text and in the Supplementary information, the ordered structure of COFs is observed in all corresponding TEM images. However, we have not observed an orderly micelle arrangement or additional mesoporous structure in many previous experiments, which is also a big regret for us. Based on this, we tried to observe the mesopores by slicing and thinning the sample, but the incompatibility of the sample with the encapsulant epoxy resin made the experiment impossible. Subsequently, we planned to further thin the sample and expose the ordered mesopores by comminuting the ultrasound for a long time, but the sample was still very thick after tens of hours of crushing, so we still did not observe the ordered mesopores. On the other hand, we believe that COFs are very sensitive to electron beams, and the ordered arrangement of micelles or mesopores is 'defective' compared to COFs, so they are more sensitive to electron beams, which is one of the reasons why we do not observe their orderly arrangement. We regret that we did not observe the ordered arrangement of micelles or mesopores under TEM, but this does not affect the construction of ordered hierarchical micropores/mesoporous COFs by the soft template method proposed in this work, because other experiments including nitrogen adsorption-desorption, FT-IR, ^{13}C SSNMR, XPS, and subsequent U(VI)/Th(IV) adsorption experiments have demonstrated the ordered micropores/mesopores of the materials.

7. BET test of the COFs synthesized by soft template method showed relatively small specific surface area, is the template not removed clean? How to prove that the template

has been removed clean? Does the residual template affect the XRD diffraction peak?

Response: We appreciate the questions raised by the reviewers and apologize for the low specific surface area of the previous material. According to the results of our experiments, the template of the material was completely removed, which can be determined by the many characterizations after the removal of the template, which we have described in the article ‘FT-IR and ^{13}C SSNMR were used to characterize the structural changes in OHMMCOF-1/OHMMCOF-2 after template removal (Figs. 3C-D). The C=N ($\sim 1606\text{ cm}^{-1}$ in FT-IR and $\sim 158\text{ ppm}$ in ^{13}C SSNMR), C=O ($\sim 1683\text{ cm}^{-1}$ in FT-IR and $\sim 198\text{ ppm}$ in ^{13}C SSNMR) and S-OH (1024 cm^{-1} in FT-IR and $\sim 124\text{ ppm}$ in ^{13}C SSNMR) underwent a weak redshift after the removal of the templates, and methylene groups disappeared at the same time. In addition, the XPS high-resolution spectra also showed a faint redshift for C=N (399.3 eV, N 1s), C=O (530.9 eV, O 1s), and S 2p (S $2p_{3/2}$ = 168.6 eV and S $2p_{1/2}$ = 167.4 eV), while C-N⁺ disappeared at the same time (Figs. 3E-F and Supplementary Figs. 17-18).’ ‘EA showed that the C/N ratios of the two OHMMCOF-1/OHMMCOF-2 were significantly reduced due to the removal of long-chain alkyl groups compared to OHMMCOF-DTAB/OHMMCOF-OTAB (Supplementary Table 4). Increased hydrophilicity and the absence of a weightless platform at around 270-360 °C in TGA further confirmed the removal of the templates (Supplementary Figs. 20-21).’.

The reasons for our low specific surface area are mainly the following two points. Firstly, it may be that the widely free anions/cations in the pores, which is also the key factor for the generally low BET of other reported ions COFs (ACS Appl. Mater. Interfaces 2016, 8, 18505-18512: $69\text{ m}^2\text{ g}^{-1}$ for NUS-10(R); Chem. Mater. 2016, 28, 1489-1494: $215\text{ m}^2\text{ g}^{-1}$ for TpPa-SO₃H; J. Am. Chem. Soc. 2018, 140, 896-899: $220\text{ m}^2\text{ g}^{-1}$ for CON-Cl; Chem. Commun. 2023, 59, 14435-14438: 53, 103 and $113\text{ m}^2\text{ g}^{-1}$ for C4-IL/ICOF, C6-IL/ICOF and C10-IL/ICOF, respectively; J. Am. Chem. Soc. 2024, 146, 2313-2318: $57\text{ m}^2\text{ g}^{-1}$ for TpPa-SO₃H). On the other hand, it may be that our activation process is not complete, resulting in the presence of water or some solvents in the pores. Sulfonic acid is highly hydrophilic, which leads to some pores

being occupied by water molecules, thus reducing the specific surface area. Additionally, according to the research results of Zhao Dan et al., the introduction of guest molecules may promote the interlayer sliding of 2D COFs (J. Am. Chem. Soc. 2023, 145, 1359-1366; J. Am. Chem. Soc. 2020, 142, 12995-13002), increasing the disorder of the system and further reducing the material's specific surface area. Therefore, we resynthesized the COFs and pretreated the material under vacuum conditions with an activation temperature of 150 °C, and then tested their specific surface area, the results of which are shown in Fig. R3-4. All activated COFs showed higher BET values than the original manuscript, further verifying the significance of hierarchical porous COFs.

Fig. R3-4 The nitrogen adsorption-desorption isotherms at 77 K: a) in the original manuscript after activation at 120 °C for 24 h and b) in the revised manuscript after high-temperature activation at 150 °C for 24 h.

The template has been completely removed, so there is no residual template in OHMMCOF-1/2/3 after template removal, and the XRD diffraction peaks have not been affected by them.

List of modifications:

Location	Original	Revised
Manuscript Page 11 Lines 1-6	Figure 2. High-resolution XPS	

Location	Original	Revised
	spectra of A) N 1s and B) S 2p, C) nitrogen adsorption-desorption isotherms at 77 K and D) the corresponding pore size distributions, and E) the HRTEM images. For all pictures: (a) MPCOF, (b) OHMMCOF-DTAB, and (c) OHMMCOF-OTAB.	Fig. 2 Characterizations of the structure and porosity parameters of MPCOF, OHMMCOF-DTAB, and OHMMCOF-OTAB. High-resolution XPS spectra of A) N 1s and B) S 2p, C) nitrogen adsorption-desorption isotherms at 77 K and D) the corresponding pore size distributions, and E) the HRTEM images. For all pictures: (a) MPCOF, (b) OHMMCOF-DTAB, and (c) OHMMCOF-OTAB.
Manuscript Page 12 Line 27 and Page 13 Line 1	The BET surface areas of OHMMCOF-1/OHMMCOF-2 after removal of DTAB and OTAB increased to 99.6 and 37.9 m²/g compared to the OHMMCOF-DTAB/OHMMCOF-OTAB, respectively, which could be attributed to the removal of long-chain alkyl quaternary amine ions in the pores. The pore size distributions calculated by NLDFT revealed predominant pore sizes of 1.8 nm and additional mesoporous pore sizes of 3.3 nm for OHMMCOF-1, and 1.7 nm and 4.2 nm for OHMMCOF-2, which is very close to the theoretical values (Figure 1b).	The BET surface areas of OHMMCOF-1/OHMMCOF-2 after removal of DTAB and OTAB increased to 170.8 and 112.2 m²/g compared to the OHMMCOF-DTAB/OHMMCOF-OTAB, respectively, which could be attributed to the removal of long-chain alkyl quaternary amine ions in the pores. The pore size distributions calculated by NLDFT revealed predominant pore sizes of 1.8 nm and additional mesoporous pore sizes of 3.3 nm for OHMMCOF-1, and 1.8 nm and 4.3 nm for OHMMCOF-2, which is very close to the theoretical values (Fig. 1b).
Manuscript Page 13 Lines 11-17	 Figure 3. A) Illustration of template removal using ion exchange method, B) PXRD patterns, C) FT-IR spectra, D) ¹³C SSNMR spectra, high-resolution XPS spectra of E) N 1s and F) S 2p, G) nitrogen adsorption-	 Fig. 3 Characterizations of crystallinity, structure, and porosity parameters of OHMMCOF-1 and OHMMCOF-2. A) Illustration of template removal using ion exchange method, B) PXRD patterns, C) FT-IR

Location	Original	Revised
	desorption isotherms at 77 K and H) the corresponding pore size distributions. For pictures B-H: (a) OHMMCOF-1 and (b) OHMMCOF-2.	spectra, D) ¹³C SSNMR spectra (where the asterisks (*) represent the spinning sidebands), high-resolution XPS spectra of E) N 1s and F) S 2p, G) nitrogen adsorption-desorption isotherms at 77 K and H) the corresponding pore size distributions. For pictures B-H: (a) OHMMCOF-1 and (b) OHMMCOF-2.
Manuscript Page 14 Lines 23-30 and Page 15 Lines 1-7	These results showed that the template proportion in the final COFs increased with the increase of their concentration, and this phenomenon was especially obvious when using the less hydrophobic DTAB as the template. The difference in solubility between DTAB and OTAB contributed to their distinct behavior in the reaction. The higher solubility of DTAB allowed for a continued increase in its actual reaction concentration, while the lower solubility of OTAB limited its participation in the reaction (Figure S33).	These results showed that the template proportion in the final COFs increased with the increase of their concentration, implying that the mesoporous/microporous ratio can also be easily adjusted by this pathway, and this phenomenon was especially obvious when using the less hydrophobic DTAB as the template. The difference in solubility between DTAB and OTAB contributed to their distinct behavior in the reaction. The higher solubility of DTAB allowed for a continued increase in its actual reaction concentration, while the lower solubility of OTAB limited its participation in the reaction (Supplementary Fig. 27). OHMMCOF-DTAB-0.5 eq and OHMMCOF-DTAB-3.0 eq were selected for template removal experiments, followed by N₂ adsorption-desorption tests, and compared with OHMMCOF-DTAB-1.0 eq to verify the above conjectures. OHMMCOF-1-0.5 eq exhibited slightly lower specific surface area (159.0 m²/g) and S_{BET-meso}/S_{BET-micro} (5.5) than OHMMCOF-1-1.0 eq (170.8 m²/g and 6.8), while OHMMCOF-1-3.0 eq exhibited higher specific surface area (178.1 m²/g) and S_{BET-meso}/S_{BET-micro} (7.5) (Fig. 4C-D and Supplementary Fig. 28, Table 3, 6).

Location	Original	Revised
Manuscript Page 15 Lines 24-30 and Page 16 Lines 1-4	Although changing the ratio of DTAB/OTAB in the system could not synthesize ordered hierarchically microporous/mesoporous COFs with two different mesopores, it was unexpectedly found that the micelle size in the final COFs could be controlled, which opens up new possibilities for more straightforward changes in mesopore size.	Although changing the ratio of DTAB/OTAB in the system could not synthesize ordered hierarchically microporous/mesoporous COFs with two different mesopores, it was unexpectedly found that the micelle size in the final COFs could be controlled, which means that this may be a new way to control the mesoporous size. Two materials with DTAB/OTAB of 14:1 and 1:2 was selected for the template removal experiment, and the resulting COFs were named OHMMCOF-M-14:1 and OHMMCOF-M-1:2 (while M represents the mixture), respectively, and then N₂ adsorption-desorption experiments were performed to verify the above conjecture (Fig. 4F and Supplementary Fig. 29, Table 7). The mesoporous pore size of OHMMCOF-M-14:1 was mainly 3.6 nm, which was relatively close to that of the mesopores constructed entirely with DTAB self-assembled micelles (3.3 nm), while the mesoporous pore size of OHMMCOF-M-1:2 was mainly 4.5 nm, which was also relatively close to the mesopores constructed entirely with OTAB self-assembled micelles (4.3 nm) (Fig. 4G).
Manuscript Page 16 Lines 20, 23	As shown in Figure S24, OHMMCOF-3 also exhibited IV-type characteristic nitrogen adsorption isotherms as well as H3-type hysteresis loop, indicating the presence of mesopores in the material. The specific surface area was measured to be 26.9 m²/g, which was relatively low due to the large number of anions present in the framework. However, NLDFT calculation revealed the presence of micropores	As shown in Fig. 4I and Supplementary Fig. 30, OHMMCOF-3 also exhibited IV-type characteristic nitrogen adsorption isotherms as well as H3-type hysteresis loop, indicating the presence of mesopores in the material. The specific surface area was measured to be 104.3 m²/g, which was relatively low due to the large number of anions present in the framework. However, NLDFT calculation revealed the presence of micropores with a size of

Location	Original	Revised
	with a size of 1.3 nm, closer to the theoretical value. Additionally, the removal of spherical micelles constructed with SLS resulted in the formation of additional mesopores with a size of 3.3 nm.⁷¹	1.3 nm, closer to the theoretical value. Additionally, the removal of spherical micelles constructed with SLS resulted in the formation of additional mesopores with a size of 3.3 nm.⁷¹
Manuscript Page 17 Lines 1-10	 Figure 5. PXRD patterns of A) OHMMCOF-DTAB and B) OHMMCOF-OTAB corresponding to different input concentration, adsorption kinetics at C) 24 h and D) 10 min, for pictures C and D: (a) OHMMCOF-1-0.5 eq-U, (b) OHMMCOF-1-0.5 eq-Th, (c) OHMMCOF-1-3.0 eq-U, and (d) OHMMCOF-1-3.0 eq-Th, E) PXRD patterns corresponding to templates with different concentration ratios of DTAB/OTAB, F) simulated and experimental PXRD patterns (inset: the simulated size of the SLS micelles) of OHMMCOF-3.	 Fig. 4 Pore regulation and system expansion of OHMMCOFs. PXRD patterns of A) OHMMCOF-DTAB and B) OHMMCOF-OTAB corresponding to different input concentration, C) nitrogen adsorption-desorption isotherms at 77 K and D) the corresponding pore size distributions, for C and D: a) OHMMCOF-1-0.5 eq and b) OHMMCOF-1-3.0 eq, E) PXRD patterns corresponding to templates with different concentration ratios of DTAB/OTAB, F) nitrogen adsorption-desorption isotherms at 77 K and G) the corresponding pore size distributions, for F and G: a) OHMMCOF-M-14:1 and b) OHMMCOF-M-1:2, H) simulated and experimental PXRD patterns of OHMMCOF-3 (inset: the simulated size of the SLS micelles), and I) nitrogen adsorption-desorption isotherms at 77 K of OHMMCOF-3 (inset: the corresponding pore size distributions).

8. How to control the ordered hierarchical microporous/mesoporous of COF using soft template? As the main innovation point of this paper, this mechanism should be analyzed clearly.

Response: We are very grateful for the reviewers' suggestions, and we also agree that our previous manuscript had some shortcomings in how to control the ordered hierarchical micropores/mesopores of COFs using soft templates. In our previous manuscript, we proposed that we could first alter the mesoporous size of OHMMCOFs by changing the alkyl chain length of the surfactant, which has been confirmed by PXRD and the pore size distribution of the materials 'The BET surface areas of OHMMCOF-1/OHMMCOF-2 after removal of DTAB and OTAB increased to 170.8 and 112.2 m²/g compared to the OHMMCOF-DTAB/OHMMCOF-OTAB, respectively, which could be attributed to the removal of long-chain alkyl quaternary amine ions in the pores (Supplementary Fig. 19 and Table 3). The pore size distributions calculated by NLDFT revealed predominant pore sizes of 1.8 nm and additional mesoporous pore sizes of 3.3 nm for OHMMCOF-1, and 1.8 nm and 4.3 nm for OHMMCOF-2, which is very close to the theoretical values (Fig. 1B)'.

After determining that the strategy we designed could effectively construct ordered hierarchically microporous/mesoporous COFs, we studied the effect of template concentration. The results showed that the 2θ value of the corresponding self-assembly peak remained unchanged as the concentration of DTAB increased, but the intensity gradually increased compared with the (100) crystal plane of COFs, indicating an increase in the actual concentration of DTAB involved in the reaction, which we speculated that the ratio of mesoporous/micropores of OHMMCOFs after template removal may also be gradually increasing, but we did not give the relevant pore parameters at that time. In this supplementary experiment, we resynthesized the relevant samples and performed nitrogen adsorption-desorption tests. The pore parameters of the two samples after removing the template at DTAB concentrations of 0.5 eq and 3.0 eq were obtained, and the results are shown in Fig. R3-5 and Table R3-2. OHMMCOF-1-0.5 eq exhibited slightly lower specific surface area (159.0 m²/g) and

$S_{BET-meso}/S_{BET-micro}$ (5.5) than OHMMCOF-1-1.0 eq (170.8 m^2/g and 6.8), while OHMMCOF-1-3.0 eq exhibited higher specific surface area (178.1 m^2/g) and $S_{BET-meso}/S_{BET-micro}$ (7.5). This confirms our conjecture and illustrates that our proposed method can indeed change the ratio of mesoporous/micropores in the structure of COFs by varying the concentration of surfactant.

Fig. R3-5. A) Nitrogen adsorption-desorption isotherms at 77 K and B) the corresponding pore size distributions: a) OHMMCOF-1-0.5 eq and b) OHMMCOF-1-3.0 eq.

Table R3-2. Pore volume parameters of OHMMCOF-1-0.5/3.0 eq.

COF	V_{total} (cm^3/g)	$S_{BET-micro}$ (m^2/g)	$S_{BET-meso}$ (m^2/g)	$S_{BET-meso}/S_{BET-micro}$
OHMMCOF-1-0.5 eq	0.59	24.5	134.5	5.5
OHMMCOF-1-3.0 eq	0.78	21.0	157.1	7.5

After confirming that changes in template concentration can affect the mesoporous/microporous ratio, we tried to add different ratios of DTAB/OTAB during the synthesis process, hoping to construct trimodal ordered hierarchically microporous/mesoporous COFs with two different mesopores. The results show that when OTAB accounted for more than 1/2 of the system, the 2θ value of the ordered structure in the XRD pattern is 2.0° , corresponding to a d-space of 4.3 nm. As the OTAB concentration gradually decreased in the system, both the 2θ value and the FWHM showed a controllable gradual decrease, indicating the size of the orderly arrangement micelles in the material structure is decreasing, and finally close to the size (3.3 nm) of the micelles formed by the DTAB template, which was also supported by the continuous enhancement of the relative strength of the (100) crystal plane peak from COFs. Although changing the ratio of DTAB/OTAB in the system could not synthesize

ordered hierarchically microporous/mesoporous COFs with two different mesopores, it was unexpectedly found that the micelle size in the final COFs could be controlled. This means that this may be a new way to control the mesoporous size, but we did not give the relevant pore parameters at that time. In this experiment, we resynthesized the relevant samples and performed nitrogen adsorption-desorption tests. Two materials with DTAB/OTAB of 14:1 and 1:2 were selected for the template removal experiment, and the resulting COFs were named OHMMCOF-M-14:1 and OHMMCOF-M-1:2 (while M represents the mixture), respectively (Fig. R3-6 and Table R3-3). The mesoporous pore size of OHMMCOF-M-14:1 was mainly 3.6 nm, which was relatively close to that of the mesopores constructed entirely with DTAB self-assembled micelles (3.3 nm), while the mesoporous pore size of OHMMCOF-M-1:2 was mainly 4.5 nm, which was also relatively close to the mesopores constructed entirely with OTAB self-assembled micelles (4.3 nm). This also confirms our conjecture and illustrates that our proposed method can indeed change the size of the mesopores in the COF structure by changing the mixing ratio of different surfactants.

Fig. R3-6. A) Nitrogen adsorption-desorption isotherms at 77 K and B) the corresponding pore size distributions: a) OHMMCOF-M-14:1 and b) OHMMCOF-M-1:2.

Table R3-3. Pore volume parameters of OHMMCOF-M-14:1/1:2.

COF	V_{total} (cm^3/g)	$S_{\text{BET-micro}}$ (m^2/g)	$S_{\text{BET-meso}}$ (m^2/g)	$S_{\text{BET-meso}}/S_{\text{BET-micro}}$
OHMMCOF-M-14:1	0.44	23.5	147.7	6.3
OHMMCOF-M-1:2	0.35	14.5	120.1	8.3

Detailed data updates have been highlighted in the manuscript of the response, please check it out.

List of modifications:

Located	Original	Revised
Manuscript Page 14 Lines 23-30 and Page 15 Lines 1-7	These results showed that the template proportion in the final COFs increased with the increase of their concentration, and this phenomenon was especially obvious when using the less hydrophobic DTAB as the template. The difference in solubility between DTAB and OTAB contributed to their distinct behavior in the reaction. The higher solubility of DTAB allowed for a continued increase in its actual reaction concentration, while the lower solubility of OTAB limited its participation in the reaction (Figure S33).	These results showed that the template proportion in the final COFs increased with the increase of their concentration, implying that the mesoporous/microporous ratio can also be easily adjusted by this pathway, and this phenomenon was especially obvious when using the less hydrophobic DTAB as the template. The difference in solubility between DTAB and OTAB contributed to their distinct behavior in the reaction. The higher solubility of DTAB allowed for a continued increase in its actual reaction concentration, while the lower solubility of OTAB limited its participation in the reaction (Supplementary Fig. 27). OHMMCOF-DTAB-0.5 eq and OHMMCOF-DTAB-3.0 eq were selected for template removal experiments, followed by N₂ adsorption-desorption tests, and compared with OHMMCOF-DTAB-1.0 eq to verify the above conjectures. OHMMCOF-1-0.5 eq exhibited slightly lower specific surface area (159.0 m²/g) and S_{BET-meso}/S_{BET-micro} (5.5) than OHMMCOF-1-1.0 eq (170.8 m²/g and 6.8), while OHMMCOF-1-3.0 eq exhibited higher specific surface area (178.1 m²/g) and S_{BET-meso}/S_{BET-micro} (7.5) (Fig. 4C-D and Supplementary Fig. 28, Table 3, 6).
Manuscript Page 15 Lines 24-30 and Page 16 Lines 1-4	Although changing the ratio of DTAB/OTAB in the system could not synthesize ordered hierarchically microporous/mesoporous COFs with two different mesopores, it was unexpectedly found that the micelle	Although changing the ratio of DTAB/OTAB in the system could not synthesize ordered hierarchically microporous/mesoporous COFs with two different mesopores, it was unexpectedly found that the micelle size

	size in the final COFs could be controlled, which opens up new possibilities for more straightforward changes in mesopore size.	in the final COFs could be controlled, which means that this may be a new way to control the mesoporous size. Two materials with DTAB/OTAB of 14:1 and 1:2 was selected for the template removal experiment, and the resulting COFs were named OHMMCOF-M-14:1 and OHMMCOF-M-1:2 (while M represents the mixture), respectively, and then N₂ adsorption-desorption experiments were performed to verify the above conjecture (Fig. 4F and Supplementary Fig. 29, Table 7). The mesoporous pore size of OHMMCOF-M-14:1 was mainly 3.6 nm, which was relatively close to that of the mesopores constructed entirely with DTAB self-assembled micelles (3.3 nm), while the mesoporous pore size of OHMMCOF-M-1:2 was mainly 4.5 nm, which was also relatively close to the mesopores constructed entirely with OTAB self-assembled micelles (4.3 nm) (Fig. 4G).
--	--	---

REVIEWERS' COMMENTS

Reviewer #1 (Remarks to the Author):

The authors have mostly addressed the concerns by this reviewer. I would like to recommend the publication of this manuscript in Nature Communcations.

Reviewer #2 (Remarks to the Author):

The authors improved the manuscript over their original version. I recommend publishing the manuscript.

Reviewer #3 (Remarks to the Author):

The concerns have been addressed and it is now acceptable.